# A cytosolic surveillance mechanism activates the mitochondrial UPR

F. X. Reymond Sutandy[1,2], Ines Gößner[1,2], Georg Tascher[1] & Christian Münch[1✉]

The mitochondrial unfolded protein response (UPR[mt]) is essential to safeguard mitochondria from proteotoxic damage by activating a dedicated transcriptional response in the nucleus to restore proteostasis[1,2]. Yet, it remains unclear how the information on mitochondria misfolding stress (MMS) is signalled to the nucleus as part of the human UPR[mt] (refs. 3,4). Here, we show that UPR[mt] signalling is driven by the release of two individual signals in the cytosol−mitochondrial reactive oxygen species (mtROS) and accumulation of mitochondrial protein precursors in the cytosol (c-mtProt). Combining proteomics and genetic approaches, we identified that MMS causes the release of mtROS into the cytosol. In parallel, MMS leads to mitochondrial protein import defects causing c-mtProt accumulation. Both signals integrate to activate the UPR[mt]; released mtROS oxidize the cytosolic HSP40 protein DNAJA1, which leads to enhanced recruitment of cytosolic HSP70 to c-mtProt. Consequently, HSP70 releases HSF1, which translocates to the nucleus and activates transcription of UPR[mt] genes. Together, we identify a highly controlled cytosolic surveillance mechanism that integrates independent mitochondrial stress signals to initiate the UPR[mt]. These observations reveal a link between mitochondrial and cytosolic proteostasis and provide molecular insight into UPR[mt] signalling in human cells.

Maintenance of mitochondrial protein homoeostasis is crucial for mitochondrial function. Upon proteotoxic stress, mitochondria activate the mitochondrial unfolded protein response (UPR[mt]), a nuclear transcriptional response that induces mitochondrial chaperones, such as *HSPD1*, *HSPE1* and *HSPA9*, and proteases, including *LONP1*, to re-establish homoeostasis in mitochondria[2,3,5]. The molecular events underlying the retrograde mitochondria−nucleus communication to induce the UPR[mt] in humans remain unclear. The integrated stress response (ISR) has been shown to contribute to the cellular rearrangements observed during the UPR[mt] and during mitochondrial stress responses in general[2,6–8]. However, recent findings indicated that the mitochondrial stress response/ISR and the UPR[mt] are two independent processes that are part of a more complex stress response[1,9–11].

To study the role of the ISR in UPR[mt] signalling, we monitored the early responses to mitochondrial misfolding stress (MMS). Treatment with the mitochondrial HSP90 inhibitor gamitrinib-triphenylphosphonium (GTPP)[1] causes MMS and significantly induced the UPR[mt] genes within 2−3 h but not general mitochondrial genes (Extended Data Fig. 1a−c). The primary ISR effector *ATF4* was induced before the UPR[mt], while *CHOP* (a direct target of ATF4) induction showed a similar profile to UPR[mt] genes (Extended Data Fig. 1a,d). However, knockout (KO) of both main ISR effectors did not reduce the UPR[mt], suggesting that the ISR−ATF4 axis was not required for UPR[mt] induction (Extended Data Fig. 1e−h).

## Mitochondrial reactive oxygen species are required for UPR[mt] activation

To identify the molecular signatures that signal the UPR[mt], we carried out time-resolved transcriptomic analyses of cells within 3 h of GTPP treatment (Fig. 1a). Principal component analysis revealed that cells treated with GTPP showed distinct transcriptomic patterns over time (Extended Data Fig. 2a), indicating a dynamic transcriptional response during UPR[mt] activation, with 489 and 383 transcripts gradually increased and decreased, respectively (Fig. 1b, Extended Data Fig. 2b−d and Supplementary Table 1). Genes prominently enriched during UPR[mt] activation included 'response to oxidative stress' (Fig. 1b and Extended Data Fig. 2c,e), suggesting that reactive oxygen species (ROS) may contribute to UPR[mt] signalling. In line with this hypothesis, induction of MMS caused increased mitochondrial reactive oxygen species (mtROS; $O_2^{\cdot-}$) levels (Fig. 1c and Extended Data Fig. 2f). While high levels of ROS can be detrimental, mitochondria often use ROS to communicate with different organelles[12]. To test whether ROS are necessary for UPR[mt] activation, we carried out cotreatments with the antioxidants *N*-acetylcysteine (NAC) and reduced glutathione (GSH) and the superoxide dismutase mimetic MnTBAP. Strikingly, all three antioxidants inhibited UPR[mt] induction (Fig. 1d and Extended Data Fig. 2g−l) without affecting mitochondrial protein aggregate formation (Fig. 1e). The opposite experimental paradigm, cotreatment with the complex III inhibitor antimycin A to increase mtROS, enhanced UPR[mt]

[1]Institute of Biochemistry II, Faculty of Medicine, Goethe University Frankfurt, Frankfurt am Main, Germany. [2]These authors contributed equally: F. X. Reymond Sutandy, Ines Gößner. ✉e-mail: ch.muench@em.uni-frankfurt.de

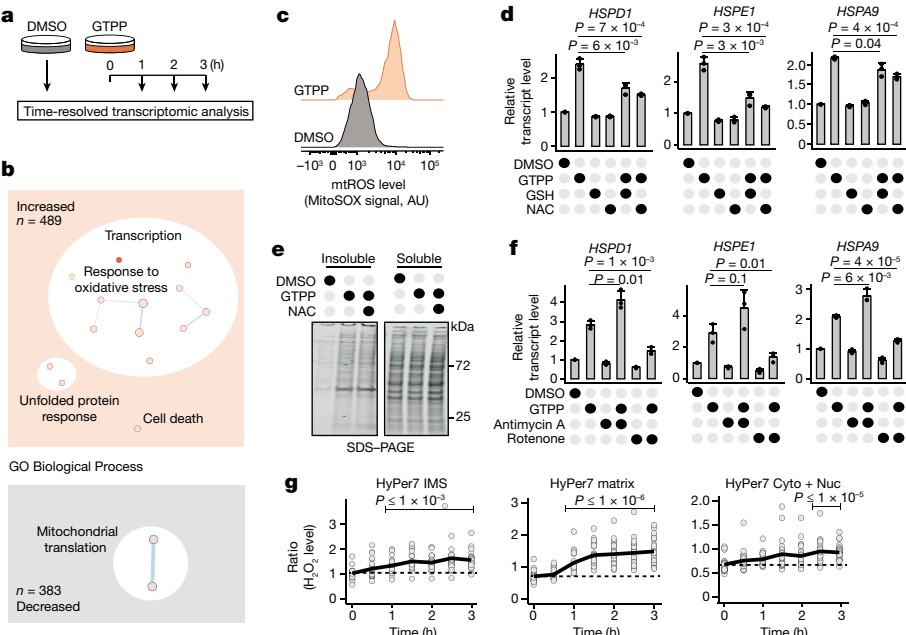

**Fig. 1 | ROS are produced and required to activate the UPR^mt upon MMS.**
**a**, Scheme of the experimental design for time-resolved transcriptomics (RNA sequencing) upon GTPP treatment. Dimethyl sulfoxide (DMSO) was used as negative control. **b**, Enrichment maps of GO Biological Process (BP) terms from the transcriptomic analysis. Circles represent GO terms with a false discovery rate of less than 0.1. GO BP 'transcriptional response to oxidative stress' (GO:0036091) is marked with a dark orange circle. The total numbers of transcripts belonging to increased and decreased groups are represented as *n*. **c**, Representative FACS measurement of mtROS levels with MitoSOX upon GTPP treatment. AU, arbitrary unit. **d**, Bar plots showing the mean of relative transcript levels of UPR^mt genes of GTPP-treated cells upon cotreatments with the antioxidants NAC and GSH measured with qPCR (*n* = 3 biological replicates).

**e**, Gel image of mitochondrial insoluble and soluble fractions upon different treatments. Insoluble fractions are a measure for aggregate formation. **f**, Bar plots showing the mean of relative transcript levels of UPR^mt genes of GTPP-treated cells upon cotreatments with mtROS inducers antimycin A and rotenone measured with qPCR (*n* = 3 biological replicates). **g**, Microscopy-based measurement of $H_2O_2$ levels with HyPer7 reporters targeted to the IMS and the matrix and untargeted (cytosol + nucleus (Cyto + Nuc)) upon GTPP treatment. Each point represents an individual cell measurement. Lines represent the mean $H_2O_2$ levels across different time points (*n* = 18 cells) during 3 h of measurement from five biological replicates. All *P* values are calculated with a two-tailed unpaired Student's *t* test and indicated. All error bars represent mean ± s.d.

activation (Fig. 1f and Extended Data Fig. 2m). These findings show that mitochondria employ mtROS as an essential signal to activate and scale the UPR^mt. However, increasing mtROS levels alone was not sufficient to activate the UPR^mt (Fig. 1f and Extended Data Fig. 2m), indicating that additional factors are required for UPR^mt activation.

Next, we investigated the source of mtROS to understand how mtROS mediate UPR^mt activation. Intriguingly, we found that elevating mtROS production by cotreatment with the complex I inhibitor rotenone had the opposite effect to antimycin A (Fig. 1f and Extended Data Fig. 2m), indicating site specificity of mtROS production to signal the UPR^mt. To monitor compartment-specific mtROS production, we used the ultrasensitive $H_2O_2$ probe HyPer7 (ref. 13) targeted to different mitochondrial compartments, as $H_2O_2$ is the most common type of ROS used in intracellular signalling[14] (Extended Data Fig. 3a–f). $H_2O_2$ levels increased significantly in the intermembrane space (IMS) and the matrix within 1 h of GTPP treatment (Fig. 1g). At later time points, $H_2O_2$ levels also significantly increased in the cytosol (Fig. 1g and Extended Data Fig. 3c–f). These findings support a model in which mtROS diffuse into the cytosol and signal the UPR^mt. Consistently, blocking ROS transport between mitochondria and the cytosol with 4,4′diisothiocyanatostilbene-2,2′-disulfonate (DIDS), an inhibitor of the outer membrane pore VDAC1, abolished UPR^mt activation (Extended Data Fig. 3g–i). Together, our findings show that mtROS accumulation and diffusion into the cytosol are essential for UPR^mt signalling.

## DNAJA1 oxidation regulates the UPR^mt

We considered that mtROS produced during MMS oxidize a cytosolic target to mediate UPR^mt signalling. To identify proteins oxidized

upon UPR^mt activation, we carried out unbiased, multiplexed redox proteomics to identify cytosolic proteins that are reversibly cysteine oxidized upon GTPP treatment (Fig. 2a and Extended Data Fig. 4a). We monitored oxidation changes within 3 h of MMS induction to identify changes involved in early UPR^mt signalling. Four cysteine residues showed increased oxidation upon GTPP treatment (Fig. 2b, Extended Data Fig. 4b and Supplementary Table 2). Intriguingly, one of the proteins identified to be oxidized during UPR^mt activation was the cytosolic HSP40 (DNAJA1), a cochaperone of cytosolic HSP70 (ref. 15). We found increased DNAJA1 oxidation at cysteines 149 and 150 upon GTPP treatment (Fig. 2b). Oxidation of DNAJA1 and its homologues has been shown to influence its activity by regulating its zinc finger-like regions (ZFLRs)[16,17]. This renders DNAJA1 sensitive to redox changes, presenting a potential target for the redox signalling of the UPR^mt. Indeed, depletion of DNAJA1, but not other HSP40 members, prevented UPR^mt activation (Fig. 2c and Extended Data Fig. 4c–e), showing that DNAJA1 is essential for UPR^mt signalling.

To better understand DNAJA1's role in UPR^mt signalling, we examined its activity during MMS. Quantitative interaction proteomics showed that DNAJA1 exhibited significantly increased binding to a large number of mitochondrial proteins and the cytosolic HSP70s (HSPA1A and HSPA1B) during GTPP treatment (Fig. 2d–f, Extended Data Fig. 5a,b and Supplementary Table 3). Notably, the DNAJA1–HSP70 interaction was ROS dependent (Fig. 2g,h). DNAJA1 cysteines 149 and 150 are part of the ZFLR (Fig. 2i). Their oxidation during the UPR^mt may interfere with zinc ion binding[16]. To assess whether HSP70 recruitment to DNAJA1 during GTPP treatment was mediated by conformational changes of its ZFLR, we introduced C149V and C150V mutations to mimic the effect of oxidation by removal of the cysteines required for the interaction with zinc ions

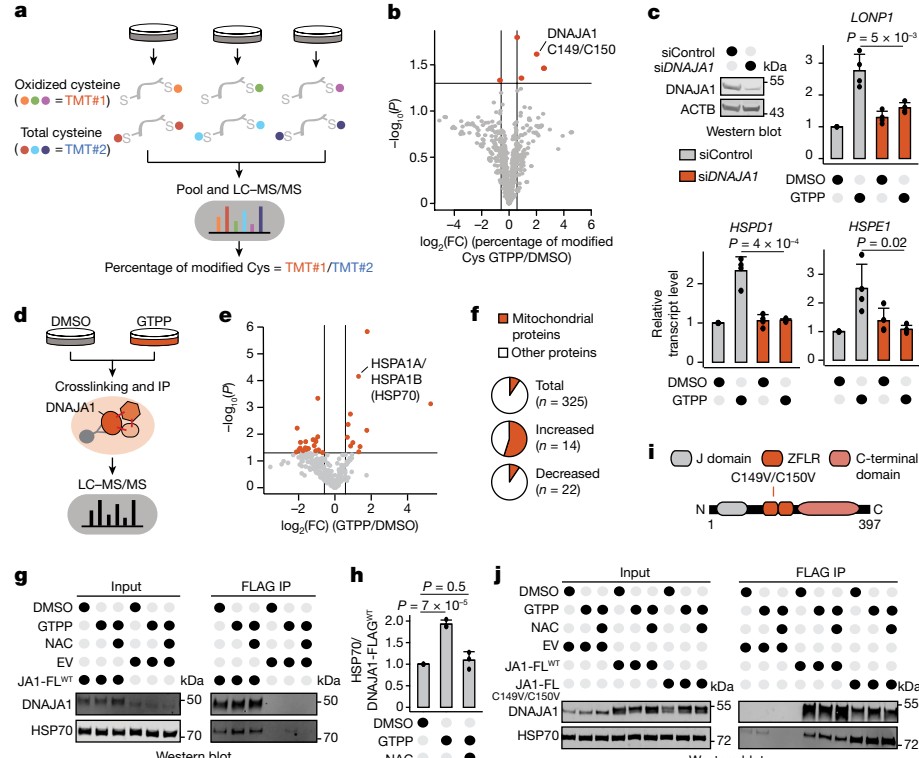

**Fig. 2 | DNAJA1 mediates the signalling to activate the UPR^mt. a**, Scheme of the redox proteomics for cells treated with DMSO or GTPP (in biological triplicates). Oxidized and total cysteine side chains were labelled with different iodoTMTs, pooled into one six-plex sample and measured by LC–MS/MS. **b**, Volcano plot of the redox proteome showing changes in cysteine side-chain oxidation upon GTPP treatment. Dark orange points represent significantly changed cysteine side chains ($P \le 0.05$, fold change (FC) > 1.5). Statistical analysis was performed with a two-tailed unpaired Student's *t* test. **c**, Bar plots showing the mean of relative transcript levels of UPR^mt genes of GTPP-treated cells upon knockdown of *DNAJA1* measured with qPCR ($n = 4$ biological replicates). The knockdown efficiency is shown in the western blot image (left). **d**, Scheme of the experimental steps for DNAJA1 quantitative interaction proteomics. IP, immunoprecipitation. **e**, Volcano plot of the DNAJA1 interaction proteomics upon GTPP treatment. Dark orange points represent significantly changed

interactions ($P \le 0.05$, FC > 1.5). Statistical analysis was performed with a two-tailed unpaired Student's *t* test. **f**, Pie charts representing the proportions of mitochondrial proteins on different groups of DNAJA1 interacting partners. The total numbers of proteins from individual groups are indicated. **g,h**, Representative western blot images of wild-type FLAG-tagged DNAJA1 (JA1-FL^WT)–HSP70 interactions upon different conditions (**g**) and quantification of three biological replicates (**h**). EV represents the empty vector control. **i**, Schematic representation of DNAJA1 domain composition. The exact positions of DNAJA1 mutations used in the experiments are indicated. **j**, Western blot images of the wild-type, C149V and C150V FLAG-tagged DNAJA1 mutant interaction with HSP70 upon different conditions ($n = 2$ biological replicates). All *P* values are calculated with a two-tailed unpaired Student's *t* test and indicated. All error bars represent mean ± s.d. Gel source data are in Supplementary Figs. 1 and 2.

(Fig. 2i). Indeed, mimicking DNAJA1 oxidation increased the DNAJA1–HSP70 interaction, similar to the effect we observed during GTPP treatment (Fig. 2j). These findings suggest that the MMS-induced mtROS lead to oxidation of the DNAJA1 ZFLR to increase HSP70 recruitment.

DNAJA1 preselects and delivers specific client proteins to HSP70 (ref. 15). Thus, the increase in DNAJA1–HSP70 interaction might indicate formation of an active DNAJA1–client complex mediating binding to HSP70. This is consistent with the observed increase in interactions between DNAJA1 and mitochondrial proteins during GTPP treatment (Fig. 2f and Extended Data Fig. 5b). The DNAJA1–HSP70 interaction with mitochondrial proteins occurred in the cytosol, ruling out a potential mislocalization of DNAJA1 to mitochondria during MMS (Extended Data Fig. 5c–e). The presence of mitochondrial proteins in the cytosol was not associated with apoptotic cell death (Extended Data Fig. 5f,g). Together, our results identify DNAJA1 as an integral component of UPR^mt signalling.

## The UPR^mt requires mitochondrial protein precursor accumulation in the cytosol

Next, we evaluated the underlying reasons for the increased interaction of DNAJA1–HSP70 with mitochondrial proteins during UPR^mt activation.

The majority of mitochondrial proteins are synthesized in the cytosol as precursors that need to be imported into mitochondria, where they are processed into their mature form[18]. In yeast, mitochondrial protein precursors can accumulate in the cytosol during stress and cause activation of cytosolic stress responses that aim at restoring proteostasis[19–21]. Whether such mechanisms exist in humans is unclear. We speculated that similar precursors may accumulate in the cytosol and serve as DNAJA1 clients during MMS in humans. Indeed, mitochondrial protein precursors increased during MMS (Fig. 3a). Given that these mitochondrial proteins showed increased interaction with DNAJA1 upon GTPP treatment, we checked whether accumulation of mitochondrial protein precursors in the cytosol (c-mtProt) was required for UPR^mt signalling. Preventing c-mtProt accumulation by inhibiting cytosolic protein translation with cycloheximide (CHX) decreased UPR^mt activation (Fig. 3b). This effect appeared to be selective for mitochondrial proteins since preventing transcription of mitochondrial genes (but not mitochondrial chaperones) via depletion of the mitochondrial biogenesis factor *NRF1* was sufficient to reproduce these effects (Extended Data Fig. 6a–c). These observations show that c-mtProt accumulation is a key component of UPR^mt signalling.

Our previous work had shown that different stressors compromising mitochondrial import cause c-mtProt accumulation[22]. To check

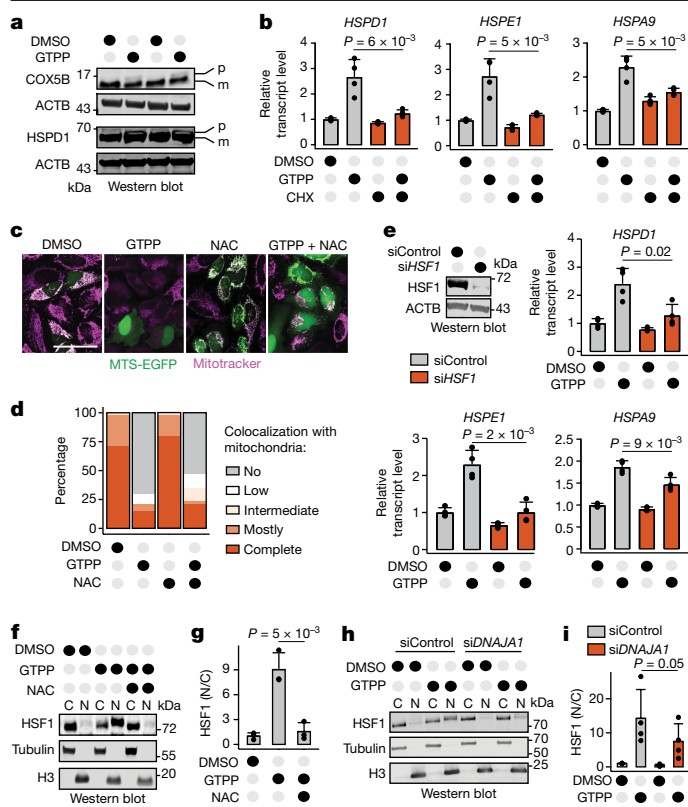

**Fig. 3 | ROS and accumulation of c-mtProt activate the DNAJA1–HSF1 axis to induce the UPRᵐᵗ. a**, Western blot images of mitochondrial proteins in their precursor (p) and mature (m) forms upon GTPP treatment. **b**, Bar plots showing the mean of relative transcript levels of UPRᵐᵗ genes of GTPP-treated cells upon cotreatment with CHX measured with qPCR ($n = 4$ biological replicates). **c**, Representative microscopy images of MTS-EGFP (green) localization in comparison with mitochondria (magenta) upon different treatments. Scale bar, 50 µm. **d**, Bar plot depicting mean quantification of microscopy images ($n = 100$ cells). **e**, Bar plots showing the mean of relative transcript levels of UPRᵐᵗ genes of GTPP-treated cells upon knockdown of *HSF1* measured with qPCR ($n = 4$ biological replicates). The knockdown efficiency is shown by western blotting (upper left). **f**, Representative western blot images of HSF1 in the cytosolic (C) and nuclear (N) fractions of cells under different treatments. **g**, Bar plot depicting the mean of the nuclear-to-cytosolic ratio (N/C) of HSF1 from triplicate images of the western blots. **h**, Representative western blot images of HSF1 in C and N fractions of cells upon knockdown of *DNAJA1*. **i**, Bar plot depicting the mean of the N/C of HSF1 from four replicate images of the western blots. *P* values are calculated with a two-tailed unpaired (**b**,**e**,**g**) or paired (**h**) Student's *t* test and indicated. All error bars represent mean ± s.d. Gel source data are in Supplementary Figs. 1 and 2.

whether inducing MMS leads to mitochondrial import defects, we used an MTS-EGFP reporter[23]. During GTPP treatment, we observed a decreased mitochondrial MTS-EGFP signal and in parallel, an increased signal in the cytosol and nucleus (Fig. 3c,d). These import defects were ROS independent (Fig. 3c,d). Monitoring newly synthesized Halo-tagged mitochondrial proteins resulted in the same observations (Extended Data Fig. 6d–j). Consequently, accumulation of c-mtProt constitutes a second signal of the UPRᵐᵗ, in addition to mtROS.

## mtROS and c-mtProt activate DNAJA1–HSF1

We next addressed which downstream factor might integrate the mtROS and c-mtProt accumulation signals to convey the UPRᵐᵗ to the nucleus. In yeast, c-mtProt accumulation has been shown to remodel transcription by regulating heat shock factor 1 (ref. 24). We checked whether HSF1 integrates mitochondrial signals to activate the UPRᵐᵗ.

Indeed, depletion of *HSF1* abrogated UPRᵐᵗ induction (Fig. 3e and Extended Data Fig. 7a–c). Strikingly, basal mitochondrial chaperone protein levels were also reduced in *HSF1* KO cells (Extended Data Fig. 7a), suggesting that HSF1 serves as a constitutive key regulator of mitochondrial chaperone transcription. This hypothesis is in line with a previous finding, which showed that HSF1 mediates mitochondrial chaperone expression during mitochondrial stress[25]. In addition to the dependency on HSF1 expression, we found activation of HSF1 during GTPP treatment, monitored by its translocation from the cytosol to the nucleus (Fig. 3f,g). Notably, transcription of non-UPRᵐᵗ-related mitochondrial proteins was not controlled by HSF1, indicating a separate regulation of the UPRᵐᵗ and general mitochondrial biogenesis (Extended Data Fig. 7d,e).

We next tested whether HSF1 activation during MMS requires mtROS and c-mtProt accumulation. Inhibition of ROS signalling by antioxidants (Figs. 1d and 3f,g and Extended Data Fig. 7f), reduction of c-mtProt via cotreatment with CHX (Extended Data Fig. 7g,h) or knockdown of *NRF1* (Extended Data Fig. 7i,j) prevented HSF1 translocation and the UPRᵐᵗ. These findings indicate that HSF1 activation might take part in the signalling cascade to activate the UPRᵐᵗ via DNAJA1. To test this hypothesis, we checked whether DNAJA1 was required for the HSF1 activation observed upon MMS. Indeed, depletion of *DNAJA1* significantly inhibited HSF1 translocation to the nucleus (Fig. 3h,i).

Under basal conditions, HSP70 binds to HSF1 and represses its transcriptional activity[26]. Immunoprecipitation experiments of HSF1 showed that the HSF1–HSP70 interaction decreased upon GTPP treatment (Extended Data Fig. 7k,l), suggesting that the recruitment of HSP70 to c-mtProt via DNAJA1 titrates HSP70 away from HSF1. This then leads to HSF1 activation and its subsequent translocation to the nucleus to activate the UPRᵐᵗ. Overall, our findings define HSF1 as the transcription factor downstream of DNAJA1 that responds to mtROS and c-mtProt accumulation to induce the UPRᵐᵗ.

## The DNAJA1–HSF1 axis activates the UPRᵐᵗ

Here, we found a specific pathway that signals the UPRᵐᵗ driven by a cytosolic surveillance mechanism that conveys information on both mtROS and c-mtProt accumulation to the nucleus via DNAJA1 and HSF1. While we did not identify the ISR–ATF4 axis to be essential for UPRᵐᵗ signalling (Extended Data Fig. 1e–h), previous work has shown a potential involvement of the ISR target gene *ATF5* (ref. 6). However, depletion of *ATF5* did not inhibit UPRᵐᵗ activation (Extended Data Fig. 8a–e). Intriguingly, *ATF5* transcription increased after UPRᵐᵗ activation (Extended Data Fig. 8f) and depended on both DNAJA1 and HSF1 (Fig. 4a,b). Thus, we suggested that ATF5 may largely act downstream after UPRᵐᵗ activation by the cytosolic surveillance. Indeed, analyses of available HSF1 chromatin immunoprecipitation and sequencing (CHIP–seq) data showed that HSF1 binds the ATF5 regulatory region (Extended Data Fig. 8g), and both HSF1 and ATF5 are able to bind directly to the promoters of mitochondrial chaperones (Extended Data Fig. 8h). Thus, while ATF5 was not required for UPRᵐᵗ activation, it might fulfil essential functions later.

We next assessed whether the identified UPRᵐᵗ cytosolic surveillance is the general mechanism for UPRᵐᵗ activation upon MMS. Activation of the UPRᵐᵗ upon inhibition of two different mitochondrial proteases—LON protease (LONP) with 2-cyano-3,12-dioxoolean-1,9-dien-28-oic acid (CDDO) and HTRA2 with dihydro-5-[[5-(2-nitrophenyl)-2-furanyl]methylene]-1,3-diphenyl-2-thioxo-4,6(1H,5H)-pyrimidinedione (Ucf-101)—was also dependent on ROS and accumulation of c-mtProt (Fig. 4c–f), and it was mediated by DNAJA1 and HSF1 (Fig. 4g–j). Thus, signalling across the ROS + c-mtProt–DNAJA1–HSF1 axis is a common pathway used for UPRᵐᵗ activation in human cells. Next, we induced the UPRᵐᵗ genetically by overexpression of the aggregation-prone protein Abeta in mitochondria or double knockdown of the mitochondrial proteases *LONP1* and *PITRM1* and found that it was also dependent

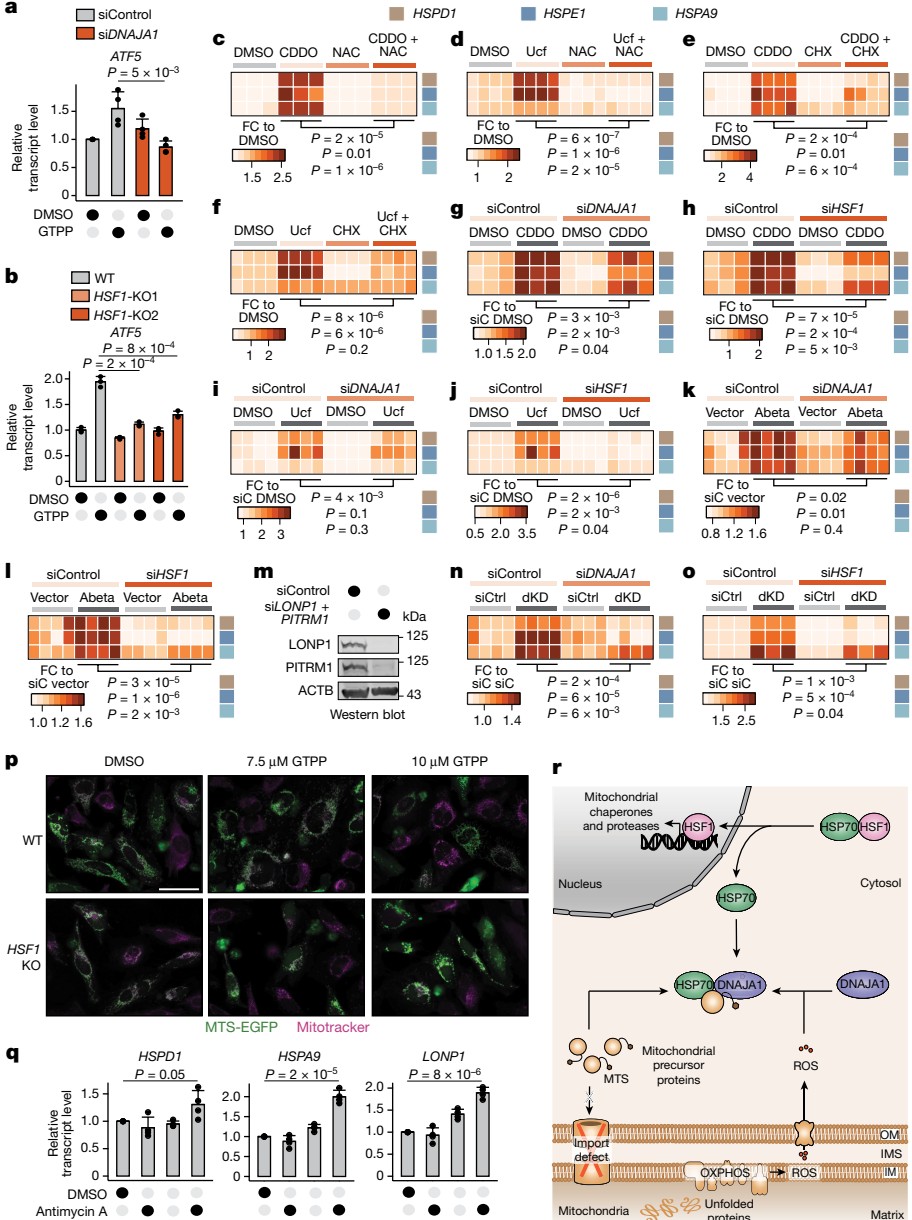

**Fig. 4 | Different stressors activate the UPR^mt via the cytosolic surveillance mechanism. a,b,** Bar plots showing the mean of relative transcript levels of *ATF5* upon *DNAJA1* knockdown (*n* = 4 biological replicates) (**a**) and *HSF1* KO (*n* = 3 biological replicates) (**b**) measured by qPCR. **c,d,** Heat maps of relative transcript levels of UPR^mt genes in CDDO- (**c**) or Ucf-101-treated (**d**) cells upon cotreatments with NAC (*n* = 3 and 4 biological replicates for CDDO and Ucf-101, respectively). **e,f,** Heat maps of relative transcript levels of UPR^mt genes in CDDO- (**e**) or Ucf-101-treated (**f**) cells upon cotreatments with CHX (*n* = 4 biological replicates). **g–j,** Heat maps of relative transcript levels of UPR^mt genes in *DNAJA1* knockdown cells treated with CDDO (**g**) or Ucf-101 (**i**), or *HSF1* knockdown cells treated with CDDO (**h**) or Ucf-101 (**j**) (*n* = 3 and 4 biological replicates for CDDO and Ucf-101, respectively) (**g–j**) measured by qPCR. siC, siControl. **k,l,** Heat maps of relative transcript levels of UPR^mt genes upon knockdown of *DNAJA1* (**k**) and *HSF1* (**l**) in cells overexpressing MTS-Abeta (*n* = 4 biological

replicates) measured by qPCR. **m,** Western blot image of double knockdown of *LONP1* and *PITRM1* (*n* = 2 biological replicates). **n,o,** Heat maps of relative transcript levels of UPR^mt genes upon knockdown of *DNAJA1* (**n**) and *HSF1* (**o**) in cells within an *LONP1* and *PITRM1* double knockdown (dKD) background (*n* = 4 biological replicates) measured by qPCR. Transcript levels are represented as relative FCs. **p,** Representative microscopy images of MTS-EGFP (green) localization in comparison with mitochondria (magenta) in wild-type (WT) and *HSF1* KO cells (*n* = 5 biological replicates). Scale bar, 50 μm. **q,** Bar plots showing the mean of relative transcript levels of UPR^mt genes measured with qPCR (*n* = 4 biological replicates). **r,** Working model of the cytosolic UPR^mt surveillance mechanism and transcriptional UPR^mt activation. OM, outer membrane; IM, inner membrane. All *P* values are calculated with a two-tailed unpaired Student's *t* test and indicated. All error bars represent mean ± s.d. Gel source data are in Supplementary Fig. 1.

on DNAJA1 and HSF1 (Fig. 4k–o and Extended Data Fig. 9a,b). These observations underline the relevance of the cytosolic surveillance mechanism for the maintenance of mitochondrial proteostasis in a physiological context. Indeed, knocking out *HSF1* increased mitochondrial vulnerability upon MMS, leading to import defects and reduced overall cell survivability (Fig. 4p and Extended Data Fig. 9c–f).

Finally, we evaluated whether ROS + c-mtProt can directly activate the UPR^mt signalling cascade without upstream induction of MMS. Employing oxidative phosphorylation (OXPHOS) inhibitors antimycin A or oligomycin A individually was not sufficient to simultaneously induce mtROS and c-mtProt and to activate the UPR^mt (refs. 27,28) (Extended Data Fig. 10a–h). However, the combination of antimycin

A and oligomycin A increased ROS and c-mtProt, and it induced the UPR$^{mt}$ without mitochondrial protein aggregate formation (Fig. 4q and Extended Data Fig. 10a–i). In addition, inspecting different mitochondrial stressors confirmed that individual induction of ROS or c-mtProt accumulation alone did not induce the UPR$^{mt}$ (Extended Data Fig. 10j–l). These findings support that mitochondrial stress can activate UPR$^{mt}$ signalling only when both ROS and c-mtProt accumulation are induced.

## Discussion

Our data uncover the signalling molecules used by mitochondria to initiate the UPR$^{mt}$, a two-pronged signalling cascade composed of mtROS and c-mtProt (Fig. 4r). mtROS and c-mtProt signals converge at the DNAJA1-mediated activation of HSF1, forming a surveillance mechanism in the cytosol to initiate the transcriptional programme of mitochondrial chaperones and proteases. These findings reveal an unexpected connection between the mitochondrial and cytosolic proteostasis networks during UPR$^{mt}$ activation. Intriguingly, while a canonical UPR$^{mt}$ has not been described in yeast, c-mtProt has been shown to cause remodelling of cytosolic proteostasis[19–21,24,29]. In *Caeno-rhabditis elegans*, perturbation of mitochondrial protein homoeostasis had also been shown to activate cytosolic proteostasis coordinated by lipid biosynthesis[30]. Human cells appear to have evolved these principles to a complex cytosolic surveillance system for UPR$^{mt}$ activation, which links mitochondrial proteostasis to a broader network of cellular homoeostasis. Ultimately, this pathway might provide explanations for diseases in which the breakdown of cytosolic proteostasis is linked to mitochondrial dysfunction, including ageing.

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

# Methods

## Data reporting

No statistical methods were used to predetermine sample size. The experiments were not randomized and investigators were not blinded to allocation during experiments and outcome assessment.

## Cell culture and treatments

HeLa ovarian carcinoma cells from the American Type Culture Collection were used for all experiments unless stated otherwise. They were confirmed to be mycoplasma negative and grown in RPMI medium (Thermo Fisher Scientific) and 10% fetal bovine serum. Knockdown experiments were performed with Lipofectamine RNAiMAX according to the manufacturer's instructions. The small interfering RNAs (siRNAs) used were from OriGene oligo duplex *ATF5* (SR307793) and custom made for *LONP1* (sense 5′-GGACGUCCUGGAAGAGACCAAU AUU-3′, anti-sense 5′-AAUAUUGGUCUCUUCCAGGACGUCC). MISSION esiRNA (Sigma) were *ATF5* (EHU039491), *DNAJA1* (EHU114481), *HSF1* (EHU107721), *DNAJA2* (EHU005311), *DNAJB1* (EHU109151), *NRF1* (EHU069871) and *PITRM1* (EHU011041). Gene KOs were conducted by CRISPR–Cas9-mediated genome editing. The single guide RNAs (sgRNAs) were cloned into eSpCas9 (1.1; Addgene, catalogue no. 71814). The sgRNA sequences used were 5′-GCAACAGAAAGTCGTCAACA-3′ (*HSF1*), 5′-TCTCTTAGATGATTACCTGG-3′ (*ATF4*), TCAGCCAAGCCAG AGAAGCA-3′, 5′-ATTTCCAGGAGGTGAAACAT-3′ (*DDIT3*), 5′-TGGCTCCC TATGAGGTCCTT-3′ (*ATF5_1*) and 5′-AGACTATGGGAAACTCCCCC-3′ (*ATF5_2*). Together with sgRNA-containing plasmid, cells were cotransfected with puromycin-resistant plasmids and selected for 24 h with 1 μg ml⁻¹ puromycin (Invivogen). After the selection, single cells were seeded into 96-well plates and incubated for 2 weeks. Resulting colonies were expanded, and gene KO was confirmed by Sanger sequencing and western blot.

Transient overexpression of MTS-Abeta-GFP was carried out with Lipofectamine 2000 according to the manufacturer's instruction. Cells were harvested after 24 h.

Acute induction of the UPR$^{mt}$ was performed with 10 μM GTPP (Shanghai Chempartner), 5 μM CDDO (Cayman Chemical) and 40 μM Ucf-101 (Cayman Chemical) for 6 h unless stated otherwise (for early response, a 3 h incubation was used). For the cell viability assay, a toxic concentration of 15 μM GTPP for 16 h was applied (Extended Data Fig. 9d–f). mtROS induction was done by treating cells with 10 μM antimycin A (Sigma) or 2 μM rotenone (Sigma) for 6 h. To scavenge ROS, cells were pretreated with 10 mM NAC (Sigma) or 10 mM GSH (Cayman Chemical) for 1 h or 100 μM MnTBAP (Sigma) overnight that was continued as a cotreatment. For Hyper7 references, 20 μM antimycin A and 1 mM H$_2$O$_2$ (Carl Roth) were used; 4,4′diisothiocyanatostilbene-2,2′-disulfonate (75 μM, Sigma) was used to inhibit VDAC1 for 6 h. General translation was blocked by treatment with 35 μM CHX for 30 min and continued as cotreatment for 6 h. Mitochondrial import inhibition was performed with 5 μM oligomycin A (Sigma) for 6 h. Different mitochondrial stressors were applied by 6 h of treatment with 10 μM carbonyl cyanide m-chlorophenyl hydrazone (CCCP, Abcam), 100 μM deferiprone (DFP, Sigma) or 10 μM Menadione (Sigma). The hypoxic condition was generated by incubating cells in the BD GasPak EZ Pouch system (BD Diagnostics) for 6 h. Staurosporine (Cayman Chemical) was used to induce apoptosis as a control treatment at 1 μM for 3 h or 200 nM overnight for the *HSF1* KO cell viability assay (Extended Data Fig. 9d–f).

## Cloning

For the generation of the construct MTS-Abeta-GFP, pcDNA5/FRT/TO (Thermo) was used as a backbone. The following inserts were amplified by Q5 High-Fidelity DNA Polymerase (NEB) and cloned into the backbone via NEBuilder HiFi DNA Assembly Master Mix (NEB): MTS (2× COX8 presequence in tandem) amplified from pCMV CEPIA2mt (Addgene, catalogue no. 58218), Abeta (Aβ1-42) amplified from HeLa wild-type complementary DNA (cDNA) with primers (5′-TCC ATG CGG GGT TCT GAT GCA GAA TTC CGA CAT GAC TCA GGA TAT G-3′ and 5′-CTC GCC CTT GCT CAC GGA TCC CGC TAT GAC AAC ACC GCC CAC C-3′, containing a GS linker) and enhanced green fluorescent protein (EGFP) amplified from Su9-EGFP (Addgene, catalogue no. 23214).

## RNA sequencing

Total RNAs were extracted from cells using the NucleoSpin RNA Plus kit (Macherey-Nagel) following the manufacturer's instructions and subsequently digested with Turbo DNase (Thermo Fisher Scientific). Library preparation for bulk sequencing of poly(A)-RNA was done as described previously[31]. Briefly, barcoded cDNA of each sample was generated with a Maxima RT polymerase (Thermo Fisher Scientific) using an oligo-dT primer containing barcodes, unique molecular identifiers (UMIs) and an adaptor. Ends of the cDNAs were extended by a template switch oligo, and full-length cDNA was amplified with primers binding to the template switch oligo site and the adaptor. The NEB UltraII FS kit was used to fragment cDNA. After end repair and A tailing, a TruSeq adaptor was ligated, and 3′-end fragments were finally amplified using primers with Illumina P5 and P7 overhangs. In comparison with Parekh et al.[31], the P5 and P7 sites were exchanged to allow sequencing of the cDNA in read1 and barcodes and UMIs in read2 to achieve a better cluster recognition. The library was sequenced on a NextSeq 500 (Illumina) with 63 cycles for the cDNA in read1 and 16 cycles for the barcodes and UMIs in read2.

## RNA sequencing analysis

Gencode gene annotations v.35 and the human reference genome GRCh38 were derived from the Gencode homepage (European Molecular Biology's European Bioinformatics Institute (EMBL-EBI)). Drop-Seq tools (v.1.12)[32] were used for mapping raw sequencing data to the reference genome. The resulting UMI filtered count matrix was imported into R (v.4.0.5), and lowly expressed genes were subsequently filtered out. Data were then variance stabilized via the rlog function as implemented in DESeq2 (v.1.18.1)[33]. For accurate dispersion estimation, the experimental design (treatment at a given time point) was provided to the function. rlog normalized data were used to perform clustering analysis (fuzzy C means) with R package mFuzz (v.2.50.0)[34]. Transcripts with rlog normalized values of less than three were excluded from the analysis. The number of clusters was set to three. Transcripts were assigned to increased and decreased cluster groups based on cluster membership greater than or equal to 0.8 for each cluster. Gene Ontology (GO) enrichment analysis was performed on each cluster group by using Database for Annotation, Visualization and Integrated Discovery (DAVID). GO enrichments were visualized with the EnrichmentMap (v.3.3.2) plug-in in Cytoscape (v.3.7.1).

## Quantitative polymerase chain reaction analysis

Total RNAs were extracted from cells using the NucleoSpin RNA Plus kit (Macherey-Nagel) following the manufacturer's instructions. cDNA synthesis was performed with the High-capacity cDNA reverse transcription kit (Applied Biosystems). Quantitative polymerase chain reaction (qPCR) analysis was performed with primaQuant SYBRGreen master mix without ROX (Steinbrenner Laborsysteme) according to the manufacturer's instructions. KiCqStart primers SYBR green from Sigma (Supplementary Table 4) were used to perform qPCR measurement with LightCycler 480 SW (v.1.5) on the LightCycler 480 real-time PCR system (Roche) in 384-well format. *ACTB* was used as an internal control. Fold changes of the transcript level were calculated using the comparative CtΔΔCt (cycle threshold) method.

## FACS measurement

MitoSOX Red (Thermo Fisher Scientific) was used to measure mtROS production according to the manufacturer's instructions. Cell deaths

were measured with a combination of Annexin V conjugated to Alexa Fluor 488 (Thermo Fisher Scientific) and propidium iodide (Thermo Fisher Scientific) according to the manufacturer's instructions. Fluorescence-activated cell sorting (FACS) was performed with FACSDiva (v.6.1.3) on FACSCanto II and FACSymphony A5 flow cytometry systems (BD) for MitoSOX and Annexin V measurements, respectively. Analysis of FACS data was performed with FlowJo v.10 software.

## Immunoblotting

Cells were lysed in RIPA buffer containing Complete Mini EDTA-free protease inhibitor (Roche) and GENIUS nuclease (Santa Cruz Biotechnology). Lysates were prepared in 1× Laemmli buffer and boiled for 10 min at 95 °C. Proteins were separated with SDS–PAGE using the Invitrogen Novex system and transferred to nitrocellulose membrane by using Mini Trans-Blot cell (Bio-Rad). Primary antibodies were added to immunoblots for 1 h at room temperature (RT). Antibodies used for the detection were anti-ACTB (SantaCruz, catalogue no. sc69879, 1:4,000), anti-HSPD1 (Abcam, catalogue no. ab46798, 1:2,000), anti-COX5B (Proteintech, catalogue no. 11418-2-AP, 1:1,000), anti-HSF1 (Cell Signaling, catalogue no. 4356, 1:1,000), anti-HSF1 (Abcam, catalogue no. ab2923, 1:10,000), anti-DNAJA1 (Proteintech, catalogue no. 11713-1-AP, 1:2,000), anti-Hsp70 (Proteintech, catalogue no. 10995-1-AP, 1:2,000), anti-NRF1 (Cell Signaling (D9K6P), catalogue no. 46743, 1:1,000), anti-α-tubulin (Cell Signaling (DM1A), catalogue no. 3873, 1:3,000), anti-histone H3 (Active Motif, catalogue no. 39163, 1:5,000), anti-FLAG (Sigma, catalogue no. F1804, 1:5,000), anti-CHOP (Thermo Fisher Scientific, catalogue no. MA1-250, 1:1,000), anti-ATF4 (Cell Signaling, catalogue no. 11815, 1:1,000), anti-PITRM1 (Novus, catalogue no. H00010531-M03, 1:500), anti-LONP1 (Proteintech, catalogue no. 15440-1-AP, 1:2,000), anti-cleaved PARP1 (Cell Signaling, catalogue no. 5625, 1:2,000) and anti-Caspase3 (Cell Signaling, catalogue no. 9661, 1:1,000). Secondary antibodies used were anti-rabbit IgG (H + L) HRP Conjugate (Promega, catalogue no. W4021, 1:10,000), IRDye 800CW goat anti-rabbit IgG (H + L; Li-Cor, catalogue no. 926–32211, 1:15,000) and IRDye 680RD donkey anti-mouse IgG (H + L; Li-Cor, catalogue no. 926–68072, 1:15,000). Appropriate secondary antibodies were used for imaging with Odyssey DLx (LI-COR) or ChemiDoc MP (Bio-Rad) imaging system. Data were collected with Image Studio (v.5.2) or ImageLab v.6.0.1.

## Mitochondrial insoluble fraction analysis

Mitochondrial fractions were prepared as previously described[35]. Briefly, cells were homogenized by passing them through a 27-gauge needle syringe in buffer containing 10 mM HEPES (pH 7.4), 50 mM sucrose, 0.4 M mannitol, 10 mM KCl and 1 mM EGTA. Mitochondrial enrichment was performed with a two-step differential centrifugation at 1,000$g$ followed by 13,000$g$ for 15 min each at 4 °C. The mitochondria-enriched pellets were resuspended in a buffer containing 20 mM HEPES (pH 7.4), 0.4 M mannitol, 10 mM NaH$_2$PO$_4$ and 0.5 M EGTA. An equal volume of lysis buffer containing 2% (vol/vol) NP40 was added and spun down to separate mitochondrial fractions. The resulting supernatants and pellets were kept as the soluble and insoluble fractions, respectively. Proteins were resolved with SDS–PAGE in 1× Laemmli buffer and visualized with InstantBlue Coomassie stain (Expedeon).

## Nuclear and cytosolic fractionation

Cells were fractionated with the REAP method[36]. Cell fractions were prepared by resuspending cells in PBS containing 0.1% (vol/vol) NP40, followed by five times resuspension with a p1000 micropipette (Gilson). Cells were fractionated with a 'pop spin' for 10 s at 4 °C in an Eppendorf tabletop microfuge. Supernatants were collected as the cytosolic fractions. Pellets were washed once with 0.1% (vol/vol) NP40 and collected as the nuclear fractions. Both the cytoplasmic and nuclear fractions were used to perform immunoblotting. The ratio of nuclear to cytosolic HSF1 was calculated as follows:

HSF1 (N/C) = (Nuclear HSF1/Histone H3)/(Cytoplasmic HSF1/Tubulin).

## Immunoprecipitation

Crosslinking was performed by incubating cells in PBS containing 0.8 mg ml$^{-1}$ dithiobis[succinimidyl propionate] (Proteochem) for 30 min at RT[37]. Crosslinking reactions were quenched with PBS containing 200 µM glycine for 15 min at RT. Cells were lysed in cell lysis buffer (50 mM Tris (pH 8.0), 150 mM NaCl, 1% (vol/vol) NP40) containing protease inhibitor and allowed to incubate for 30 min at 4 °C. Lysates containing 2 mg of total proteins were used to perform immunoprecipitation with 10 µl Dynabeads protein A (Thermo Fischer Scientific) containing 1 µg of appropriate antibodies or 10 µl Anti-FLAG M2 magnetic beads (Sigma) for 2 h at 4 °C. Immunoprecipitated proteins were eluted from beads for immunoblotting or digested for interaction proteomics.

## Sample preparation for LC–MS/MS

For redox proteomics, cells were lysed in HES buffer (1 mM EDTA, 0.1% (wt/vol) SDS, 50 mM HEPES (pH 8.0)) supplemented with protease inhibitor and 10% (vol/vol) TCA and incubated for 2 h at 4 °C. Each sample was divided into two fractions: (1) oxidized Cys fraction and (2) total Cys fraction. Proteins were precipitated with a TCA and acetone precipitation. For fraction 2, 100 µg of proteins were resuspended in HES buffer supplemented with 5 mM TCEP and incubated for 1 h at 50 °C to reduce all Cys thiols. For fraction 1, 100 µg of the proteins were resuspended in denaturing buffer (6 M urea, 1% (wt/vol) octyl ß-glucopyranoside, 50 mM HEPES (pH 8.0)) supplemented with protease inhibitor and 200 mM iodoacetamide and incubated for 1 h at 37 °C in the dark to block free Cys thiols. Oxidized Cys thiols were reduced as described previously for fraction 2. Proteins were cleaned up by TCA and acetone precipitation. To label the free Cys thiols, proteins were resuspended in denaturing buffer supplemented with iodoTMT#1 (Thermo Fisher Scientific) for fraction 1 or iodoTMT#2 for fraction 2 and incubated for 1 h at 37 °C in the dark. Labelling reactions were quenched with 20 mM DTT. Labelled proteins were pooled together and cleaned up with TCA and acetone precipitation. Proteins were digested with 1:50 (wt/wt) LysC (Wako Chemicals) and 1:100 (wt/wt) Trypsin (Promega) in 10 mM EPPS (pH 8.2) containing 1 M urea overnight at 37 °C. Peptides were purified with (50-mg) SepPak columns (Waters) and then dried. IodoTMT-labelled peptides were enriched with anti-TMT antibody resin (Thermo Fisher Scientific) according to the manufacturer's instructions. Enriched pools of labelled peptides were subjected to high-pH reverse-phase fractionation with the High pH RP Fractionation kit (Thermo Fisher Scientific) following the manufacturer's instructions. Fractionated peptides were concatenated into four separate fractions.

To perform interaction proteomics, after immunoprecipitation steps 25 µl of SDC (2% SDC (wt/vol), 1 mM TCEP, 4 mM chloroacetamide, 50 mM Tris (pH 8.5)) buffer was added to the beads. The mixtures were heated up to 95 °C, and the supernatants were collected. For digestion, 25 µl of 50 mM Tris (pH 8.5) containing 1:50 (wt/wt) LysC (Wako Chemicals) and 1:100 (wt/wt) trypsin (Promega) was added and allowed to incubate overnight at 37 °C. Digestion was stopped by adding 150 µl of isopropanol containing 1% (vol/vol) TFA. Peptide purification was performed with the SDB-RPS disc (Sigma) and then dried.

## LC–MS/MS

Peptides were resuspended in a 2% (vol/vol) acetonitrile/1% (vol/vol) formic acid solution and separated on an Easy nLC 1200 (Thermo Fisher Scientific) and a 35-cm-long, 75-µm-inner-diameter fused-silica column, which had been packed in house with 1.9-µm C18 particles (ReproSil-Pur, Dr. Maisch) and kept at 50 °C using an integrated column oven (Sonation). For redox proteome, peptides were eluted by a nonlinear gradient from 4 to 36% (vol/vol) acetonitrile over 90 min and directly sprayed into a QExactive HF mass spectrometer equipped with a nanoFlex ion source (Thermo Fisher Scientific) at a spray voltage of 2.3 kV. Full-scan MS spectra (350–1,400 $m/z$) were acquired at a resolution of 120,000 at $m/z$ 200, a maximum injection time of 25 ms

and an automatic gain control (AGC) target value of $3 \times 10^6$. Up to 20 of the most intense peptides per full scan were isolated using a 1-Th window and fragmented using higher-energy collisional dissociation (normalized collision energy of 35). MS/MS spectra were acquired with a resolution of 45,000 at $m/z$ 200, a maximum injection time of 86 ms and an AGC target value of $1 \times 10^5$. Ions with charge states of one, five to eight and more than eight as well as ions with unassigned charge states were not considered for fragmentation. Dynamic exclusion was set to 20 s to minimize repeated sequencing of already acquired precursors.

For interaction proteomics, peptides were eluted by a nonlinear gradient from 3.2 to 32% acetonitrile over 60 min followed by a stepwise increase to 95% B in 6 min, which was kept for another 9 min and sprayed into an Orbitrap Fusion Lumos Tribrid Mass Spectrometer (Thermo Fisher Scientific) at a spray voltage of 2.3 kV. Full-scan MS spectra (350–1,500 $m/z$) were acquired at a resolution of 60,000 at $m/z$ 200, a maximum injection time of 50 ms and an AGC target value of $4 \times 10^5$. The most intense precursors with a charge state between two and six per full scan were selected for fragmentation ('Top Speed' with a cycle time of 1.5 s) and fragmented using higher-energy collisional dissociation (normalized collision energy of 30). MS/MS spectra were acquired with a resolution of 15,000 at $m/z$ 200, a maximum injection time of 22 ms and an AGC target value of $1 \times 10^5$. Ions with charge states of one and more than six as well as ions with unassigned charge states were not considered for fragmentation. Dynamic exclusion was set to 45 s to minimize repeated sequencing of already acquired precursors.

## LC–MS/MS data analysis

For analysis of redox proteomics data, raw files were analysed using Proteome Discoverer 2.4 software (Thermo Fisher Scientific). Spectra were selected using default settings and database searches performed using the SequestHT node in Proteome Discoverer. Database searches were performed against a trypsin-digested *Homo sapiens* SwissProt database and FASTA files of common contaminants ('contaminants. fasta' provided with MaxQuant) for quality control. Dynamic modifications were set as methionine oxidation (C, +15.995 Da), iodoTMT6plex (C, +329.227 Da) and carbamidomethyl (C, +57.021 Da) at cysteine residues. One search node was set up to search with Met loss + acetyl (M, −89.030 Da) as dynamic modifications at the N terminus. Searches were performed using Sequest HT. After each search, posterior error probabilities were calculated, and peptide spectrum matches were filtered using Percolator with default settings. Consensus workflow for reporter ion quantification was performed with default settings, except that the minimal signal-to-noise ratio was set to 10. Results were then exported to Excel files for further processing. Non-normalized abundances were used for quantification. The percentage of cysteine oxidation for each peptide was calculated as follows:

Percentage of oxidized Cys = (abundance of fraction 1/abundance of fraction 2) × 100%.

For peptides with several different Cys modifications, fold changes of the percentage of oxidized Cys from each different combination were considered.

For DNAJA1 interaction proteomics, MS raw data processing was performed with MaxQuant (v.1.6.17.0) and its in-build label-free quantification algorithm MaxLFQ applying default parameters[38]. Acquired spectra were searched against the human reference proteome (Taxonomy identification 9606) downloaded from UniProt (12-03-2020; 'One sequence per gene', 20,531 sequences) and a collection of common contaminants (244 entries) using the Andromeda search engine integrated in MaxQuant[39]. Identifications were filtered to obtain false discovery rates below 1% for both peptide spectrum matches (minimum length of seven amino acids) and proteins using a target-decoy strategy[40]. Results were then exported to Excel files for further processing. Abundance of interactors was normalized to the abundance of DNAJA1 from each sample. Fold changes were calculated from normalized data. GO enrichment analysis of DNAJA1 interactome was performed by using DAVID. GO enrichments were visualized with the EnrichmentMap (v.3.3.2) plug-in in Cytoscape (v.3.7.1). Subcellular locations of increased interactors upon GTPP treatment were manually curated from UniProt.

## Microscopy analysis

For HyPer7 measurements, cells were transfected with different constructs of HyPer7 (ref. 13) (Addgene, catalogue nos. 136466, 136469 and 136470) with Lipofectamine 2000 (Thermo Fisher Scientific) according to the manufacturer's instructions. Measurements were performed 24 h after transfection in a 96-well plate format at 37 °C. Time-series live cell imaging was done with a CQ1 confocal imaging cytometer (Yokogawa). HyPer7 was excited sequentially with 405- and 488-nm laser beams. Emission was collected using a 525/50-bandpass emission filter. After five images were acquired, 10 μM GTPP was added to each group of cells expressing different constructs. Image analysis was performed using ImageJ (v.1.53). Fluorescence was calculated for regions of interests inside the imaged cell. The ratiometric signal of HyPer7 was calculated by dividing the intensity of the emission signals excited by 488/405 nm.

Monitoring of DNAJA1 localization was performed using SP8 Confocal (Leica). Cells were incubated in media containing 150 nM MitoTracker deep red FM (Thermo Fisher Scientific) for 30 min at 37 °C in the dark. After subsequent washes, the cells were fixed with 4% (vol/vol) formaldehyde in PBS and permeabilized with 0.01% (vol/vol) TritonX100. Cells were blocked with a PBS buffer containing 1% (wt/vol) bovine serum albumin (BSA), 300 mM glycine and 0.1% (vol/vol) Tween20 for 30 min at RT. Cells were incubated overnight at 4 °C with 1:100 dilutions of anti-DNAJA1 antibody (11713-1-AP, Proteintech) in PBS containing 1% (wt/vol) BSA and 0.1% (vol/vol) Tween20. After 3× washes, 1:1,000 dilution of Alexa Fluor 488 anti-rabbit IgG in 1% BSA PBS was used to incubate the cells for 1 h at RT as a secondary antibody. A drop of ProLong diamond antifade mountant containing DAPI (Thermo Fisher Scientific) was used to mount the cells. Data were collected with Leica Application Suite X. Image analysis was performed using ImageJ (v.1.53). Pearson's and Mander's colocalization coefficients were calculated with the JACoP plug-in[41]. Calculation from the independent images was reported.

For monitoring mitochondrial import, MTS-EGFP (Addgene, catalogue no. 23214) was transiently transfected together with 10 μM GTPP treatment. After 6 h of incubation, cells were stained with 50 nM Mitotracker Deep Red FM (Thermo Fisher Scientific) for 15 min and transferred to RPMI 10% fetal calf serum for live cell imaging. CQ1 (Yokogawa) with 40× magnification was used with the following laser settings: 488-nm excitation and 525/50-nm emission for EGFP and 640-nm excitation and 685/40-nm emission for Mitotracker Deep Red FM. Three wells with a total of 100 cells per condition were manually characterized into one of five categories.

For Halo-tagged reporter assay, T-Rex-HeLa cells stably expressing Halo-tagged ATP5A1 and GREPL1 were used. Blocking of previously synthesized Halo-tagged proteins was done by incubating cells in media containing 5 μM empty HaloTag ligand (Promega) overnight. Treatments were started on the following day. Newly synthesized Halo-tagged proteins were labelled with 5 μM HaloTag TMR ligand (Promega) in the last hour of the treatments. Mitochondria were stained with 50 nM Mitotracker Deep Red FM (Thermo Fisher Scientific). CQ1 (Yokogawa) with 40× magnification was used with following laser settings: 488-nm excitation and 525/50-nm emission for TMR and 561-nm excitation and 617/73-nm emission for Mitotracker Deep Red FM. Several images were collected from three independent replicates, and image analysis was performed using ImageJ (v.1.53). Pearson's and Mander's colocalization coefficients were calculated with the JACoP plug-in[41]. Calculation from three independent replicates was reported.

## Cell viability assay

Cell viability was measured with Cell Counting Kit-8 (Dojindo) according to the manufacturer's instructions. Cell viability was evaluated 16 h (overnight) after treatment with the different chemicals used in the experiment.

## Statistics and plots

A general statistical analysis was performed with a two-tailed Student's $t$ test (considered significant for $P \leq 0.05$), unless it was stated otherwise. All plots were created using the R packages ggplot2 (v.3.3.3), gplots (v.3.1.1) and RColorBrewer (v.1.1-2). Visualization of the final figures was done with Adobe Illustrator CS5.

## Reporting summary

Further information on research design is available in the Nature Portfolio Reporting Summary linked to this article.

## Data availability

The transcriptomics data have been deposited to the European Nucleotide Archive at EMBL-EBI under accession number PRJEB61069. The mass spectrometry proteomics data have been deposited to the ProteomeXchange Consortium[42] via the PRIDE partner repository[43] with the dataset identifier PXD031948 for the DNAJA1 interaction proteomics and PXD032011 for the redox proteomics data. HSF1 and ATF5 CHIP–seq data were obtained from the ENCODE[44,45] database (https://www.encodeproject.org/). The HSF1 CHIP–seq dataset accession is ENCSR000EET, and the file accession is ENCFF797ENQ. The ATF5 CHIP–seq dataset accession is ENCSR887TWV, and the file accession is ENCFF638RRU. Datasets representing the key findings of this paper are within the main and supplementary figures and tables of this article. Unprocessed images are available on request from the corresponding author. Source data are provided with this paper.

## Code availability

No custom code was used in this study.

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

**Acknowledgements** We thank T. Engleitner, R. Öllinger and R. Rad (TU Munich) for conducting the RNA sequencing run, mapping and normalization; the Quantitative Proteomics Unit at Goethe University for support with mass spectrometry measurements; and J. Riemer (University of Cologne) and R. Brandes (Goethe University, Frankfurt) for their critical feedback on the manuscript. This work was supported by the European Research Council under the European Union's Horizon 2020 Research and Innovation Programme (Grant ERC StG 803565 to C.M.), the Deutsche Forschungsgemeinschaft (German Research Foundation; Projects 390339347 (Emmy Noether Programme), 259130777 (SFB1177 Selective Autophagy), 456687919 (SFB1531 Stroma-vascular damage control) and 403765277 (mass spectrometer) to C.M., the Hessian Ministry for Arts and Sciences EnABLE Consortium and Fraunhofer High-Performance Center Innovative Therapeutics (TheraNova).

**Author contributions** F.X.R.S. and I.G. performed the experiments and data analysis. G.T. performed the mass spectrometry run and analysis of the DNAJA1 interactome. F.X.R.S., I.G. and C.M. interpreted the data and conceptualized the project. C.M. conceived and supervised this project and obtained funding. F.X.R.S. and C.M. wrote the paper with contributions from all authors.

**Competing interests** The authors declare no competing interests.

**Additional information**
**Correspondence and requests for materials** should be addressed to Christian Münch.

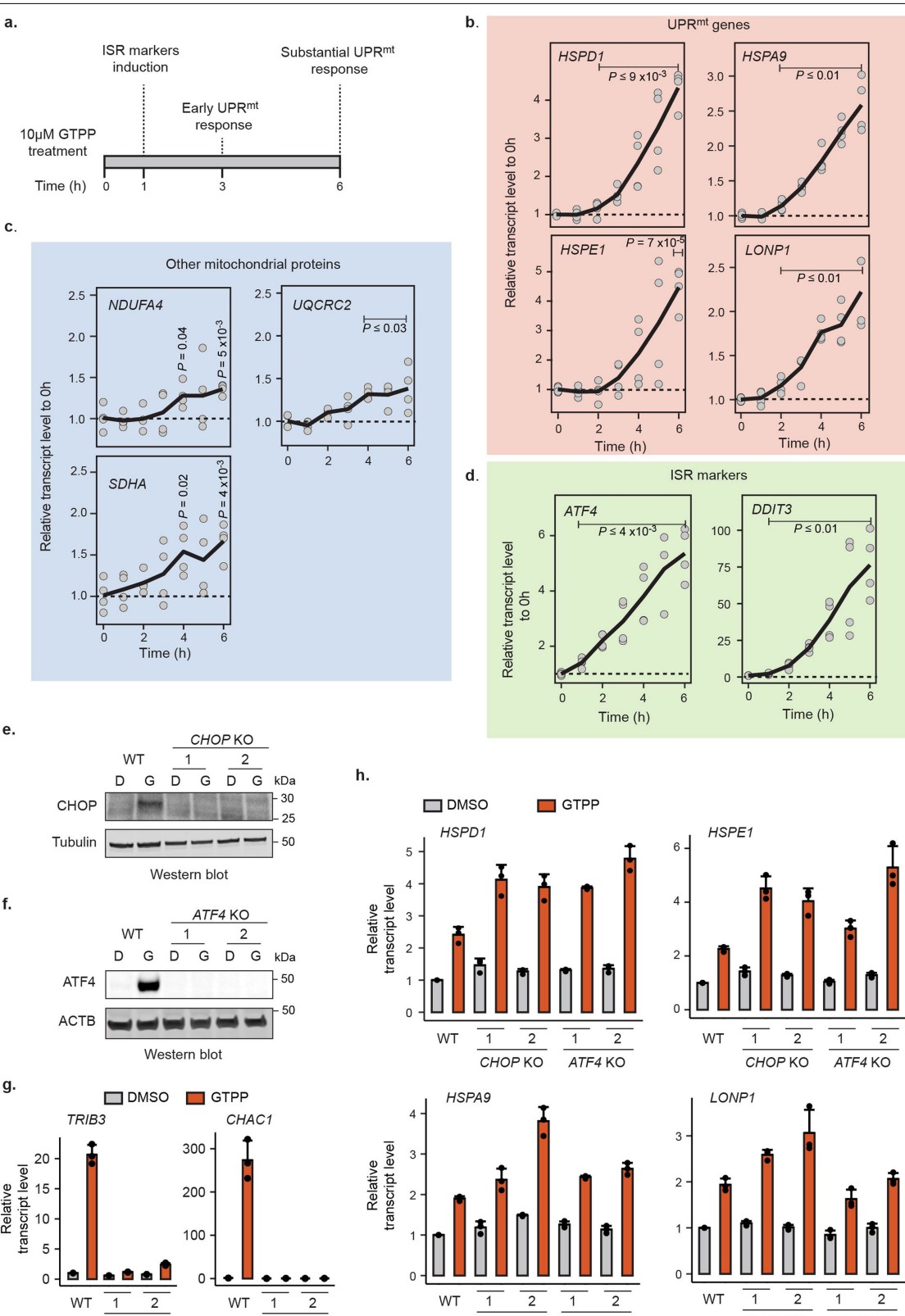

**Extended Data Fig. 1** | See next page for caption.

**Extended Data Fig. 1 | The ATF4-ISR axis is not required for UPR^mt induction.**
**a**, The experimental timeline of UPR^mt gene induction upon GTPP treatment.
**b**–**d**, Time-resolved monitoring of relative transcript levels of (**b**) UPR^mt genes,
(**c**) other mitochondrial proteins, and (**d**) ISR markers upon GTPP treatment
measured with qPCR ($n = 4$ biological replicates). The line indicates the mean
of transcript levels across different time points. **e**,**f**, Western blot images of
(**e**) CHOP and (**f**) ATF4 in HeLa *CHOP* and *ATF4* knock-out cells, respectively in
comparison to wild type (WT). "D" represents DMSO- and "G" represents
GTPP-treated HeLa cells. Numbers indicate different clones used in the
experiments. **g**, Barplots showing the mean of relative transcript levels of
known ATF4 targets upon GTPP treatment in HeLa WT and *ATF4* knock-out cells
measured with qPCR ($n = 3$ biological replicates). **h**, Barplots showing the mean
of relative transcript levels of indicated UPR^mt genes upon GTPP treatment in
HeLa *CHOP* and *ATF4* knock-out cells in comparison to WT measured with qPCR
($n = 3$ biological replicates). All *P* values are calculated with a two-tailed unpaired
Student's t-test and indicated in the figure. All error bars represent mean+SD.
For gel source data, see Supplementary Fig. 1.

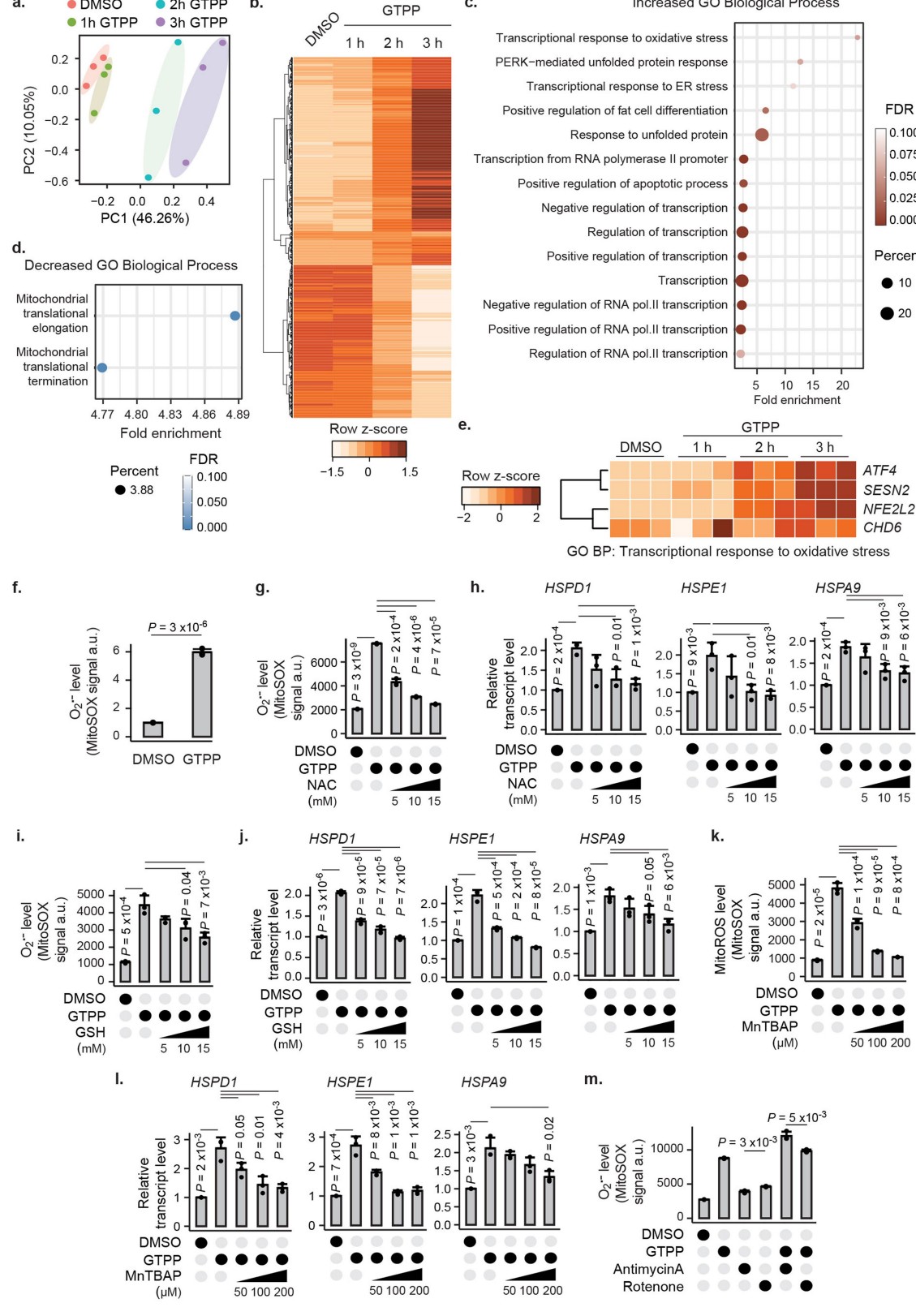

**Extended Data Fig. 2 |** See next page for caption.

**Extended Data Fig. 2 | Time-resolved transcriptomic analyses reveal ROS requirement for UPR^mt induction. a**, Principal component analysis plot of transcriptomics from GTPP treated samples ($n = 3$ biological replicates). **b**, Heat map of increased (489) and decreased (383) transcripts within 3 h GTPP treatment. Transcript levels are represented as z-scores normalised for each gene (row z-score). **c,d**, Enrichment of GO BP terms from transcripts that are (**c**) increased or (**d**) decreased within 3 h GTPP treatment. Only GO terms with FDR < 0.1 are represented on the figures. **e**, Heat map representation of transcript levels of detected genes in the transcriptomic analyses that belong to GO BP: Transcriptional response to oxidative stress (GO:0036091). Transcript levels are represented as z-scores normalised for each gene (row z-score). **f**, Barplot showing the mean of mitochondrial ROS ($O_2^{\cdot-}$) levels of HeLa cells treated with GTPP measured on FACS using MitoSOX ($n = 3$ biological replicates). **g,h**, Barplots showing the mean of mitochondrial ROS ($O_2^{\cdot-}$) levels measured on FACS using MitoSOX (**g**) and relative transcript levels of indicated UPR^mt genes measured with qPCR (**h**) of HeLa cells upon co-treatments of GTPP with titrated concentrations of NAC ($n = 3$ biological replicates). **i,j**, Barplots showing the mean of mitochondrial ROS ($O_2^{\cdot-}$) levels measured on FACS using MitoSOX (**i**) and relative transcript levels of indicated UPR^mt genes measured with qPCR (**j**) of HeLa cells upon co-treatments of GTPP with titrated concentrations of GSH ($n = 3$ biological replicates). **k,l**, Barplots showing the mean of mitochondrial ROS ($O_2^{\cdot-}$) levels measured on FACS using MitoSOX (**k**) and relative transcript levels of indicated UPR^mt genes measured with qPCR (**l**) of HeLa cells upon co-treatments of GTPP with titrated concentrations of MnTBAP ($n = 3$ biological replicates). **m**, Barplot showing the mean of mitochondrial ROS ($O_2^{\cdot-}$) levels of HeLa cells upon co-treatments of GTPP with antimycin A or rotenone measured on FACS using MitoSOX ($n = 3$ biological replicates). All $P$ values are calculated with a two-tailed unpaired Student's t-test and indicated in the figure. All error bars represent mean+SD.

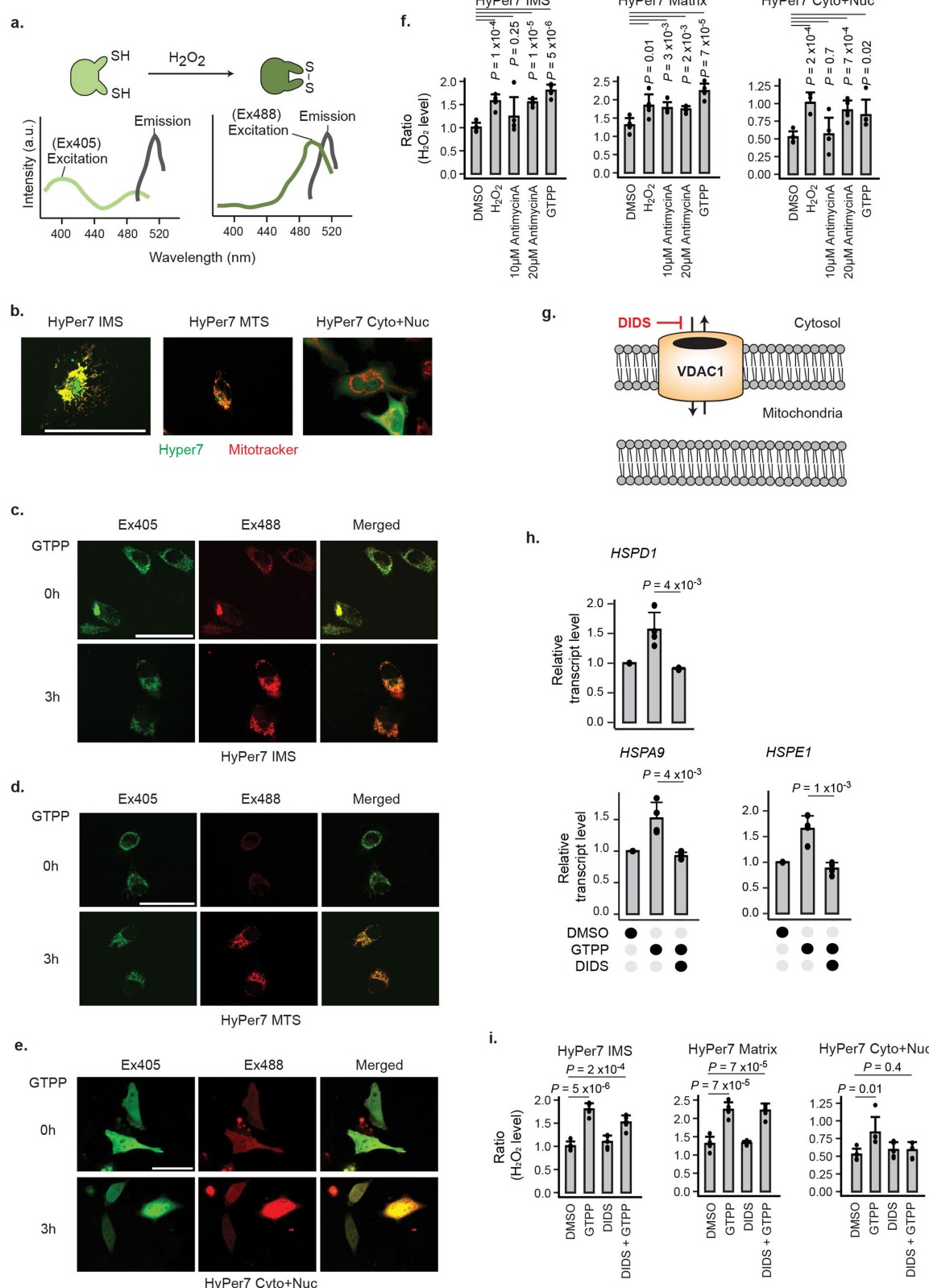

**Extended Data Fig. 3** | See next page for caption.

**Extended Data Fig. 3 | ROS are produced in mitochondria upon MMS.**
**a**, Schematic illustration of HyPer7 fluorophore changes upon reaction with $H_2O_2$. Excitation of reduced and oxidized forms of the fluorophore are represented as light and dark green, respectively. **b**, Microscopy images of different constructs of HyPer7 targeted to the inter membrane space (IMS), the matrix (MTS) and untargeted (Cyto+Nuc) ($n$ = 3). HyPer7 and mitotracker signals are shown in green and red, respectively. **c**–**e**, Representative microscopy images of HyPer7 targeted to the IMS (**c**), the matrix (**d**) or untargeted (**e**) of HeLa cells treated with GTPP for 3h ($n$ = 5 biological replicates). Reduced (Ex405) and oxidized (Ex488) forms of HyPer7 are shown in green and red, respectively. **f**, Barplots showing the mean of $H_2O_2$ level measured with Hyper7 reporters targeted to the IMS, the matrix, and untargeted (cytosol+nucleus) upon 3h GTPP and different treatments as references ($n$ = 5 biological replicates, 100 cells were analysed for each replicate). **g**, Schematic illustration for the mechanistic inhibition of VDAC1 by DIDS. **h**, Barplots depicting the mean of relative transcript levels of indicated UPR[mt] genes of GTPP-treated HeLa cells upon co-treatments with DIDS ($n$ = 4 biological replicates). **i**, Barplots showing the mean of $H_2O_2$ level measured with Hyper7 reporters targeted to the IMS, the matrix and untargeted (cytosol+nucleus) upon co-treatments of GTPP with DIDS ($n$ = 5 biological replicates, >100 cells were analysed for each replicate). All scale bars indicate 50 µm. All $P$ values are calculated with a two-tailed unpaired Student's t-test and indicated in the figure. All error bars represent mean+SD.

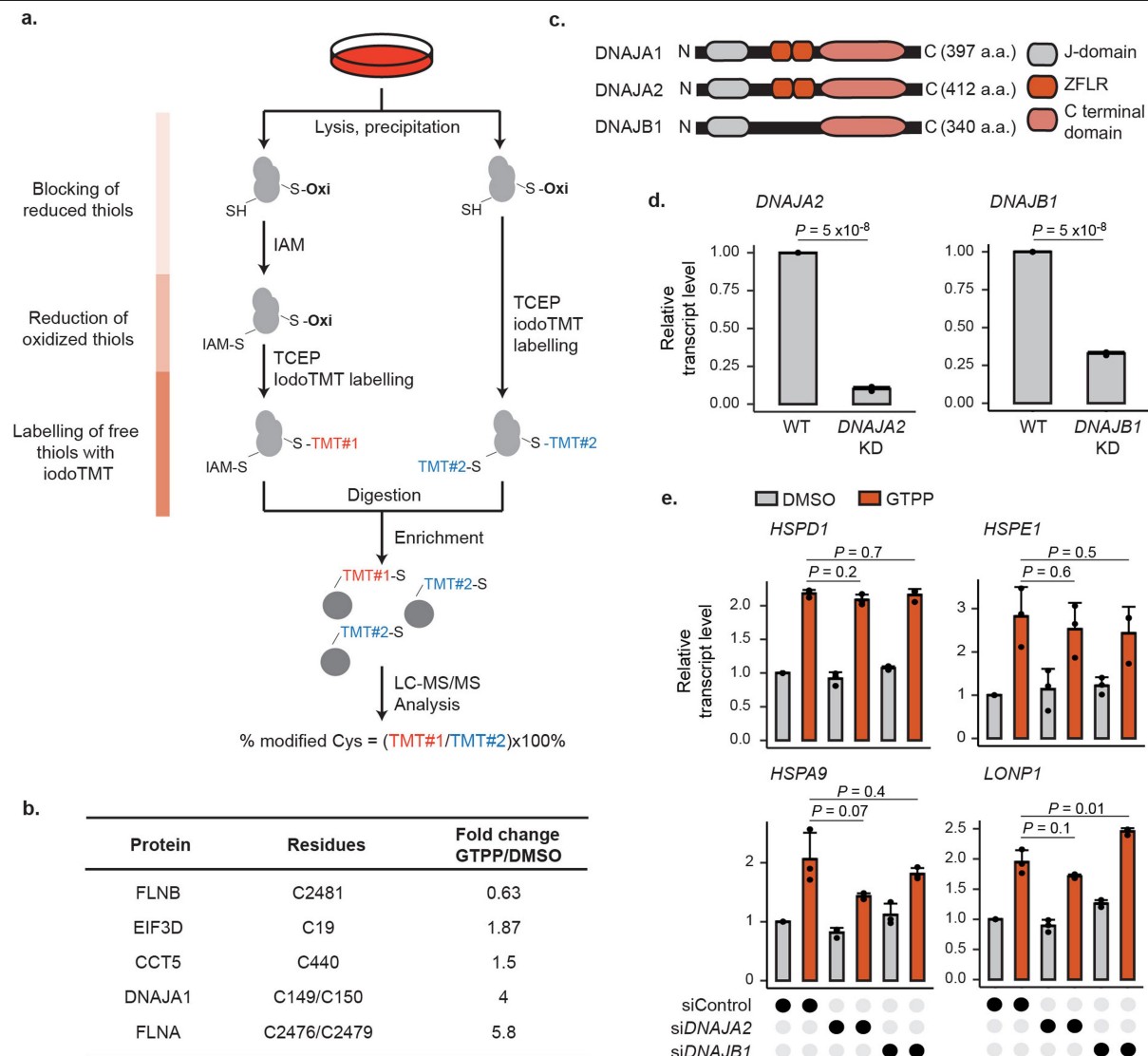

**Extended Data Fig. 4 | DNAJA1 oxidation by MMS-induced ROS is an essential step in UPR^mt signalling. a**, Experimental scheme for redox proteomics upon GTPP treatment in HeLa cells. **b**, List of cysteines with significantly changed oxidation levels upon GTPP treatment ($P \leq 0.05$). **c**, Scheme of domain composition from different DNAJ family members. Amino acid (a.a.) lengths are indicated in brackets. ZFLR denotes the zinc finger like region. **d**, Barplots depicting the mean of relative transcript levels of *DNAJA2* and *DNAJB1* in HeLa

*DNAJA2* and *DNAJB1* knock-down cells measured with qPCR, respectively ($n = 3$ biological replicates). **e**, Barplots depicting the mean of relative transcript levels of indicated UPR^mt genes of GTPP-treated HeLa cells upon knock-down of *DNAJA2* or *DNAJB1* measured with qPCR ($n = 3$ biological replicates). All $P$ values are calculated with a two-tailed unpaired Student's t-test and indicated in the figure. All error bars represent mean+SD.

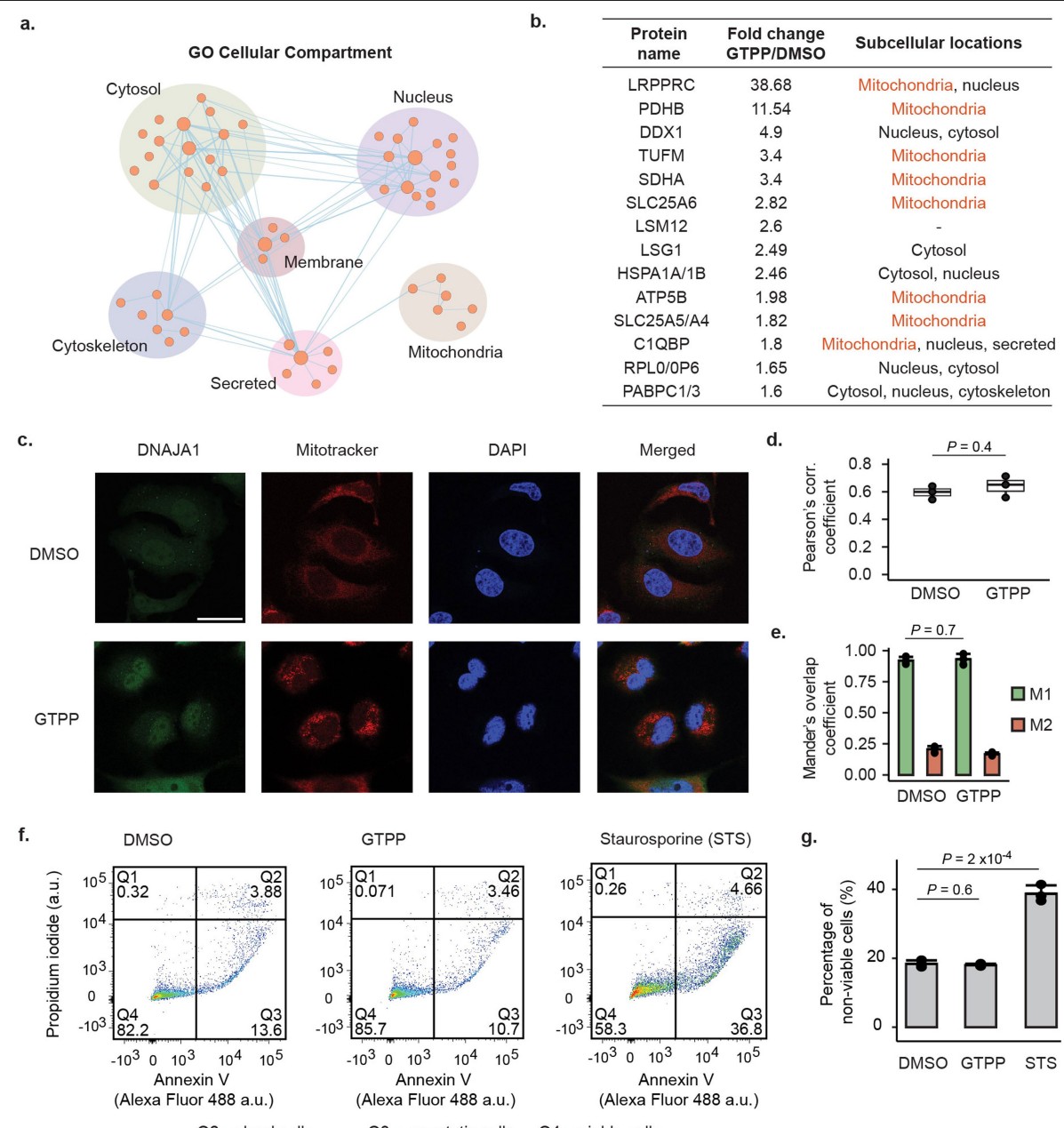

**a**, **GO Cellular Compartment**

Cytosol · Nucleus · Membrane · Cytoskeleton · Secreted · Mitochondria

**b.**

| Protein name | Fold change GTPP/DMSO | Subcellular locations |
|---|---|---|
| LRPPRC | 38.68 | Mitochondria, nucleus |
| PDHB | 11.54 | Mitochondria |
| DDX1 | 4.9 | Nucleus, cytosol |
| TUFM | 3.4 | Mitochondria |
| SDHA | 3.4 | Mitochondria |
| SLC25A6 | 2.82 | Mitochondria |
| LSM12 | 2.6 | - |
| LSG1 | 2.49 | Cytosol |
| HSPA1A/1B | 2.46 | Cytosol, nucleus |
| ATP5B | 1.98 | Mitochondria |
| SLC25A5/A4 | 1.82 | Mitochondria |
| C1QBP | 1.8 | Mitochondria, nucleus, secreted |
| RPL0/0P6 | 1.65 | Nucleus, cytosol |
| PABPC1/3 | 1.6 | Cytosol, nucleus, cytoskeleton |

**c.** DNAJA1 · Mitotracker · DAPI · Merged — DMSO, GTPP

**d.** $P = 0.4$ — Pearson's corr. coefficient — DMSO, GTPP

**e.** $P = 0.7$ — Mander's overlap coefficient — M1, M2 — DMSO, GTPP

**f.** DMSO · GTPP · Staurosporine (STS)

Propidium iodide (a.u.) — Annexin V (Alexa Fluor 488 a.u.)

DMSO: Q1 0.32, Q2 3.88, Q4 82.2, Q3 13.6
GTPP: Q1 0.071, Q2 3.46, Q4 85.7, Q3 10.7
STS: Q1 0.26, Q2 4.66, Q4 58.3, Q3 36.8

Q2 = dead cells   Q3 = apoptotic cells   Q4 = viable cells

**g.** $P = 2 \times 10^{-4}$, $P = 0.6$ — Percentage of non-viable cells (%) — DMSO, GTPP, STS

**Extended Data Fig. 5 | DNAJA1 interacts with c-mtProt upon UPR^mt induction. a**, Enrichment map of GO Cellular Compartment (CC) terms of DNAJA1 interactors from the interaction proteomic analysis in HeLa cells. Circles represent GO terms with FDR<0.1. **b**, List of proteins with increased interactions to DNAJA1 upon GTPP treatment. **c**–**e**, Microscopy images of DNAJA1 (green), mitochondria (red), and nucleus (blue) upon GTPP treatment (**c**). Degrees of co-localization between DNAJA1 and mitochondria (Mitotracker) are represented both as a Pearson's correlation coefficient (**d**) and a Mander's overlap coefficient (**e**) with DNAJA1 as probe 1 and Mitotracker as probe 2 ($n = 3$ biological replicates). Boxplots represent the mean of Pearson's correlation coefficients (**d**), while barplot depicts the mean of Mander's overlap coefficients (**e**). Box indicates the interquartile range (IQR) and whiskers denote the 1.5 × IQR beyond the box. **f**. Changes in cell death upon GTPP treatment, measured by annexin V + propidium iodide staining and cell sorting. Viable cells are represented on the fourth quadrant (Q4), while non-viable cells on the second (Q2) and third quadrants (Q3). Staurosporine (STS) was used as a positive control treatment. The mean of the percentage of non-viable cells is shown as a barplot in **g** ($n = 3$ biological replicates). Scale bar indicates 50 μm. All $P$ values are calculated with a two-tailed unpaired Student's t-test and indicated in the figure. All error bars represent mean+SD.

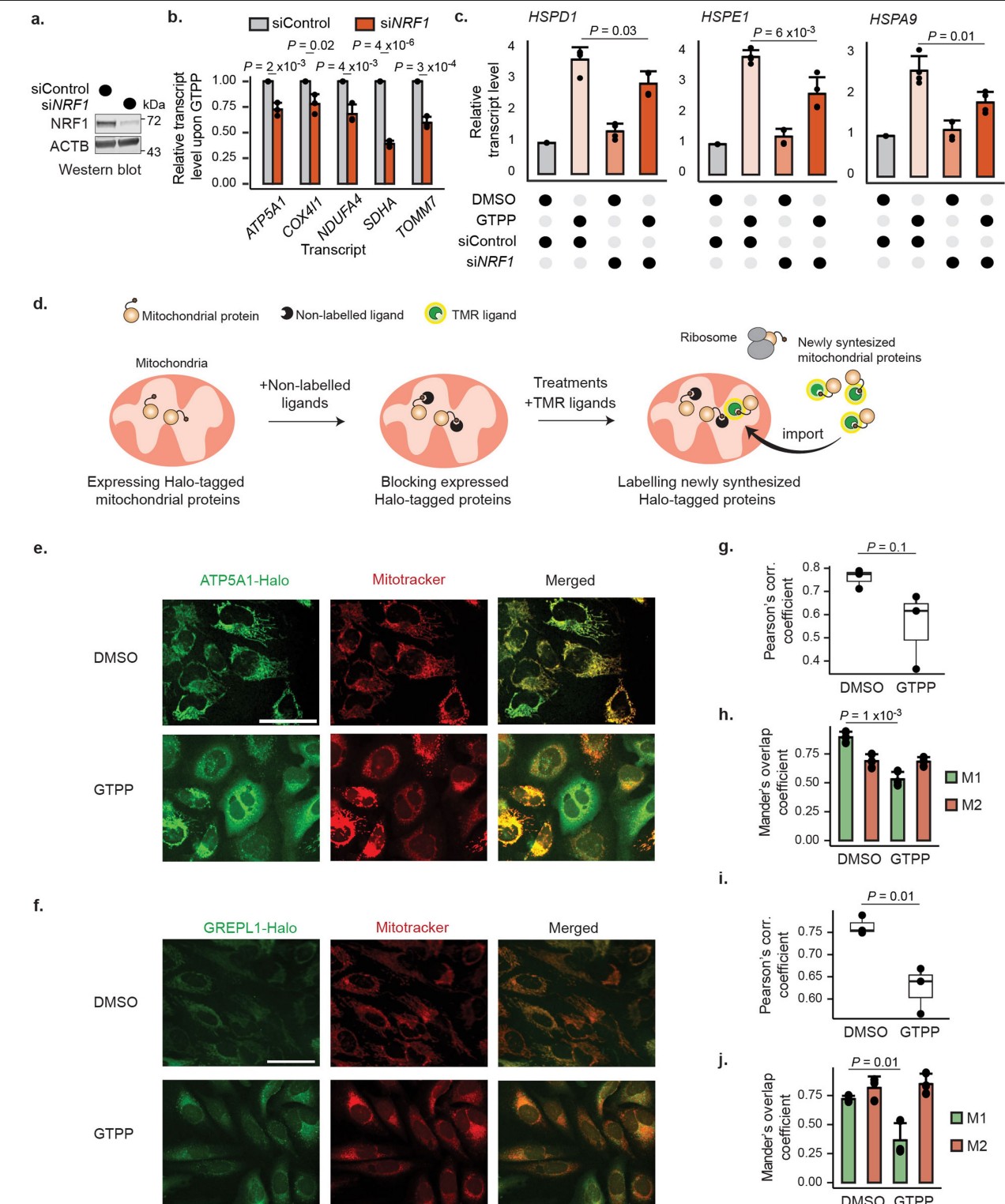

**Extended Data Fig. 6** | See next page for caption.

**Extended Data Fig. 6 | c-mtProt accumulate upon MMS. a**, Western blot images of HeLa *NRF1* knock-down efficiency in comparison to control siRNA (siControl). **b,c**, Barplots depicting the mean of relative transcript levels for several mitochondrial proteins (**b**) and indicated UPR$^{mt}$ genes (**c**) in HeLa *NRF1* knock-down cells upon treatment with GTPP in comparison to siControl measured with qPCR (*n* = 3 biological replicates). **d**, Scheme of Halo-tagged protein reporter assay to monitor mitochondrial precursor protein import. HeLa cells stably expressing Halo-tagged mitochondrial proteins were incubated overnight with non-labelled Halo ligands to block the Halo tags of existing proteins. Newly synthesized proteins (non-blocked Halo tag) can be monitored with TMR-labelled Halo ligands. **e**, Representative microscopy images of cells stably expressing Halo-tagged ATP5A1. **f,g** Degrees of co-localization between Halo-tagged ATP5A1 and mitochondria (Mitotracker) are represented both as a Pearson's correlation coefficient (**f**) and a Mander's overlap coefficient (**g**).

**h**, Representative microscopy images of cells stably expressing Halo-tagged GRPEL1. **i,j** Degrees of co-localization between Halo-tagged GRPEL1 and mitochondria (Mitotracker) are represented both as a Pearson's correlation coefficient (**i**) and a Mander's overlap coefficient (**j**). Both Halo-tagged protein quantifications were done with 3 biological replicates. For all Mander's overlap coefficient quantifications, Halo-tagged proteins were used as probe 1 and Mitotracker as probe 2. Boxplots (**g,i**) represent the mean of Pearson's correlation coefficients, while barplots (**h,j**) depict the mean of Mander's overlap coefficients. All boxes indicate the interquartile range (IQR) and whiskers denote the 1.5 × IQR beyond the box. All scale bars indicate 50 μm. All *P* values are calculated with a two-tailed unpaired Student's t-test and indicated in the figure. All error bars represent mean+SD. For gel source data, see Supplementary Fig. 1.

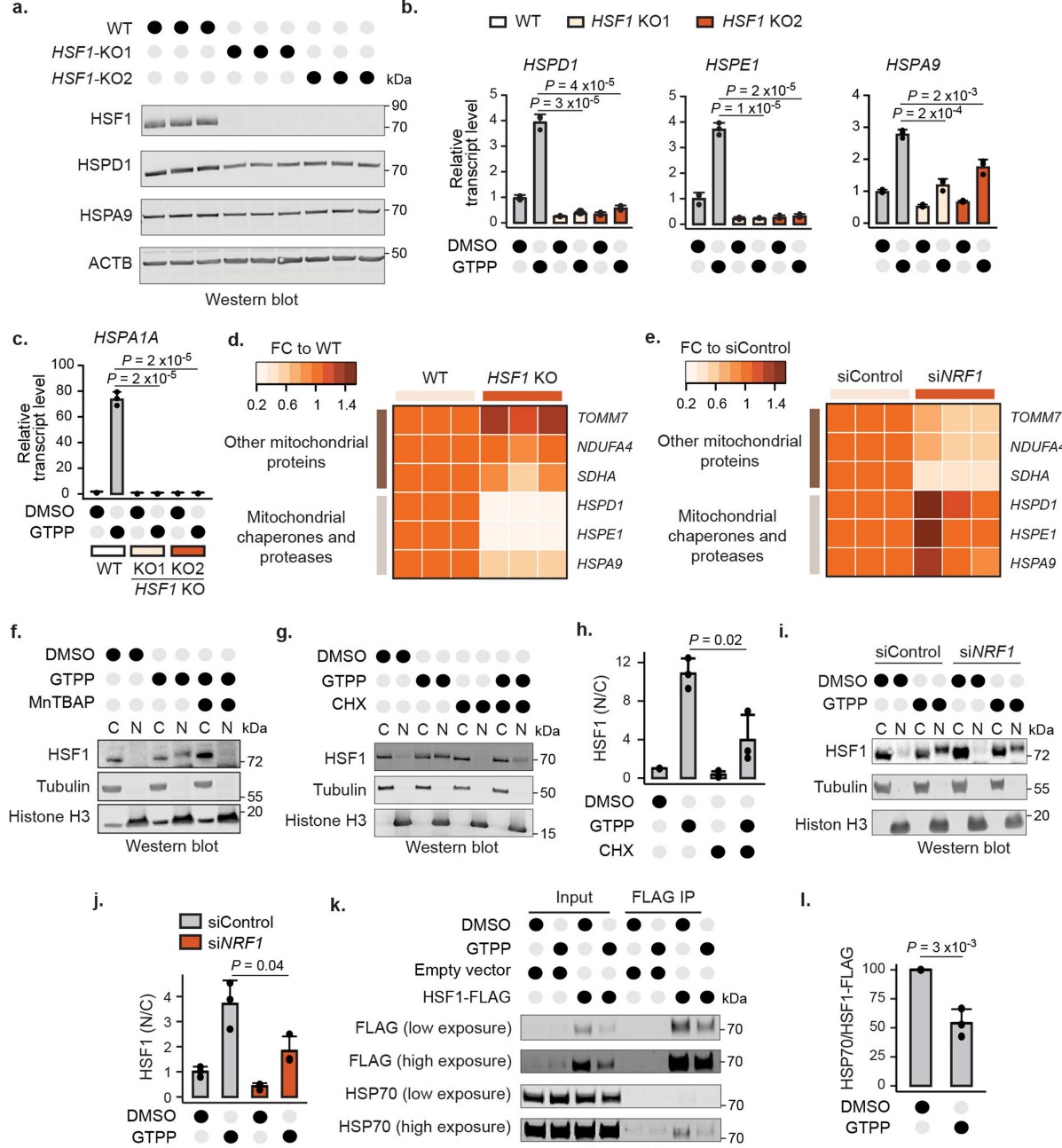

**Extended Data Fig. 7 | HSF1 activation during UPR^mt is dependent on the translation of mitochondrial proteins. a,** Western blot images of mitochondrial chaperones in HeLa WT and *HSF1* knock-out cells. **b,c,** Barplots depicting the mean of relative transcript levels of (**b**) several UPR^mt markers and (**c**) an HSF1 target gene in HeLa *HSF1* knock-out cells upon treatment with GTPP in comparison to WT measured with qPCR (*n* = 3 biological replicates). **d,e,** Heat maps of relative transcript levels of indicated UPR^mt genes (mitochondrial chaperones and proteases) and other mitochondrial proteins upon (**d**) *HSF1* knock-out and (**e**) *NRF1* knock-down measured with qPCR (*n* = 3 biological replicates). **f,g,** Representative Western blot images of HSF1 in cytosolic (C) and nuclear (N) fractions of GTPP-treated HeLa cells upon co-treatment with (**f**) MnTBAP and (**g**) cycloheximide. **h,** Barplot depicting the mean of the

nuclear to cytosolic HSF1 ratio (N/C) from the Western blots shown in **g** (*n* = 3 biological replicates). **i,** Representative Western blot images of HSF1 in cytosolic (C) and nuclear (N) fractions in HeLa *NRF1* knock-down cells in comparison to siControl. **j,** Barplot depicting the mean of the nuclear to cytosolic HSF1 ratio (N/C) from the Western blots shown in **i** (*n* = 3 biological replicates). **k,** Representative Western blot images of wild type FLAG tagged HSF1-HSP70 interactions upon GTPP treatment. **l,** Barplot depicting the mean of the HSP70 band intensities in comparison to HSF1 from the Western blots shown in **j** (*n* = 3 biological replicates). All *P* values are calculated with a two-tailed unpaired Student's t-test and indicated in the figure. All error bars represent mean+SD. For gel source data, see Supplementary Figs. 1, 2.

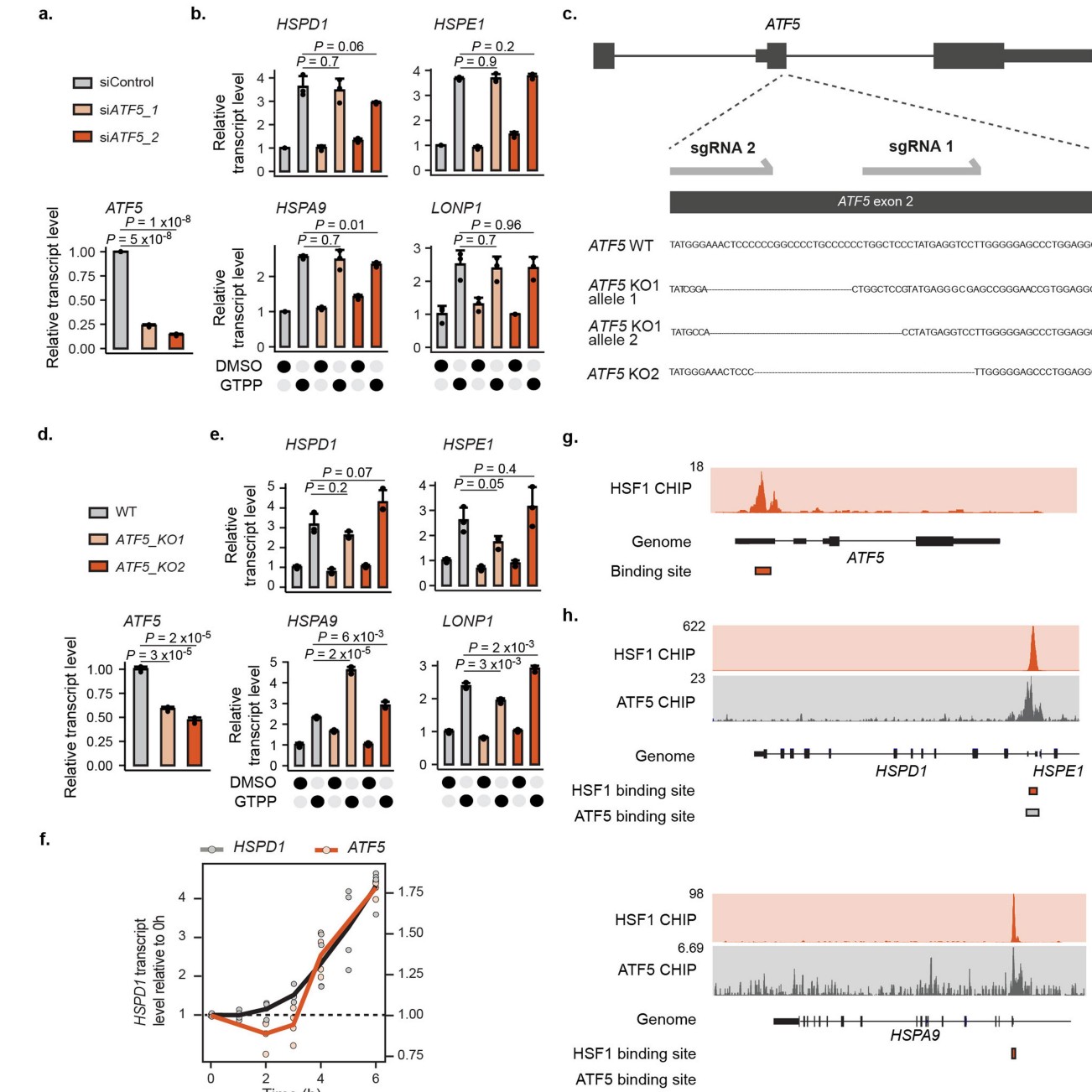

**Extended Data Fig. 8 | HSF1 acts upstream of ATF5. a**, **b**, Barplots depicting the mean of relative transcript levels of (**a**) *ATF5* and (**b**) several UPR<sup>mt</sup> markers in HeLa *ATF5* knock-down cells upon treatment with GTPP in comparison to siControl measured with qPCR (*n* = 3 biological replicates). **c**, Sequencing map showing the location of deleted sequences on HeLa *ATF5* knock-out cells identified with Sanger sequencing. **d**,**e**, Barplots depicting the mean of relative transcript levels of (**d**) *ATF5* and (**e**) several UPR<sup>mt</sup> markers in HeLa *ATF5* knock-out cells upon treatment with GTPP in comparison to WT measured with qPCR (*n* = 3 biological replicates). **f**, Time-resolved monitoring of relative transcript levels of *ATF5* upon 6 h of GTPP treatment (*n* = 4 biological

replicates). The lines indicate the mean of transcript levels across different time points. **g**, HSF1 CHIP-seq data at the *ATF5* gene location. At the genome panel, boxes represent exons and strings represent introns. Read counts represent binding sites at the particular genome location. Binding sites were identified as read counts passing the irreproducible discovery rate threshold. **h**, HSF1 and ATF5 CHIP-seq data at the *HSPD1*, *HSPE1* and *HSPA9* gene locations in orange and black, respectively. All *P* values are calculated with a two-tailed unpaired Student's t-test and indicated in the figure. All error bars represent mean+SD.

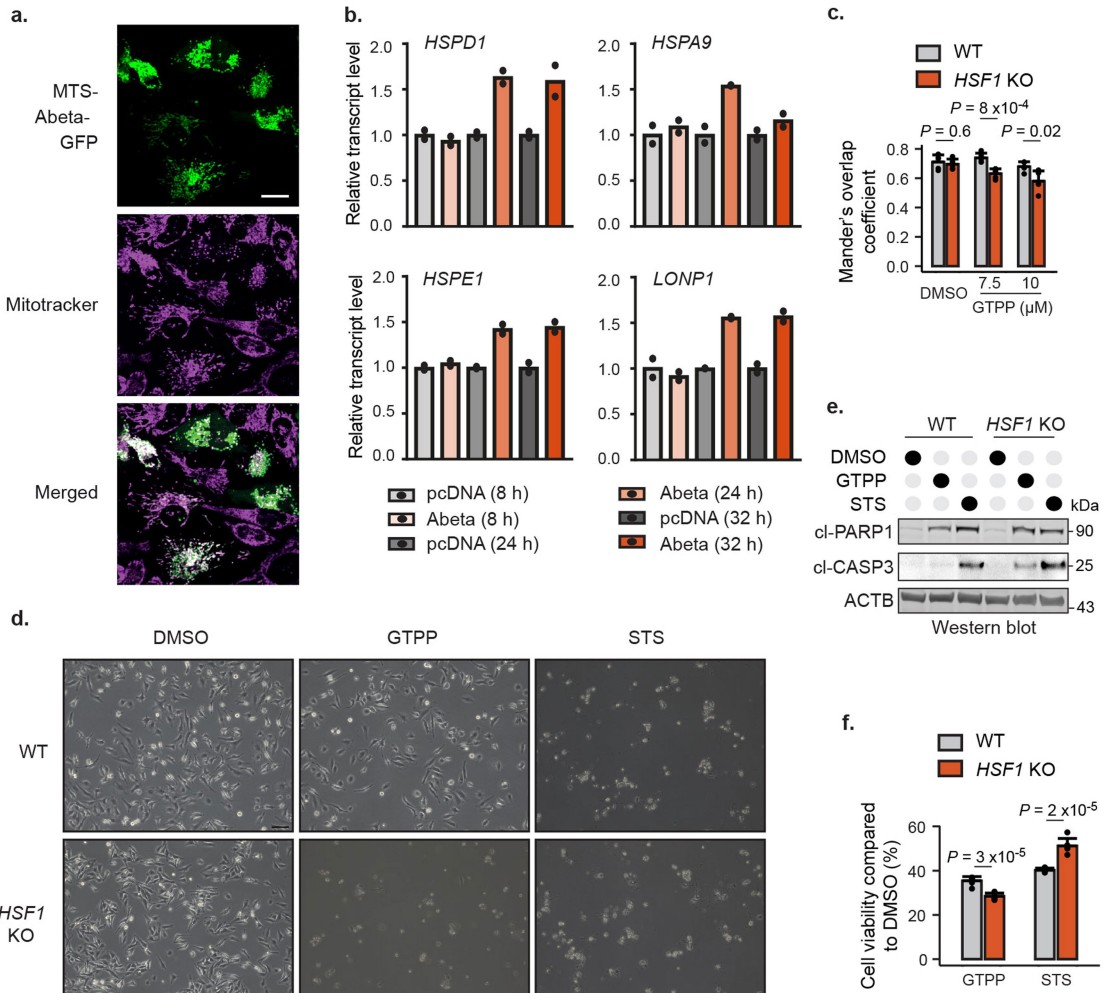

**Extended Data Fig. 9 | HSF1 increases cell survival upon MMS.**
**a**, Representative microscopy images of HeLa cells transiently expressing MTS-GFP-tagged Abeta (green). Mitochondria were stained with Mitotracker (magenta). Scale bar indicates 20 μm. **b**, Barplots depicting the mean of relative transcript levels of indicated UPR[mt] genes of HeLa cells upon overexpression of MTS-GFP tagged Abeta measured with qPCR at different time points post-transfection ($n$ = 2 biological replicates). **c**, Degree of co-localization of MTS-EGFP and mitochondria (Mitrotracker) is represented as a Mander's overlap coefficient (M1) with MTS-EGFP as probe 1 and Mitotracker as probe 2 ($n$ = 5 biological replicates). Barplot represents the mean of Mander's overlap

coefficients. **d**–**f**, (**d**) Representative microscopy and (**e**) western blot images of WT and *HSF1* knock-out HeLa cells treated for 16 h with DMSO, 15 μM GTPP and 200 nM staurosporine (STS). Scale bar indicates 100 μm. Western blot images show the levels of apoptosis markers, including cleaved-PARP1 and cleaved-caspase 3 under the different treatments. Viability of cells under the treatments is calculated and represented as (**f**) barplot ($n$ = 6 biological replicates). All $P$ values are calculated with a two-tailed unpaired Student's t-test and indicated in the figure. All error bars represent mean+SD. For gel source data, see Supplementary Fig. 1.

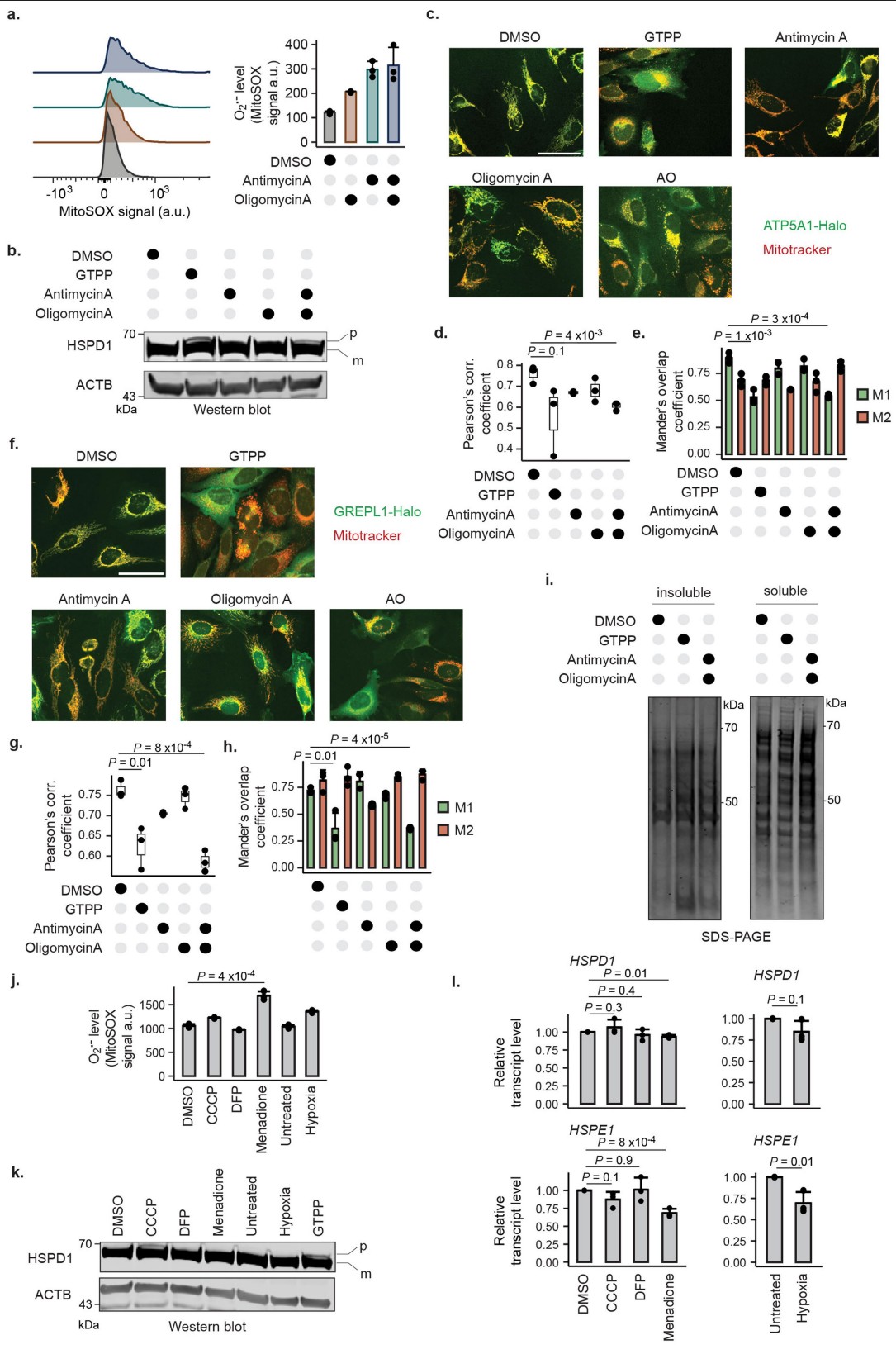

**Extended Data Fig. 10** | See next page for caption.

**Extended Data Fig. 10 | ROS and accumulation of c-mtProt are required as signals to induce the UPR^mt. a**, Measurement of mitochondrial ROS ($O_2^{\cdot-}$) levels on FACS with MitoSOX of HeLa cells treated with different conditions (left panel) and the mean of triplicate quantification as a barplot (right panel). **b**, Western blot images of mitochondrial proteins in their precursor (p) and mature (m) forms upon different treatments. **c**, Representative microscopy images of cells stably expressing Halo-tagged ATP5A1. **d,e** Degrees of co-localization between Halo-tagged ATP5A1 and mitochondria (Mitotracker) are represented both as a Pearson's correlation coefficient (**d**) and a Mander's overlap coefficient (**e**) ($n$ = 3 biological replicates). **f**, Representative microscopy images of cells stably expressing Halo-tagged GRPEL1. **g,h** Degrees of co-localization between Halo-tagged GRPEL1 and mitochondria (Mitotracker) are represented both as a Pearson's correlation coefficient (**g**) and a Mander's overlap coefficient (**h**) ($n$ = 3 biological replicates). For all Mander's overlap coefficient quantifications, Halo-tagged proteins were used as probe 1 and Mitotracker as probe 2. Boxplots (**d**,**g**) represent the mean of Pearson's correlation coefficients, while barplots (**e**,**h**) depict the mean of Mander's overlap coefficients. All boxes indicate the interquartile range (IQR) and whiskers denote the 1.5 × IQR beyond the box. Halo-tagged proteins and Mitotracker are shown in green and red, respectively. **i**, Gel image of mitochondrial insoluble and soluble fractions upon different treatments. Insoluble fractions are used as a measure for aggregate formation. **j**, Barplot showing the mean of mitochondrial ROS ($O_2^{\cdot-}$) levels measured on FACS using MitoSOX of HeLa cells treated with different mitochondrial stressors ($n$ = 3 biological replicates). **k**, Western blot images of mitochondrial proteins in their precursor (p) and mature (m) forms upon treatment with different mitochondrial stressors. **l**, Barplots depicting the mean of relative transcript levels of indicated UPR^mt genes of HeLa cells upon treatment with different mitochondrial stressors measured with qPCR ($n$ = 3 biological replicates). All scale bars indicate 50 μm. All *P* values are calculated with a two-tailed unpaired Student's t-test and indicated in the figure. All error bars represent mean+SD. For gel source data, see Supplementary Fig. 1.

# Reporting Summary

## Statistics

For all statistical analyses, confirm that the following items are present in the figure legend, table legend, main text, or Methods section.

| n/a | Confirmed | |
|---|---|---|
| ☐ | ☒ | The exact sample size (*n*) for each experimental group/condition, given as a discrete number and unit of measurement |
| ☐ | ☒ | A statement on whether measurements were taken from distinct samples or whether the same sample was measured repeatedly |
| ☐ | ☒ | The statistical test(s) used AND whether they are one- or two-sided<br>*Only common tests should be described solely by name; describe more complex techniques in the Methods section.* |
| ☒ | ☐ | A description of all covariates tested |
| ☒ | ☐ | A description of any assumptions or corrections, such as tests of normality and adjustment for multiple comparisons |
| ☐ | ☒ | A full description of the statistical parameters including central tendency (e.g. means) or other basic estimates (e.g. regression coefficient) AND variation (e.g. standard deviation) or associated estimates of uncertainty (e.g. confidence intervals) |
| ☐ | ☒ | For null hypothesis testing, the test statistic (e.g. $F$, $t$, $r$) with confidence intervals, effect sizes, degrees of freedom and $P$ value noted<br>*Give P values as exact values whenever suitable.* |
| ☒ | ☐ | For Bayesian analysis, information on the choice of priors and Markov chain Monte Carlo settings |
| ☒ | ☐ | For hierarchical and complex designs, identification of the appropriate level for tests and full reporting of outcomes |
| ☐ | ☒ | Estimates of effect sizes (e.g. Cohen's *d*, Pearson's *r*), indicating how they were calculated |

*Our web collection on statistics for biologists contains articles on many of the points above.*

## Software and code

Policy information about availability of computer code

| | |
|---|---|
| Data collection | For data collection, following softwares were used : FACSDiva (v.6.1.3), LightCycler 480 SW (v.1.5), Image Studio (v.5.2), ImageLab 6.0.1, Leica Application Suite X, CQ1 Measurement |
| Data analysis | For data analysis following softwares were used : Drop-Seq tools (v.1.12), DESeq2 (v1.18.1), mFuzz (v.2.50.0), EnrichmentMap (v.3.3.2) , Cytoscape (v.3.7.1), FlowJo (v.10) , Proteome Discoverer (v.2.4), MaxQuant (v.1.6.17.0) , ImageJ (v.1.53) with JACoP plugin, RStudio (v.4.0.5), ggplot2 (v.3.3.3), gplots (v.3.1.1), RColorBrewer (v.1.1-2), Adobe Illustrator CS5. |

For manuscripts utilizing custom algorithms or software that are central to the research but not yet described in published literature, software must be made available to editors and reviewers. We strongly encourage code deposition in a community repository (e.g. GitHub). See the Nature Portfolio guidelines for submitting code & software for further information.

## Data

Policy information about availability of data

All manuscripts must include a data availability statement. This statement should provide the following information, where applicable:
- Accession codes, unique identifiers, or web links for publicly available datasets
- A description of any restrictions on data availability
- For clinical datasets or third party data, please ensure that the statement adheres to our policy

The transcriptomics data have been deposited to the European Nucleotide Archive (ENA) at EMBL-EBI under accession number PRJEB61069. The mass spectrometry proteomics data have been deposited to the ProteomeXchange Consortium via the PRIDE partner repository with the dataset identifier PXD031948 for the DNAJA1 interaction proteomics, and PXD032011 for the redox proteomics data. HSF1 and ATF5 CHIP-seq data were obtained from the ENCODE database (https://www.encodeproject.org/). HSF1 CHIP-seq dataset accession is ENCSR000EET and the file accession is ENCFF797ENQ. ATF5 CHIP-seq dataset accession is

ENCSR887TWV and the file accession is ENCFF638RRU. Data sets representing the key findings of this paper are within the main and supplementary figures/tables of this article. Supplementary Information for this manuscript contains: Supplementary Fig. 1. Uncropped gel scans, Supplementary Fig 2. All gel images used for quantification, Supplementary Fig. 3. Examples for gating strategy of flow cytometry analysis, and Supplementary Table 4. Primer list for quantitative PCR analysis. Unprocessed images are available on request from the corresponding author. The data underlying the graphical representations used in the main and extended figures are provided as Source Data.

# Field-specific reporting

Please select the one below that is the best fit for your research. If you are not sure, read the appropriate sections before making your selection.

☒ Life sciences ☐ Behavioural & social sciences ☐ Ecological, evolutionary & environmental sciences

For a reference copy of the document with all sections, see nature.com/documents/nr-reporting-summary-flat.pdf

# Life sciences study design

All studies must disclose on these points even when the disclosure is negative.

| | |
|---|---|
| Sample size | Sample sizes are indicated in the figure legends. No sample size calculation was performed, sample sizes were chosen based on previous experience and on what is common practice in the field (Münch Nature 2016, Michaelis Nat. Commun. 2022, Fiorese Curr. Bio. 2016, Schäfer Mol. Cell 2021). |
| Data exclusions | No data were excluded |
| Replication | Replicates were used for all experiments. At least 2 biological replicates were used to make sure reproducibility of the experiments. For quantification at least 3 biological replicates were used. All attempts to replicate the experiments were successful. |
| Randomization | All samples were randomly assigned to different treatment groups. |
| Blinding | RNAseq experiment was done by third parties who were not aware of specific treatments that were applied to the cells. Injections of peptides for mass spectrometry analysis were performed blindly. For all other experiments the investigators were not blinded to allocation during experiments and outcome assessment, because the findings were not predicted to follow a single hypothesis that might lead to a bias. |

# Behavioural & social sciences study design

All studies must disclose on these points even when the disclosure is negative.

| | |
|---|---|
| Study description | Briefly describe the study type including whether data are quantitative, qualitative, or mixed-methods (e.g. qualitative cross-sectional, quantitative experimental, mixed-methods case study). |
| Research sample | State the research sample (e.g. Harvard university undergraduates, villagers in rural India) and provide relevant demographic information (e.g. age, sex) and indicate whether the sample is representative. Provide a rationale for the study sample chosen. For studies involving existing datasets, please describe the dataset and source. |
| Sampling strategy | Describe the sampling procedure (e.g. random, snowball, stratified, convenience). Describe the statistical methods that were used to predetermine sample size OR if no sample-size calculation was performed, describe how sample sizes were chosen and provide a rationale for why these sample sizes are sufficient. For qualitative data, please indicate whether data saturation was considered, and what criteria were used to decide that no further sampling was needed. |
| Data collection | Provide details about the data collection procedure, including the instruments or devices used to record the data (e.g. pen and paper, computer, eye tracker, video or audio equipment) whether anyone was present besides the participant(s) and the researcher, and whether the researcher was blind to experimental condition and/or the study hypothesis during data collection. |
| Timing | Indicate the start and stop dates of data collection. If there is a gap between collection periods, state the dates for each sample cohort. |
| Data exclusions | If no data were excluded from the analyses, state so OR if data were excluded, provide the exact number of exclusions and the rationale behind them, indicating whether exclusion criteria were pre-established. |
| Non-participation | State how many participants dropped out/declined participation and the reason(s) given OR provide response rate OR state that no participants dropped out/declined participation. |
| Randomization | If participants were not allocated into experimental groups, state so OR describe how participants were allocated to groups, and if allocation was not random, describe how covariates were controlled. |

# Ecological, evolutionary & environmental sciences study design

All studies must disclose on these points even when the disclosure is negative.

| | |
|---|---|
| Study description | *Briefly describe the study. For quantitative data include treatment factors and interactions, design structure (e.g. factorial, nested, hierarchical), nature and number of experimental units and replicates.* |
| Research sample | *Describe the research sample (e.g. a group of tagged Passer domesticus, all Stenocereus thurberi within Organ Pipe Cactus National Monument), and provide a rationale for the sample choice. When relevant, describe the organism taxa, source, sex, age range and any manipulations. State what population the sample is meant to represent when applicable. For studies involving existing datasets, describe the data and its source.* |
| Sampling strategy | *Note the sampling procedure. Describe the statistical methods that were used to predetermine sample size OR if no sample-size calculation was performed, describe how sample sizes were chosen and provide a rationale for why these sample sizes are sufficient.* |
| Data collection | *Describe the data collection procedure, including who recorded the data and how.* |
| Timing and spatial scale | *Indicate the start and stop dates of data collection, noting the frequency and periodicity of sampling and providing a rationale for these choices. If there is a gap between collection periods, state the dates for each sample cohort. Specify the spatial scale from which the data are taken* |
| Data exclusions | *If no data were excluded from the analyses, state so OR if data were excluded, describe the exclusions and the rationale behind them, indicating whether exclusion criteria were pre-established.* |
| Reproducibility | *Describe the measures taken to verify the reproducibility of experimental findings. For each experiment, note whether any attempts to repeat the experiment failed OR state that all attempts to repeat the experiment were successful.* |
| Randomization | *Describe how samples/organisms/participants were allocated into groups. If allocation was not random, describe how covariates were controlled. If this is not relevant to your study, explain why.* |
| Blinding | *Describe the extent of blinding used during data acquisition and analysis. If blinding was not possible, describe why OR explain why blinding was not relevant to your study.* |

Did the study involve field work?   ☐ Yes   ☐ No

## Field work, collection and transport

| | |
|---|---|
| Field conditions | *Describe the study conditions for field work, providing relevant parameters (e.g. temperature, rainfall).* |
| Location | *State the location of the sampling or experiment, providing relevant parameters (e.g. latitude and longitude, elevation, water depth).* |
| Access & import/export | *Describe the efforts you have made to access habitats and to collect and import/export your samples in a responsible manner and in compliance with local, national and international laws, noting any permits that were obtained (give the name of the issuing authority, the date of issue, and any identifying information).* |
| Disturbance | *Describe any disturbance caused by the study and how it was minimized.* |

# Reporting for specific materials, systems and methods

We require information from authors about some types of materials, experimental systems and methods used in many studies. Here, indicate whether each material, system or method listed is relevant to your study. If you are not sure if a list item applies to your research, read the appropriate section before selecting a response.

## Materials & experimental systems

| n/a | Involved in the study |
|---|---|
| ☐ | ☒ Antibodies |
| ☐ | ☒ Eukaryotic cell lines |
| ☒ | ☐ Palaeontology and archaeology |
| ☒ | ☐ Animals and other organisms |
| ☒ | ☐ Human research participants |
| ☒ | ☐ Clinical data |
| ☒ | ☐ Dual use research of concern |

## Methods

| n/a | Involved in the study |
|---|---|
| ☒ | ☐ ChIP-seq |
| ☐ | ☒ Flow cytometry |
| ☒ | ☐ MRI-based neuroimaging |

## Antibodies

| | |
|---|---|
| Antibodies used | anti-ACTB (SantaCruz, sc69879, 1:4000), anti-HSPD1 (Abcam, ab46798, 1:2000), anti-COX5B (Proteintech, 11418-2-AP, 1:1000), anti-HSF1 (Cell Signaling, #4356, 1:1000), anti-HSF1 (Abcam, ab2923, 1:10000), anti-DNAJA1 (Proteintech, 11713-1-AP, 1:2000), anti-Hsp70 (Proteintech, 10995-1-AP, 1:2000), anti-NRF1 (Cell Signaling (D9K6P), #46743, 1:1000), anti-α-tubulin (Cell Signaling (DM1A), #3873, 1:3000), anti-histone H3 (Active Motif, 39163, 1:5000), anti-FLAG (Sigma, F1804, 1:5000) , anti-CHOP (Thermo Fisher Scientific, MA1-250, 1:1000), anti-ATF4 (Cell Signaling, #11815, 1:1000), anti-PITRM1 (Novus, H00010531-M03, 1:500), anti-LONP1 (Proteintech, 15440-1-AP, 1:2000), anti-cleaved PARP1 (Cell Signaling, #5625, 1:2000), anti-Caspase3 (Cell Signaling, #9661, 1:1000). Secondary antibodies used were anti-Rabbit IgG (H+L) HRP Conjugate (Promega, #W4021, 1:10000), IRDye 800CW goat anti-rabbit IgG (H + L) (Li-Cor, #926-32211, 1:15000), IRDye 680RD donkey anti-mouse IgG (H + L) (Li-Cor, #926-68072, 1:15000). |
| Validation | Antibodies for DNAJA1 (Fig.2c), HSF1 (Fig.3e, Extended Data Fig.8a), ATF4 (Extended Data Fig.1f), NRF1 (Extended Data Fig.6a), CHOP (Extended Data Fig.1e), LONP1 (Fig.4m), and PITRM1 (Fig.4m) were validated in this study with knockdown and/or knockout cells. Antibodies for HSPD1, ACTB and COX5B were used in previous publications (Michaelis Nat. Commun. 2022, Schäfer Mol. Cell 2021) and in the company's websites here<br>https://www.abcam.com/products/primary-antibodies/hsp60-antibody-ab46798.html<br>https://www.scbt.com/p/beta-actin-antibody-ac-15<br>https://www.ptglab.com/products/COX5B-Antibody-11418-2-AP.htm<br>Validation for antibodies of Hsp70, tubulin, histone H3, FLAG, cleaved PARP1, cleaved Caspase3 can be found in the company's websites here<br>https://www.ptglab.com/products/HSPA1A-Antibody-10995-1-AP.htm<br>https://www.cellsignal.de/products/primary-antibodies/a-tubulin-dm1a-mouse-mab/3873<br>https://www.activemotif.com/documents/tds/39163.pdf<br>https://www.sigmaaldrich.com/DE/de/product/sigma/f1804<br>https://www.cellsignal.de/products/primary-antibodies/cleaved-parp-asp214-d64e10-xp-rabbit-mab/5625<br>https://www.cellsignal.de/products/primary-antibodies/cleaved-caspase-3-asp175-antibody/9661?site-search-type=Products&N=4294956287&Ntt=9661s&fromPage=plp&_requestid=11646358 |

## Eukaryotic cell lines

Policy information about cell lines

| | |
|---|---|
| Cell line source(s) | HeLa cells were kindly provided by the Müller laboratory (obtained from ATCC) |
| Authentication | HeLa cells were authenticated by the source, and the investigators did not further authenticate the cells |
| Mycoplasma contamination | HeLa cells were tested negative for mycoplasma |
| Commonly misidentified lines<br>(See ICLAC register) | No commonly misidentified cell lines were used in this study |

## Palaeontology and Archaeology

| | |
|---|---|
| Specimen provenance | *Provide provenance information for specimens and describe permits that were obtained for the work (including the name of the issuing authority, the date of issue, and any identifying information). Permits should encompass collection and, where applicable, export.* |
| Specimen deposition | *Indicate where the specimens have been deposited to permit free access by other researchers.* |
| Dating methods | *If new dates are provided, describe how they were obtained (e.g. collection, storage, sample pretreatment and measurement), where they were obtained (i.e. lab name), the calibration program and the protocol for quality assurance OR state that no new dates are provided.* |

☐ Tick this box to confirm that the raw and calibrated dates are available in the paper or in Supplementary Information.

| | |
|---|---|
| Ethics oversight | *Identify the organization(s) that approved or provided guidance on the study protocol, OR state that no ethical approval or guidance was required and explain why not.* |

Note that full information on the approval of the study protocol must also be provided in the manuscript.

## Animals and other organisms

Policy information about studies involving animals; ARRIVE guidelines recommended for reporting animal research

| | |
|---|---|
| Laboratory animals | *For laboratory animals, report species, strain, sex and age OR state that the study did not involve laboratory animals.* |
| Wild animals | *Provide details on animals observed in or captured in the field; report species, sex and age where possible. Describe how animals were caught and transported and what happened to captive animals after the study (if killed, explain why and describe method; if released, say where and when) OR state that the study did not involve wild animals.* |
| Field-collected samples | *For laboratory work with field-collected samples, describe all relevant parameters such as housing, maintenance, temperature, photoperiod and end-of-experiment protocol OR state that the study did not involve samples collected from the field.* |

Ethics oversight

*Identify the organization(s) that approved or provided guidance on the study protocol, OR state that no ethical approval or guidance was required and explain why not.*

Note that full information on the approval of the study protocol must also be provided in the manuscript.

# Human research participants

Policy information about studies involving human research participants

Population characteristics

*Describe the covariate-relevant population characteristics of the human research participants (e.g. age, gender, genotypic information, past and current diagnosis and treatment categories). If you filled out the behavioural & social sciences study design questions and have nothing to add here, write "See above."*

Recruitment

*Describe how participants were recruited. Outline any potential self-selection bias or other biases that may be present and how these are likely to impact results.*

Ethics oversight

*Identify the organization(s) that approved the study protocol.*

Note that full information on the approval of the study protocol must also be provided in the manuscript.

# Clinical data

Policy information about clinical studies

All manuscripts should comply with the ICMJE guidelines for publication of clinical research and a completed CONSORT checklist must be included with all submissions.

Clinical trial registration

*Provide the trial registration number from ClinicalTrials.gov or an equivalent agency.*

Study protocol

*Note where the full trial protocol can be accessed OR if not available, explain why.*

Data collection

*Describe the settings and locales of data collection, noting the time periods of recruitment and data collection.*

Outcomes

*Describe how you pre-defined primary and secondary outcome measures and how you assessed these measures.*

# Dual use research of concern

Policy information about dual use research of concern

## Hazards

Could the accidental, deliberate or reckless misuse of agents or technologies generated in the work, or the application of information presented in the manuscript, pose a threat to:

No | Yes

☐ ☐ Public health

☐ ☐ National security

☐ ☐ Crops and/or livestock

☐ ☐ Ecosystems

☐ ☐ Any other significant area

## Experiments of concern

Does the work involve any of these experiments of concern:

No | Yes

☐ ☐ Demonstrate how to render a vaccine ineffective

☐ ☐ Confer resistance to therapeutically useful antibiotics or antiviral agents

☐ ☐ Enhance the virulence of a pathogen or render a nonpathogen virulent

☐ ☐ Increase transmissibility of a pathogen

☐ ☐ Alter the host range of a pathogen

☐ ☐ Enable evasion of diagnostic/detection modalities

☐ ☐ Enable the weaponization of a biological agent or toxin

☐ ☐ Any other potentially harmful combination of experiments and agents

# ChIP-seq

## Data deposition

☐ Confirm that both raw and final processed data have been deposited in a public database such as GEO.

☐ Confirm that you have deposited or provided access to graph files (e.g. BED files) for the called peaks.

| | |
|---|---|
| Data access links *May remain private before publication.* | *For "Initial submission" or "Revised version" documents, provide reviewer access links. For your "Final submission" document, provide a link to the deposited data.* |
| Files in database submission | *Provide a list of all files available in the database submission.* |
| Genome browser session (e.g. UCSC) | *Provide a link to an anonymized genome browser session for "Initial submission" and "Revised version" documents only, to enable peer review. Write "no longer applicable" for "Final submission" documents.* |

## Methodology

| | |
|---|---|
| Replicates | *Describe the experimental replicates, specifying number, type and replicate agreement.* |
| Sequencing depth | *Describe the sequencing depth for each experiment, providing the total number of reads, uniquely mapped reads, length of reads and whether they were paired- or single-end.* |
| Antibodies | *Describe the antibodies used for the ChIP-seq experiments; as applicable, provide supplier name, catalog number, clone name, and lot number.* |
| Peak calling parameters | *Specify the command line program and parameters used for read mapping and peak calling, including the ChIP, control and index files used.* |
| Data quality | *Describe the methods used to ensure data quality in full detail, including how many peaks are at FDR 5% and above 5-fold enrichment.* |
| Software | *Describe the software used to collect and analyze the ChIP-seq data. For custom code that has been deposited into a community repository, provide accession details.* |

# Flow Cytometry

## Plots

Confirm that:

☒ The axis labels state the marker and fluorochrome used (e.g. CD4-FITC).

☒ The axis scales are clearly visible. Include numbers along axes only for bottom left plot of group (a 'group' is an analysis of identical markers).

☒ All plots are contour plots with outliers or pseudocolor plots.

☒ A numerical value for number of cells or percentage (with statistics) is provided.

## Methodology

| | |
|---|---|
| Sample preparation | Cells were cultured and treated as described in the Method section prior to the measurements. Cells were harvested via trypsinisation and further processed for independent FACS measurement. MitoSOX Red (Thermo Fisher Scientific) was used to measure mitochondrial ROS production according to manufacturer's instructions. Cell deaths were measured with combination of Annexin V conjugated to Alexa Fluor 488 (ThermoFisher Scientific) and propidium iodide (Thermo Fisher Scientific) according to the manufacturer's instructions. Fluorescence-activated cell sorting was performed on FACSCanto II and FACSymphony A5 flow cytometry systems (BD) for MitoSOX and Annexin V measurements, respectively. Analysis of FACS data was performed with FlowJo v.10 software. |
| Instrument | FACS was performed on FACSCanto II and FACSymphony A5 flow cytometry systems (BD) |
| Software | Analysis of FACS data was performed with FlowJo v.10 software |
| Cell population abundance | For FACS analysis 10000 events were counted. No sample sorting was performed in this study. |
| Gating strategy | Cells were gated on FSC-A/SSC-A to exclude debris before performing the quantification. Gating for cell death assay with Annexin V is shown in Extended Data, and was set based on DMSO and Straurosporine as negative and positive controls, respectively. |

☒ Tick this box to confirm that a figure exemplifying the gating strategy is provided in the Supplementary Information.

# Magnetic resonance imaging

## Experimental design

| | |
|---|---|
| Design type | *Indicate task or resting state; event-related or block design.* |
| Design specifications | *Specify the number of blocks, trials or experimental units per session and/or subject, and specify the length of each trial or block (if trials are blocked) and interval between trials.* |
| Behavioral performance measures | *State number and/or type of variables recorded (e.g. correct button press, response time) and what statistics were used to establish that the subjects were performing the task as expected (e.g. mean, range, and/or standard deviation across subjects).* |

## Acquisition

| | |
|---|---|
| Imaging type(s) | *Specify: functional, structural, diffusion, perfusion.* |
| Field strength | *Specify in Tesla* |
| Sequence & imaging parameters | *Specify the pulse sequence type (gradient echo, spin echo, etc.), imaging type (EPI, spiral, etc.), field of view, matrix size, slice thickness, orientation and TE/TR/flip angle.* |
| Area of acquisition | *State whether a whole brain scan was used OR define the area of acquisition, describing how the region was determined.* |

Diffusion MRI ☐ Used ☐ Not used

## Preprocessing

| | |
|---|---|
| Preprocessing software | *Provide detail on software version and revision number and on specific parameters (model/functions, brain extraction, segmentation, smoothing kernel size, etc.).* |
| Normalization | *If data were normalized/standardized, describe the approach(es): specify linear or non-linear and define image types used for transformation OR indicate that data were not normalized and explain rationale for lack of normalization.* |
| Normalization template | *Describe the template used for normalization/transformation, specifying subject space or group standardized space (e.g. original Talairach, MNI305, ICBM152) OR indicate that the data were not normalized.* |
| Noise and artifact removal | *Describe your procedure(s) for artifact and structured noise removal, specifying motion parameters, tissue signals and physiological signals (heart rate, respiration).* |
| Volume censoring | *Define your software and/or method and criteria for volume censoring, and state the extent of such censoring.* |

## Statistical modeling & inference

| | |
|---|---|
| Model type and settings | *Specify type (mass univariate, multivariate, RSA, predictive, etc.) and describe essential details of the model at the first and second levels (e.g. fixed, random or mixed effects; drift or auto-correlation).* |
| Effect(s) tested | *Define precise effect in terms of the task or stimulus conditions instead of psychological concepts and indicate whether ANOVA or factorial designs were used.* |

Specify type of analysis: ☐ Whole brain ☐ ROI-based ☐ Both

| | |
|---|---|
| Statistic type for inference<br>(See Eklund et al. 2016) | *Specify voxel-wise or cluster-wise and report all relevant parameters for cluster-wise methods.* |
| Correction | *Describe the type of correction and how it is obtained for multiple comparisons (e.g. FWE, FDR, permutation or Monte Carlo).* |

## Models & analysis

| n/a | Involved in the study | |
|---|---|---|
| ☐ | ☐ | Functional and/or effective connectivity |
| ☐ | ☐ | Graph analysis |
| ☐ | ☐ | Multivariate modeling or predictive analysis |

| | |
|---|---|
| Functional and/or effective connectivity | *Report the measures of dependence used and the model details (e.g. Pearson correlation, partial correlation, mutual information).* |
| Graph analysis | *Report the dependent variable and connectivity measure, specifying weighted graph or binarized graph, subject- or group-level, and the global and/or node summaries used (e.g. clustering coefficient, efficiency, etc.).* |

Multivariate modeling and predictive analysis

*Specify independent variables, features extraction and dimension reduction, model, training and evaluation metrics.*

