## [Peer Review File · Nature]

Manuscript Title: A cytosolic surveillance mechanism activates the mitochondrial UPR

Reviewer Comments & Author Rebuttals

Reviewer Reports on the Initial Version:

Referees' comments:

Referee #1 (Remarks to the Author):

In this submission, Sutandy et al. strive to characterize the specific signals that induce the mitochondrial unfolded protein response (UPR_{mt}), a long-standing question in the field of mitochondrial protein stress signaling. This work looks to build upon previous studies by other groups that have identified ATF5 (Fiorese et al. 2016) and HSF1 (Katiyar et al., 2020) as requisite for this response. The current work makes multiple claims:

- 1.) The early induction of the UPR_{mt} requires both mitochondrially produced ROS and mitochondrial proteins that are misfolded and aggregated in the cytosol.
- 2.) The signal requires an oxidized DNAJA1 protein to recognize these misfolded mitochondrial proteins and bring them to the cytosolic HSP70.
- 3.) The recruitment of cytosolic HSP70 by DNAJA1 disrupts its interaction with HSF1.
- 4.) Freed HSF1 localizes to the nucleus where it can drive expression of mitochondrial chaperones and induce the UPR_{mt}.

While some of the conclusions in the paper are strong, particularly the requirement for DNAJA1 to induce mitochondrial chaperone expression, there are multiple issues with some of the other conclusions based on the experiments shown.

- 1.) Although the group was able to show a statistically significant increase in the amount of hydrogen peroxide in the cell—particularly in the “Cyto+Nuc” compartment—under GTPP treatment, the effect size is very small (the ratio changes from ~0.7 to 0.9 between 0 and 3 hours). This almost negligible change in the relevant cellular compartments where DNAJA1 resides is a weak foundation on which to build their model. Is this small change sufficient to cause such a targeted and drastic effect on cytosolic DNAJA1? No other controls were included to show a similar increase under antimycin A conditions or a decrease under rotenone conditions, which would be expected given the results in Fig 1F. Similarly, no controls were included to show that NAC treatment, which prevents the induction of the UPR_{mt}, results in the decrease of hydrogen peroxide in the cell.
- 2.) Only one method of cellular perturbation (chemical) was used to induce the UPR_{mt}; such chemical perturbations can be harsh and potentially non-specific. This chemical includes a triphenylphosphonium group, which itself exhibits some mitochondrial toxicity and effect on mitochondrial complexes (Kafkova et al, bioRxiv 2021). In order to ensure the specificity of the signal, multiple molecular biological approaches should have been taken to see a similar response. Previously, other groups have used the expression of misfolded proteins, such as OTC, in the cytosol and mitochondria for this purpose.
- 3.) The group claims that insoluble proteins in the cytosol are required to signal induction and that these are unaffected by NAC treatment. However, they only look at insoluble proteins in a mitochondrial fraction and not in a whole cell fraction containing cytosol, making it impossible to draw that conclusion.
- 4.) The group states that “ATF5 was not required to activate the UPR_{mt}” based on the result that it was not induced upon treatment with GTPP under DNAJA1 or HSF1 knockdown cells.” However, claiming that it is not required for induction would necessitate that the protein be knocked out or knocked down similar to previous work (Fiorese et al. 2016).

5.) DNAJA1 knock down fully abolished induction of the UPRmt when looking at chaperone transcript abundance, however, nuclear localization of HSF1 was only inhibited by approximately 50%. This leaves an appreciable amount in the nucleus to drive gene expression, yet this does not occur and is not addressed.

Given these and other issues with the manuscript, I cannot recommend publication in Nature. Although some of the conclusions seem sound, these by themselves do not represent a major advancement in the field. Other key conclusions are likely based on overinterpretation of the data and therefore may likely be incorrect. Any reconsideration of the manuscript would require substantial additions and revisions.

Referee #2 (Remarks to the Author):

Summary of key results - Mitochondrial protein misfolding induces mitochondrial unfolded protein response (UPRmt) that retrogradely upregulates nuclear genes encoding mitochondrial chaperones and proteases. This promotes the restoration of mitochondrial homeostasis under stress conditions. UPRmt has been previously reported in cultured mammalian cells, although the activation of the mitochondrial proteostatic genes is generally moderate. In the current manuscript, the authors determined the mechanism by which mitochondrial misfolding stress (MMS) reprograms nuclear transcription and induces UPRmt. In most experiments, gamitrinib-triphenylphosphonium (GTPP) was used to inhibit the mitochondrial HSP90 and to induce UPRmt, with the goal of identifying the molecular components mediating the activation of UPRmt genes. The key findings include: (1) GTPP-induced activation of mitochondrial chaperone genes is ablated by ROS scavengers; (2) Using the H₂O₂ sensor HyPer, the authors reported that GTPP induces ROS production in the intermembrane space, which is responsible for UPRmt activation; (3) Redox proteomic analysis showed that GTPP induces the oxidation of the cytosolic co-chaperone HSP40 (DNAJA1); (4) DNAJA1 depletion prevents GTPP-induced activation of mitochondrial chaperone genes; (5) GTPP treatment induces the accumulation of mitochondrial proteins in the cytosol (c-mtProt) that bind to DNAJA1 and HSP70; (6) Inhibition of global protein synthesis by cycloheximide and mitochondrial biogenesis by NRF1 depletion prevent the activation of mitochondrial chaperone genes by GTPP; (7) GTPP treatment stimulates the nuclear translocation of the HSF1 transcriptional factor that is known to regulate the mitochondrial chaperone genes; (8) GTPP treatment decreases HSP70-HSF1 interaction, releasing HSF1 that promotes nuclear translocation; (9) when antimycin and oligomycin are used to induce ROS production and to collapse membrane potential (and therefore protein import), the transcription of mitochondrial chaperone genes is also activated like does GTPP. Based on these data, the authors concluded that mitochondrial misfolding stress increases ROS release from the intermembrane space and causes c-mtProt accumulation. ROS activate DNAJA1 and c-mtProts titrate out HSP70, which in turn releases HSF1 from HSP70 to stimulate its nuclear translocation and to activate mitochondrial chaperone genes.

Overall evaluation - The manuscript comprises of a large amount of data generated by different approaches at the transcriptomic and proteomic levels. The strengths of manuscript include strong data documenting the activation of DNAJA1-HSP70-HSP1 axis by GTPP-induced ROS production and c-mtProt accumulation. However, the reviewer finds that its relevance to UPRmt remains uncertain and that the conclusion of the manuscript is weakly supported by the data. Another major concern is that the experimental readouts reporting UPRmt are rather moderate throughout the manuscript. This further weakens the conclusion and limits the potential impact of the study in the field especially when translating into in vivo systems.

Major concerns -

(1) The overall validity of using the mitochondrial HSP90 inhibitor gamitrinib-triphenylphosphonium (GTPP) as a model to induce mitochondrial misfolding stress (MMS) in the

context of mitochondrial proteostatic signaling is questionable. This is because the concentrations of GTPP that differentially inhibits mitochondrial and cytosolic HSP90 are very narrow (<https://www.jci.org/articles/view/44855>). Although GTPP is preferentially targeted to mitochondria, it also inhibits cytosolic HSp90. It is difficult to exclude the possibility that a small fraction remains in the cytosol to induce some levels of proteostatic stress and to reduce mitochondrial protein targeting (HSP90 is required for mitochondrial protein targeting), which subsequently induces the stress response with HSF1 being translocated to the nuclear to activate mitochondrial chaperone genes. If this were the case, the stress responses detected by the authors may result from cytosolic rather than mitochondrial stress. This is particularly relevant when considering the strong activation of DDIT3 (see Extended Data Figure 1d) by GTPP. DDIT3 (or CHOP) is known to be also activated by ER stress. Inhibition of cytosolic Hsp90 is known to induce ER stress. Moreover, RNA-seq experiment clearly showed that ER stress is activated by GTPP (Extended Data Figure 2c). Thus, GTPP may induce the nuclear translocation of HSF1 by cytosol-originated stress rather than MMS. The authors also used antimycin + oligomycin to stress mitochondria. However, these conditions are expected to collapse mitochondrial membrane potential and completely shut down mitochondrial biogenesis. I do not think that this is a good model for studying UPRmt. UPRmt is expected to stimulate the import of mitochondrial chaperones to restore mitochondrial homeostasis. Gene activation seen upon the shut-down of mitochondrial biogenesis may be just a secondary response to global cellular damage instead of being selective for restoring mitochondrial homeostasis.

(2) Throughout the manuscript, the activation of the mitochondrial chaperone genes by GTPP is rather moderate. Furthermore, the authors did not show whether the mitochondrial chaperones are actually increased inside mitochondria under the stress conditions. Possibilities exist that these genes are just collaterally activated by the HSF1 network. In their previous and current (Extended Data Figure 1c) studies, the authors clearly showed that OXPHOS genes are also activated by GTPP. There is no robust data supporting the idea that the mitochondrial chaperone genes are selectively activated with their gene products trafficked to mitochondria to improve mitochondrial homeostasis. Without addressing this issue, the overall significance of the data for UPRmt is questionable. In Extended Data Figure 1, the data in 1b and 1c lack statistical justification. Especially, the data are processed in different ways, with biological variability presented in 1c but not in 1b.

(3) The reviewer is not convinced for a role of ROS produced in the intermembrane space in activating UPRmt. First, the authors found that GTPP treatment induces mitochondrial proteostatic genes, which is ablated by antioxidants (Figure 2c and 2d). GTPP treatment is known to elevate ROS production. However, reduction of HSPD1, HSPE1 and HSPA9 induction by the antioxidants is rather marginal (Figure 1d). This may just reflect less mitochondrial damage in cells cotreated with antioxidants, which would attenuate stress signaling including UPRmt. Although the authors did not find changes to protein aggregation levels with or without antioxidant treatment, other damage may still play a role in signaling UPRmt (e.g., lipid peroxidation, mtDNA damage, etc.) which are protected by antioxidants. If ROS signals UPRmt, treatment with antimycin and rotenone should be able to activate the genes without GTPP. This is clearly not the case (Figure 1f). Secondly, the authors used H₂O₂ sensors targeted to different mitochondrial sub-compartments to decipher where ROS are produced to activate UPRmt. They used inhibitors of VDAC on the outer membrane and mPTP on the inner membrane to reach the conclusion that diffusion of ROS from IMS to cytosol is responsible for UPRmt activation. The reviewer finds that the rationale is weak for the experiments. The authors should make it clear that HyPer7 is a direct sensor for H₂O₂, instead of superoxide. Peroxide can diffuse through the membranes, in contrast to the negatively charged superoxide that cannot cross the membrane freely. In other words, treatment by VDAC and mPTP inhibitors may or may not affect the distribution of H₂O₂. As such, the conclusion that IMS-generated ROS play a role in activating UPRmt is weak.

(4) The data supporting the conclusion that "combined accumulation of ROS and c-mtProt signal the UPRmt" (Fig. 4) are misleading. Oligomycin does not block protein import. It does so only

when it is combined with antimycin. In these experiments, antimycin is also used as a ROS inducer. If antimycin alone does not induce UPRmt, doesn't it mean that ROS have no role? I do not think that the data allow to make a conclusion that a combined action is required for UPRmt induction. Furthermore, the quality of the data supporting a combined role of ROS accumulation and cytosolic-mtProt (c-mtProt) in inducing UPRmt is poor. In Figure 4e, the z-score map is not that informative for evaluating the robustness of gene activation.

Other points:

1. The data supporting a differential activation of UPRmt and ISR upon GTPP treatment are weak because it does not have statistical validation (Extended data Figure 1b and 1d).

2. Extended Data Figure 1 – the authors stated that “the primary ISR factor ATF4 was induced before the UPRmt...”. There is no statistical analysis to support the conclusion. Despite that the data are strong showing that activation of the proteostatic genes is independent of CHOP and ATF4, other transcriptional factors in the ISR pathway were not examined. The statement that “ISR was not required for UPRmt induction” is inaccurate.

3. The specific oxidation of DNAJA1 by GTPP-induced oxidative stress is an interesting observation. The authors showed that activation of proteostatic genes by GTPP is abolished in cells with KD of DNAJA1 (Figure 2c). However, the author did not show whether DNAJA1 KD affects global proteostasis and causes cell death. Reduced UPRmt activation may just be a secondary effect of cell death, rather than a specific defect in UPRmt signaling. HSP40 and 70 are also required for mitochondrial protein import. It is also possible that DNAJA1 KD directly affects mitochondrial biogenesis, which in turn affects cell signaling. Thus, the conclusion that “DNAJA1 is an essential component of UPRmt signaling is premature.

4. The use of “orphan mitochondrial proteins” is unnecessary. It sounds like uncharacterized or evolutionarily un-conserved proteins. Why not just “unimported proteins” or “preproteins”?

5. Reference 21: author's names are wrong.

6. Extended Data Figure 7c: no annotation for the y-axis and no explanation for the experimental groups.

7. The authors referred the study being conducted “in humans” in many places throughout the manuscript. I think using “cultured human cells” is more appropriate.

8. The transcriptional data in Figure 4c-e lack statistical analysis.

9. It would be good if the authors specify the nature of experiments (e.g., qPCR, RNS-seq, etc) and the type of cells used in the text or in the figure legends, so that one does not have to check the Method section to find this out.

Referee #3 (Remarks to the Author):

This is an interesting paper that shows that mitoROS and accumulation of mitochondrial proteins in the cytosol induce a UPRmt signal. This signal depends on oxidation of cytosolic HSP40 protein DNAJA1 leading to HSP70 recruitment to mitochondria concomitant with release of HSF1 to nucleus. Subsequently, HSF1 increases gene expression. Much of the data depends on an artificial way of inducing ROS and c-mtProtein accumulation to activate this pathway. I very much enjoyed the conceptual framework in this paper. However, some of the experimental approaches are not

optimal.

Major points:

Figure 1:

(1) The use of NAC is not appropriate (see this PMID: 29429900). Please remove all NAC experiments. I would use Martin Brand's compounds that suppress superoxide from complex I (S1QEL1.1) and complex III (S3QEL 2)- See his excellent review PMID: 33148057. Other strategies could be to try mito-tempo.

(2) Antimycin sustains the response while rotenone diminishes the response. This suggests that complex III is main site of superoxide production. I would complement this with using myxothiazol (500nM) and Antimycin (500nM). Myxothiazol prevents complex III superoxide production while antimycin maintains complex III superoxide production.

(3) What happens to this HSF1 response if you give paraquat or menadione that will generate ROS?

(4) There is no physiological or pathology response here. All artificial ways to activate this system. Hypoxia is known to increase ROS and HSF1 responses. Also, doxycycline inhibits mitochondrial translation. Does doxycycline induce this pathway? It would be important to know the answers to these important interventions.

(5) "We induced ROS production in the IMS using antimycin A and triggered c-mtProt accumulation by blocking mitochondrial protein import with the ATP synthase inhibitor oligomycin." This is peculiar statement and results that accompanied this reasoning. Wade Harper and Richard Youle use combination of antimycin and oligomycin to depolarize mitochondrial membrane potential that will induce mitophagy and block protein import. But depolarized mitochondrial membrane potential decrease ROS as well (see Michael Murphy excellent review: PMID: 19061483). Not sure how to make sense of this experiment.

(6) Please use ISRIB for ISR inhibition. PMID: 30674674

Author Rebuttals to Initial Comments:

Remarks:

We would like to thank all the referees for the time spent and their comments and suggestions to improve the manuscript. Our responses are in blue font.

In addition to the referees' comments, we also asked two experts on redox biology – Jan Riemer (Cologne, expert on mitochondrial ROS) and Ralf Brandes (Frankfurt, expert on redox signalling) – to comment on the manuscript. We have included their and the referees' comments to strengthen the manuscript. Both experts fully endorsed the manuscript and are now mentioned in the acknowledgements.

Referees' comments:

Referee #1 (Remarks to the Author):

In this submission, Sutandy et al. strive to characterize the specific signals that induce the mitochondrial unfolded protein response (UPR^{mt}), a long-standing question in the field of mitochondrial protein stress signaling. This work looks to build upon previous studies by other groups that have identified ATF5 (Fiorese et al. 2016) and HSF1 (Katiyar et al., 2020) as requisite for this response. The current work makes multiple claims:

- 1.) The early induction of the UPR^{mt} requires both mitochondrially produced ROS and mitochondrial proteins that are misfolded and aggregated in the cytosol.
- 2.) The signal requires an oxidized DNAJA1 protein to recognize these misfolded mitochondrial proteins and bring them to the cytosolic HSP70.
- 3.) The recruitment of cytosolic HSP70 by DNAJA1 disrupts its interaction with HSF1.
- 4.) Freed HSF1 localizes to the nucleus where it can drive expression of mitochondrial chaperones and induce the UPR^{mt}.

While some of the conclusions in the paper are strong, particularly the requirement for DNAJA1 to induce mitochondrial chaperone expression, there are multiple issues with some of the other conclusions based on the experiments shown.

We thank the referee for recognizing the strengths of the manuscript and pointing out ways for further improvement. We would like to point out that, consistent with the main claims identified by the referee, our work does not build on the previous ATF5 study by the Haynes lab and only addresses the Haynes publication by studying whether ATF5 plays a role in the signalling pathway identified in this manuscript. The identification of HSF1 as an important UPR^{mt} factor by Katiyar was striking and surprising. We now explain how HSF1 can be activated specifically by a signalling pathway induced by mitochondrial protein misfolding, which act as a cytosolic surveillance mechanism to activate the UPR^{mt} transcriptional response in human cells.

Furthermore, referring to the first claim summarized by referee 1, we would like to clarify that we do not make any statements about the nature of the unimported mitochondrial proteins in the cytosol (c-mtProt). At the moment, it is not clear whether these proteins are misfolded or even aggregated. In our model, the main effect of unimported mitochondrial precursor proteins is the binding and sequestration of cytosolic chaperones to the precursors.

1.) Although the group was able to show a statistically significant increase in the amount of hydrogen peroxide in the cell—particularly in the “Cyto+Nuc” compartment—under GTPP treatment, the effect size is very small (the ratio changes from ~0.7 to 0.9 between 0 and 3 hours). This almost negligible change in the relevant cellular compartments where DNAJA1 resides is a weak foundation on which to build their model. Is this small change sufficient to cause such a targeted and drastic effect on cytosolic DNAJA1? No other controls were included to show a similar increase under antimycin A conditions or a decrease under rotenone conditions, which would be expected given the results in Fig 1F. Similarly, no controls were included to show that NAC treatment, which prevents the induction of the UPR_{mt}, results in the decrease of hydrogen peroxide in the cell.

We thank referee 1 for pointing out this important point. Throughout the manuscript, 10 μ M antimycin A was used to increase ROS production. This concentration is well-established to induce various cytosolic responses¹⁻³ and confirmed to increase mitochondrial ROS (superoxide O₂⁻) levels (**Extended Data Fig. 11a**). However, we found that the H₂O₂ amount produced with this treatment is lower than the detection limit of Hyper7 (**Rebuttal Fig. 1, Extended Data Fig. 4f**). Therefore, we included 20 μ M antimycin A. Both concentrations of antimycin A have been shown to increase ROS production and induce mitochondrial stress in various publications including⁴⁻⁶. Of note, the changes in Hyper7 reporter fluorescence measured upon treatment with 20 μ M antimycin A were in agreement with previously published data of the same treatment⁵. As an additional reference point, we used 1 mM H₂O₂, which is sufficient to induce oxidative stress⁷. We found that all three treatments – 10 μ M GTPP, 20 μ M antimycin A and 1 mM H₂O₂ – produced comparable levels of ROS in the cytosol, strongly suggesting that the H₂O₂ levels produced by GTPP are sufficient for ROS signalling (**Rebuttal Fig. 1**). H₂O₂ concentrations as low as 300 μ M are capable of oxidizing and changing the activity of DNAJA1 *in vitro*⁸. Together, our data support the proposed model that the ROS levels produced during GTPP treatments are capable of inducing ROS-mediated cellular signalling via DNAJA1 oxidation. We added new figures containing these experimental data to the revised manuscript (**Extended Data Fig. 4f**).

Rebuttal Figure 1. Barplots showing the mean of H₂O₂ levels measured with Hyper7 reporters targeted to the IMS, the matrix, and untargeted (cytosol+nucleus) upon 3 h GTPP and different treatments as references (*n* = 5 biological replicates, >100 cells were analysed for each replicate).

Next, we performed experiments to measure ROS production in mitochondria during co-treatments of GTPP with NAC and GSH to support **Fig. 1d**. We used MitoSOX to directly monitor the effect of the co-treatments to mitochondrial ROS (O₂^{•-}) production upon GTPP treatment. We saw that both NAC and GSH efficiently reduced mitochondrial ROS levels induced by GTPP treatment and these antioxidant activities correlated with the inhibition of UPR^{mt} induction under the same treatments (**Rebuttal Fig. 2a-d**, **Extended Data Fig. 3b-e**). In addition, employing a different antioxidant that works as superoxide dismutase mimetic, MnTBAP, resulted in the same observation (**Extended Data Fig. 3f-g**).

We also performed experiments to measure ROS production in mitochondria during co-treatments of GTPP with antimycin A and rotenone to support **Fig. 1f** (**Rebuttal Fig. 2e**, **Extended Data Fig. 3h**). We found that co-treatment with antimycin A boosted ROS production by GTPP. This effect correlated with the UPR^{mt} induction shown in **Fig. 1f** and supported that mitochondrial ROS levels could scale the strength of UPR^{mt} induction. We could not show the effect of co-treatment with rotenone, which was expected to inhibit the site-specific ROS production caused by GTPP. However, this result is not surprising because MitoSOX measures the total mitochondrial ROS production and therefore cannot distinguish ROS production by different sites within mitochondria. While rotenone treatment might inhibit the ROS production specific for GTPP, it is also known as a potent inducer of ROS production by complex I. Therefore, the ROS production measured in co-treatment of GTPP and rotenone represents net ROS production induced by both individual treatments and the combination. Intriguingly, our data showed that, although rotenone alone induced ROS production levels more than treatment with antimycin A, the combination of GTPP+rotenone induced lower ROS levels than GTPP+antimycin A, indicating that ROS production by GTPP was synergistic with antimycin A, but not with rotenone treatment. We added new figures containing these experimental data to the revised manuscript (**Extended Data Fig. 3b-h**).

Rebuttal Figure 2. a, b, Barplots showing (a) the mean of mitochondrial ROS ($O_2^{\cdot-}$) levels measured on FACS using MitoSOX and (b) relative transcript levels of indicated UPR^{mt} genes measured with qPCR of HeLa cells upon co-treatments of GTPP with titrated concentrations of NAC ($n = 3$ biological replicates). **c, d,** Barplots showing (c) the mean of mitochondrial ROS ($O_2^{\cdot-}$) levels measured on FACS using MitoSOX and (d) relative transcript levels of indicated UPR^{mt} genes measured with qPCR of HeLa cells upon co-treatments of GTPP with titrated concentrations of GSH ($n = 3$ biological replicates). **e,** Barplot showing the mean of mitochondrial ROS ($O_2^{\cdot-}$) levels of HeLa cells upon co-treatments of GTPP with antimycin A or rotenone measured on FACS using MitoSOX ($n = 3$ biological replicates). All P values are calculated with a two-tailed Student's t -test and represented as $*P \leq 0.05$, $**P \leq 0.01$, and $***P \leq 0.001$. All error bars represent mean+SD.

2.) Only one method of cellular perturbation (chemical) was used to induce the UPR^{mt}; such chemical perturbations can be harsh and potentially non-specific. This chemical includes a triphenylphosphonium group, which itself exhibits some mitochondrial toxicity and effect on mitochondrial complexes (Kafkova et al, bioRxiv 2021). In order to ensure the specificity of the signal, multiple molecular biological approaches should have been taken to see a similar response. Previously, other groups have used the expression of misfolded proteins, such as OTC, in the cytosol and mitochondria for this purpose.

We thank the referee for bringing up this important point. We would like to emphasize that while we indeed focused on the use of GTPP for the initial findings, the results were also validated with CDDO, another established compound that induces mitochondrial protein

misfolding by an alternative mechanism⁹. Once we had identified the UPR^{mt} signalling mechanism, we were also able to induce UPR^{mt} signalling by artificially producing ROS and precursor proteins (antimycin A and oligomycin A). We now also added another mitochondrial protease inhibitor for HTRA2, Ucf-101 (**Fig. 4c-j**), to the revised manuscript, offering a fourth approach to evaluate how our findings can be applied to UPR^{mt} induction by different ways to challenge mitochondrial proteostasis.

It has been a longstanding issue in the field of stress responses that signalling events occur rapidly and are in a different timescale from effects observed upon knock-down of protein quality control components or overexpression of misfolding-prone proteins. As a result, thapsigargin and tunicamycin are still by far the most used approaches (both are chemical) to monitor UPR^{ER} signalling, despite their well-known off targets. One critical feature is to control the length of treatment. We and others have shown that at the indicated time point and concentration (10 mM GTPP for 6 hours), mitochondrial biology is not perturbed^{10,11}. It is important to stay within these early time points to be able to follow the signalling mechanism. Extended treatments, similar to the ones in Kafkova *et al.* (mainly 24 hours)¹², lead to additional perturbation and compensatory events.

We and many other laboratories have struggled to use OTC as a protein that misfolds in mitochondria¹³ (*as quoted from this publication by the Auwerx lab “To unravel if these differences were based on the type of stress, we tried to reproduce the original conditions used to trigger the UPR^{mt} in mammalian cells by transfecting a mutant form of ornithine transcarbamylase (Δ OTC), which was reported to induce protein misfolding in mitochondria. However, we could not detect the activation of the UPR^{mt}, as previously reported.”*). A number of publications have also shown that highly misfolding-prone proteins from the cytosol, such as poly-Q extended Huntingtin, do not aggregate in mitochondria¹⁴. Following the referee’s suggestion, we started another effort to test a range of candidate proteins, including Abeta, alpha-synuclein, Δ OTC, Huntingtin with a long poly-Q expansion (Htt-Q97) and TDP-43, for their aggregation propensity in mitochondria. We fused these aggregation prone proteins with a mitochondrial targeting sequence and transiently overexpressed them for 24 h in HeLa cells. We could confirm the localization of all aggregation prone proteins to the mitochondria, except for TDP-43, which largely localized to the nucleus (**Rebuttal Fig. 3a**). However, we could only observe induction of the UPR^{mt} in cells transfected with Abeta. Abeta overexpression in mitochondria only caused a mild UPR^{mt} activation, as expected due to the long time-range of genetic models. Mitochondrial Abeta-induced UPR^{mt} was dependent on HSF1 and DNAJA1 (**Rebuttal Fig. 3b, c, Fig. 4k-l, Extended Data Fig. 10a, b**).

Rebuttal Figure 3. a, Representative microscopy images of HeLa cells transiently expressing MTS-GFP-tagged aggregation prone proteins (green). Mitochondria were stained with mitotracker (magenta). **b, c,** Heat maps of relative transcript levels of indicated UPR^{mt} genes in HeLa cells transiently expressing MTS-Abeta-GFP for 24h under siControl, **(b)** *DNAJA1* and **(c)** *HSF1* knock-down background measured with qPCR ($n = 4$ biological replicates). All P values are calculated with a two-tailed Student's t -test and represented as $*P \leq 0.05$, $**P \leq 0.01$, and $***P \leq 0.001$.

In addition, we have managed to induce the UPR^{mt} genetically by simultaneous depletion of two mitochondrial proteases, LONP1 and PITRM1. LONP1 is assumed to be the main mitochondrial protease degrading misfolding matrix proteins and PITRM1 degrades mitochondrial presequences¹⁵. Depletion of these proteases had previously been used to induce UPR^{mt}^{4,16,17}. Strikingly, we found that this genetic induction of UPR^{mt} also depends on both *DNAJA1* and *HSF1* (**Rebuttal Fig. 4, Fig. 4m-o**).

Rebuttal Figure 4. a, Western blot image of siControl and *LONP1+PITRM1* double knock-down for 8 days in HeLa cells. **b, c,** Heat maps of relative transcript levels of indicated UPR^{mt} genes in (b) *DNAJA1* and (c) *HSF1* knock-down under the *LONP1+PITRM1* double knock-down (dKD) background in HeLa cells measured with qPCR ($n = 4$ and 3 biological replicates for *DNAJA1* and *HSF1* knock-down, respectively). All P values are calculated with a two-tailed Student's t-test and represented as $*P \leq 0.05$, $** P \leq 0.01$, and $***P \leq 0.001$.

Together, we have shown that our findings on cytosolic surveillance mechanism to activate the UPR^{mt} can be observed across different means to induce the UPR^{mt} in human cells. This underlines the importance and relevance of these findings for the understanding of UPR^{mt} signalling, especially in human cells. We added new figures containing these experimental data to the revised manuscript (**Fig. 4c-o, Extended Data Fig. 10a, b**).

3.) The group claims that insoluble proteins in the cytosol are required to signal induction and that these are unaffected by NAC treatment. However, they only look at insoluble proteins in a mitochondrial fraction and not in a whole cell fraction containing cytosol, making it impossible to draw that conclusion.

We would like to clarify that, in our model, we proposed that the accumulation of mitochondrial precursor proteins (c-mtProt) is required for UPR^{mt} induction. We did not identify or describe a specific role of insoluble proteins in the cytosol. We apologize for not making this sufficiently clear in the original manuscript.

4.) The group states that “ATF5 was not required to activate the UPR^{mt}” based on the result that it was not induced upon treatment with GTPP under *DNAJA1* or *HSF1* knockdown cells.” However, claiming that it is not required for induction would necessitate that the protein be knocked out or knocked down similar to previous work (Fiorese et al. 2016).

We thank the referee for this comment and, accordingly, performed depletion experiments of *ATF5* both via knock-down and knock-out approaches. As shown in **Rebuttal Fig. 5**, we found that depletion of *ATF5* did not reduce UPR^{mt} induction by GTPP treatment. These data further confirm that *ATF5* is not required for activation of the UPR^{mt}. We added new figures containing these experimental data to the revised manuscript (**Rebuttal Fig. 5a-e, Extended Data Fig. 9a-e**).

Rebuttal Figure 5. **a, b**, Barplots depicting the mean of relative transcript levels of **(a)** *ATF5* and **(b)** several UPR^{mt} markers in HeLa *ATF5* knock-down cells upon treatment with GTPP in comparison to siControl measured with qPCR ($n = 3$ biological replicates). **c**, Sequencing map showing the location of deleted sequences on HeLa *ATF5* knock-out cells confirmed by Sanger sequencing. **d, e**, Barplots depicting the mean of relative transcript levels of **(d)** *ATF5* and **(e)** several UPR^{mt} markers in HeLa *ATF5* knock-out cells upon treatment with GTPP in comparison to WT measured with qPCR ($n = 3$ biological replicates).

ATF5 is lowly expressed in most tissues and cell lines¹⁸, therefore we argue that transcriptional activation of *ATF5* would be necessary for its functional activity (**Rebuttal Fig. 6a, b**). Indeed, we found that there was a DNAJA1- and HSF1-dependent *ATF5* induction upon the UPR^{mt} (**Fig. 4a, b**).

Rebuttal Figure 6. a, b Transcript levels of *ATF5* across different **(a)** cell lines and **(b)** cell types curated from the Human Protein Atlas. Transcript levels are represented as the consensus normalised expression (“nTPM”) as described in the Human Protein Atlas.

5.) *DNAJA1* knock down fully abolished induction of the UPRmt when looking at chaperone transcript abundance, however, nuclear localization of HSF1 was only inhibited by approximately 50%. This leaves an appreciable amount in the nucleus to drive gene expression, yet this does not occur and is not addressed.

We thank the referee for pointing this out. HSF1 binding to its targets is mediated by the cis-regulatory heat shock element (HSE). Different HSF1 targets respond to the presence of active HSF1 in the nucleus with a varying sensitivity depending on several factors, including the number and sequence of heat shock factor binding sites that are composed of three to four HSE. Cytosolic chaperones such as HSPA1A and HSPA6 harbor 3 heat shock factor binding sites in their promoters^{19,20}, while mitochondrial chaperonins such as HSPD1 and HSPE1 host a single HSF1 binding site in their regulatory region¹¹. This difference renders mitochondrial chaperonins less sensitive to active HSF1 in the nucleus compared to cytosolic chaperones. Thus, reduction of active HSF1 translocating to the nucleus upon *DNAJA1* knock-down affected mitochondrial chaperonin induction stronger than cytosolic chaperones as shown in the panel below (**Rebuttal Fig. 7**). This could explain the non-proportional ratio of reduced HSF1 translocation to the nucleus and the induction of mitochondrial chaperonins, while the changes in cytosolic chaperone induction more proportionally represent the amount of active HSF1 in the nucleus.

Rebuttal Figure 7. Barplots depicting the mean of relative transcript levels of cytosolic chaperones (left panel) and mitochondrial chaperones (right panel) in HeLa *DNAJA1* knock-down cells upon treatment with GTPP in comparison to siControl measured with qPCR ($n = 4$ biological replicates).

Given these and other issues with the manuscript, I cannot recommend publication in Nature. Although some of the conclusions seem sound, these by themselves do not represent a major advancement in the field. Other key conclusions are likely based on overinterpretation of the data and therefore may likely be incorrect. Any reconsideration of the manuscript would require substantial additions and revisions.

While we do not agree, we appreciate the referee's conceptual criticism. Since the discovery of the UPR^{mt} in 1996, it has been a puzzle how the UPR^{mt} is signalled in mammalian cells. Different factors, such as components of the integrated stress response, have been suggested and later refuted. Our work is the culmination of efforts over the last 10 years and is the first study that provides a molecular link from mitochondrial misfolding stress to signalling molecules in the cytosol and HSF1 to bring about the UPR^{mt} transcriptional response. Knowing the two signals, ROS and precursor accumulation, we were able to artificially activate the signalling pathway. We believe that revealing the UPR^{mt} signalling pathway answers one of the biggest questions in the field for the last 25 years. Our edits try to make this clearer in the manuscript. We hope that our additional experiments and clarifications answer the other issues mentioned.

Referee #2 (Remarks to the Author):

Summary of key results - Mitochondrial protein misfolding induces mitochondrial unfolded protein response (UPR^{mt}) that retrogradely upregulates nuclear genes encoding mitochondrial chaperones and proteases. This promotes the restoration of mitochondrial homeostasis under stress conditions. UPR^{mt} has been previously reported in cultured mammalian cells, although the activation of the mitochondrial proteostatic genes is generally moderate. In the current manuscript, the authors determined the mechanism by which mitochondrial misfolding stress

reprograms nuclear transcription and induces UPR_{mt}. In most experiments, gamitrinib-triphenylphosphonium (GTPP) was used to inhibit the mitochondrial HSP90 and to induce UPR_{mt}, with the goal of identifying the molecular components mediating the activation of UPR_{mt} genes.

The key findings include: (1) GTPP-induced activation of mitochondrial chaperone genes is ablated by ROS scavengers; (2) Using the H₂O₂ sensor HyPer, the authors reported that GTPP induces ROS production in the intermembrane space, which is responsible for UPR_{mt} activation; (3) Redox proteomic analysis showed that GTPP induces the oxidation of the cytosolic co-chaperone HSP40 (DNAJA1); (4) DNAJA1 depletion prevents GTPP-induced activation of mitochondrial chaperone genes; (5) GTPP treatment induces the accumulation of mitochondrial proteins in the cytosol (c-mtProt) that bind to DNAJA1 and HSP70; (6) Inhibition of global protein synthesis by cycloheximide and mitochondrial biogenesis by NRF1 depletion prevent the activation of mitochondrial chaperone genes by GTPP; (7) GTPP treatment stimulates the nuclear translocation of the HSF1 transcriptional factor that is known to regulate the mitochondrial chaperone genes; (8) GTPP treatment decreases HSP70-HSF1 interaction, releasing HSF1 that promotes nuclear translocation; (9) when antimycin and oligomycin are used to induce ROS production and to collapse membrane potential (and therefore protein import), the transcription of mitochondrial chaperone genes is also activated like does GTPP. Based on these data, the authors concluded that mitochondrial misfolding stress increases ROS release from the intermembrane space and causes c-mtProt accumulation. ROS activate DNAJA1 and c-mtProts titrate out HSP70, which in turn releases HSF1 from HSP70 to stimulate its nuclear translocation and to activate mitochondrial chaperone genes.

Overall evaluation - The manuscript comprises of a large amount of data generated by different approaches at the transcriptomic and proteomic levels. The strengths of manuscript include strong data documenting the activation of DNAJA1-HSP70-HSF1 axis by GTPP-induced ROS production and c-mtProt accumulation. However, the reviewer finds that its relevance to UPR_{mt} remains uncertain and that the conclusion of the manuscript is weakly supported by the data. Another major concern is that the experimental readouts reporting UPR_{mt} are rather moderate throughout the manuscript. This further weakens the conclusion and limits the potential impact of the study in the field especially when translating into *in vivo* systems.

Major concerns - (1) The overall validity of using the mitochondrial HSP90 inhibitor gamitrinib-triphenylphosphonium (GTPP) as a model to induce mitochondrial misfolding stress in the context of mitochondrial proteostatic signaling is questionable. This is because the concentrations of GTPP that differentially inhibits mitochondrial and cytosolic HSP90 are very narrow (<https://www.jci.org/articles/view/44855>). Although GTPP is preferentially targeted to mitochondria, it also inhibits cytosolic HSp90. It is difficult to exclude the possibility that a small fraction remains in the cytosol to induce some levels of proteostatic stress and to reduce mitochondrial protein targeting (HSP90 is required for mitochondrial protein targeting), which subsequently induces the stress response with HSF1 being translocated to the nuclear to activate mitochondrial chaperone genes. If this were the case, the stress responses detected by the authors may results from cytosolic rather than mitochondrial stress. This is particularly relevant when considering the strong activation of DDIT3 (see Extended Data Figure 1d) by GTPP. DDIT3 (or CHOP) is known to be also activated by ER stress. Inhibitions of cytosolic

Hsp90 is known to induce ER stress. Moreover, RNA-seq experiment clearly showed that ER stress is activated by GTPP (Extended Data Figure 2c). Thus, GTPP may induce the nuclear translocation of HSF1 by cytosol-originated stress rather than mitochondrial misfolding stress.

We appreciate the referee's concern about the use of GTPP in our experiments. GTPP is a well-established reagent that has been used extensively to induce robust mitochondrial protein misfolding and UPR^{mt} induction^{10,11,13,21-24}. We and others have shown that it has specific functions in mitochondria when compared to a cytosolic HSP90 inhibitor, including the article mentioned by the referee (Siegelin *et al.*) that, in regard to cytosolic effects of GTPP, only states "*G-TPP is selectively delivered to mitochondria and does not affect Hsp90 homeostasis outside the organelle, i.e., chaperone client protein stability in the cytosol.*" Nevertheless, there is always a possibility of off-target effects. Consequently, we had included other treatments in the original manuscript: CDDO, a LONP1 inhibitor, and antimycin+oligomycin, which produce ROS and c-mtProt. Both approaches confirmed the results with GTPP that the UPR^{mt} is signalled by ROS+c-mtProt via DNAJA1 and HSF1.

To further strengthen this conclusion, the revised manuscript now additionally contains data from a third compound, which targets protease HTRA2 (Ucf-101), and two genetic approaches by mitochondrial overexpression of Abeta and double knock-down of mitochondrial proteases *LONP1* and *PITRM1* in the revised manuscript. Strikingly, we found that all the additional treatments reproduced our observations made for GTPP treatment (**Fig. 4c-o**). Overall, we have shown that our findings are not only specific to GTPP treatment, but instead are applicable to UPR^{mt} induction via different mitochondrial proteostasis challenges.

Indeed, we do observe a small effect of GTPP on the UPR^{ER}. However, when compared to the induction of ER stress by thapsigargin as a positive control, the effects of GTPP on XBP1 splicing and HSPA5 transcriptional induction appear minute, indicating that GTPP does not induce a broad UPR^{ER} (**Rebuttal Fig. 8**). While the induction of ISR markers such as ATF4 and CHOP (DDIT3) is expected both in ER and mitochondrial stresses^{13,25,26}, **Rebuttal Fig. 8** shows that even with the strong induction of ER stress by thapsigargin, there is no significant induction of mitochondrial chaperones. Therefore, it is very unlikely that UPR^{mt} induction by GTPP is mediated by its cytosolic effect on proteostasis indicated by the mild induction of ER stress markers. We had seen that UPR^{mt} induction by GTPP can be inhibited by co-treatment with the VDAC1 inhibitor DIDS (**Extended Data Fig. 4g-i**), indicating that signals originating from mitochondria are required for UPR^{mt} induction.

Rebuttal Figure 8. Barplots depicting the mean of relative transcript levels of UPR^{mt} markers (left panel), ER stress markers (right panel), and markers shared between the two stresses (mid panel) in HeLa cells upon treatment with GTPP or thapsigargin measured with qPCR ($n = 3$ biological replicates).

The authors also used antimycin + oligomycin to stress mitochondria. However, these conditions are expected to collapse mitochondrial membrane potential and completely shut down mitochondrial biogenesis. I do not think that this is a good model for studying UPR^{mt}. UPR^{mt} is expected to stimulate the import of mitochondrial chaperones to restore mitochondrial homeostasis. Gene activation seen upon the shut-down of mitochondrial biogenesis may be just secondary response to global cellular damage instead of being selective for restoring mitochondrial homeostasis.

We would like to apologize for any misunderstanding we may have caused and would like to clarify the rationale for our approach. We do not propose to use antimycin A+oligomycin A as a model to study the UPR^{mt}. This treatment has not been described to induce mitochondrial protein misfolding and is thus not suitable for studying an unfolded protein response (UPR). Using GTPP, we identified ROS in combination with cytosolic precursor accumulation as the signals transferred to the cytosol to signal the UPR^{mt} upon mitochondrial protein misfolding. These findings are further supported by two other inducers of mitochondrial misfolding stress, CDDO and Ucf-101. To provide with further evidence for this signalling pathway, we aimed to artificially activate the signalling cascade, bypassing the requirement of mitochondrial protein misfolding and potential alternative factors involved. Thus, we used antimycin A+oligomycin A, which produces mitochondrial ROS and blocks mitochondrial protein import respectively without causing mitochondrial protein aggregation (**Extended Data Fig. 11i**), as artificial inducers of UPR^{mt} signalling, to confirm that both signals together are essential and sufficient to activate UPR^{mt} signalling in the cytosol.

(2) Throughout the manuscript, the activation of the mitochondrial chaperone genes by GTPP is rather moderate. Furthermore, the authors did not show whether the mitochondrial chaperones are actually increased inside mitochondria under the stress conditions. Possibilities exist that these genes are just collaterally activated by the HSF1 network. In their previous and

current (Extended Data Figure 1c) studies, the authors clearly showed that OXPHOS genes are also activated by GTPP. There is no robust data supporting the idea that the mitochondrial chaperones genes are selectively activated with their gene products trafficked to mitochondria to improve mitochondrial homeostasis. Without addressing this issue, the overall significance of the data for UPR^{mt} is questionable. In Extended Data Figure 1, the data in 1b and 1c lack statistical justification. Especially, the data are processed in different ways, with biological variability presented in 1c but not in 1b.

Mitochondrial chaperones and proteases are highly abundant and constitutively expressed in human cells. We had previously found chaperonin mRNA to be among the 20 most abundant mRNAs in cells upon GTPP treatment¹⁰. Therefore, the dynamic range is limited and UPR^{mt} activation in humans typically leads to a modest induction (1.5-3 fold) of these chaperones and proteases. However, this modest induction is robust and has been consistently observed in UPR^{mt} across different cell types and inducers, as previously reported by different publications^{10,27,28}.

Extended Data Fig. 1b-d were all taken from the same sample set and represented in the same way showing all the individual data points. As suggested by the referee, we added statistical tests to **Extended Data Fig. 1b-d**. These data support that mitochondrial chaperone and protease genes are preferentially induced, while substantial induction (fold change > 1.5, P -value ≤ 0.05) of general mitochondrial proteins, such as OXPHOS, was not observed within a 6 h GTPP treatment. In our previous study¹⁰, we observed up- and down-regulation for different OXPHOS genes upon GTPP treatment. In fact, OXPHOS components were not enriched in the transcriptomic analysis of this study. On the other hand, a previous study has shown that activation of mitochondrial biogenesis was marked with increases in transcript levels of the majority of OXPHOS components²⁹, indicating that a general increase in mitochondrial biogenesis was not the main driving force for UPR^{mt} induction. To confirm this, we compared expression of mitochondrial chaperones and proteases (UPR^{mt} genes) to other general mitochondrial proteins (including OXPHOS) and found a separate regulation between the two gene sets. Knock-out of *HSF1* reduced transcript levels of UPR^{mt} genes with minimal effect to general mitochondrial proteins (**Rebuttal Fig. 9a, Extended Data Fig. 8d**). In contrary, depletion of *NRF1*, an important mitochondria biogenesis transcription factor, reduced expression levels of general mitochondrial proteins, without a substantial effect on UPR^{mt} genes (**Rebuttal Fig. 9b, Extended Data Fig. 8e**). We have also added data to inspect different mitochondrial stressors where some of them were known to induce mitochondrial biogenesis (CCCP treatment and hypoxia)^{30,31} (**Rebuttal Fig. 9c-e, Extended Data Fig. 11j-l**). However, we found that none of these treatment was able to induce the UPR^{mt}. Altogether, our data establish that while there might be interconnection, regulation of UPR^{mt} gene expression does not directly correlate with general mitochondrial biogenesis.

Rebuttal Figure 9. **a, b,** Heat maps of relative transcript levels of indicated UPR^{mt} genes (mitochondrial chaperones and proteases) and other mitochondrial proteins upon (a) *HSF1* knock-out and (b) *NRF1* knock-down measured with qPCR. **c,** Barplot showing the mean of mitochondrial ROS ($O_2^{\cdot-}$) levels measured on FACS using MitoSOX of HeLa cells treated with different mitochondrial stressors ($n = 3$ biological replicates). **d,** Western blot images of mitochondrial protein in their precursor (p) and mature (m) forms upon treatment with different mitochondrial stressors. **e,** Barplots depicting the mean of relative transcript levels of indicated UPR^{mt} genes of HeLa cells upon treatment with different mitochondrial stressors measured with qPCR ($n = 3$ biological replicates). All P values are calculated with a two-tailed Student's t -test and represented as * $P \leq 0.05$, ** $P \leq 0.01$, and *** $P \leq 0.001$. All error bars represent mean+SD

Mitochondrial chaperones and proteases induced upon the UPR^{mt} are constitutively expressed and highly abundant. In addition, these proteins have long half-lives³², which makes it difficult to observe substantial changes on their levels inside mitochondria especially in the short time period of 6 h used in this manuscript. However, treatment with GTPP has previously been shown to increase mitochondrial chaperones in the mitochondrial fraction¹⁰, supporting the relevance of UPR^{mt} response described in this manuscript in restoring mitochondrial homeostasis.

In addition, although we agree that an increase of chaperones and proteases inside the mitochondria is ultimately necessary to resolve mitochondrial misfolding stress, our publication aims at identifying the **transcriptional response** induced by mitochondrial misfolding stress. How and when the upregulated chaperones and proteases get imported, is

another important research question as studies from various model organisms reported different and even opposing findings. In yeast, at early stages during UPR^{mt}, an increased import was observed³³. In contrast, in *C. elegans* UPR^{mt} signalling depends on a compromised import³⁴ as it was also proposed for mammalian cells²⁷. Recently, a publication from our lab by Michaelis *et al.*⁴ showed mechanistic details how misfolded mitochondrial proteins decrease mitochondrial import in human cells. The physiological consequences of this have to be addressed in the future.

To further evaluate the importance of our findings on the cytosolic surveillance mechanism driven by HSF1 for the cell survival during mitochondrial misfolding stress, we have added new experimental data. Transiently expressing MTS-EGFP, we found that precursor accumulation in the cytosol happened earlier in *HSF1* KO cells compared to WT upon GTPP treatment (**Rebuttal Fig. 10a, b, Fig. 4p, Extended Data Fig. 10c**). This precursor accumulation was accompanied by changes in mitochondrial morphology (more fragmented) in *HSF1* knock-out cells. In addition, measuring cell viability upon toxic treatment of GTPP (15 μ M for 16 h) revealed lower cell survival of *HSF1* knock-out cells compared to WT (**Rebuttal Fig. 10c-e, Extended Data Fig. 10d-f**). Importantly, this reduction in cell survival was not observed when cells were treated with a non-mito-proteotoxic specific stressor, staurosporine. Together, these observations indicate that shutting down the surveillance mechanism via *HSF1* knock-out reduces cellular robustness to mitochondrial misfolding stress, underlining the relevance of our findings that induction of the UPR^{mt} is a cellular survival program. We added new figures containing these experimental data to the revised manuscript (**Fig. 4p, Extended Data Fig. 10c-f**).

Rebuttal Figure 10. **a**, Representative microscopy images of WT and *HSF1* knock-out HeLa cells transiently expressing MTS-EGFP upon 3 h GTPP treatment with different concentrations. **b**, Degree of co-localization of MTS-EGFP and mitochondria (mitotracker) is represented as a Mander's overlap coefficient (M1) with MTS-EGFP as probe 1 and mitotracker as probe 2 ($n = 5$ biological replicates). Barplot represents the mean of Mander's overlap coefficients. **c-e**, **(c)** Representative microscopy and **(d)** western blot images of WT and *HSF1* knock-out HeLa cells treated for 16 h with DMSO, 15 μ M GTPP and 200 nM staurosporine (STS). Western blot images show the levels of apoptosis markers, including cleaved-PARP1 and cleaved-caspase 3 under the different treatments. Viability of cells under the treatments is calculated and represented as **(e)** barplot ($n = 6$ biological replicates). All P values are calculated with a two-tailed Student's t-test and represented as $*P \leq 0.05$, $** P \leq 0.01$, and $***P \leq 0.001$. All error bars represent mean+SD

(3) The reviewer is not convinced for a role of ROS produced in the intermembrane space in activating UPR^{mt}. First, the authors found that GTPP treatment induces mitochondrial proteostatic genes, which is ablated by antioxidants (Figure 2c and 2d). GTPP treatment is known to elevate ROS production. However, reduction of HSPD1, HSPE1 and HSPA9 induction by the antioxidants is rather marginal (Figure 1d). This may just reflect less mitochondrial damage in cells cotreated with antioxidants, which would attenuate stress signaling including UPR^{mt}. Although the authors did not find changes to protein aggregation levels with or without antioxidant treatment, other damage may still play a role in signaling UPR^{mt} (eg., lipid peroxidation, mtDNA damage, etc.) which are protected by antioxidants. If ROS signals UPR^{mt}, treatment with antimycin and rotenone should be able to activate the genes without GTPP. This is clearly not the case (Figure 1f).

Our model proposes that both increased ROS and c-mtProt accumulation are required for UPR^{mt} induction. Thus, treatment with ROS inducers alone (antimycin A and rotenone) is not enough to initiate UPR^{mt} response. More importantly, this also suggest that it is unlikely that UPR^{mt} induction by GTPP is mainly mediated by consequences of high ROS levels, such as lipid peroxidation, mtDNA damage, etc, which should also be observed upon treatment with other ROS inducers alone. The effect of NAC as antioxidant to inhibit UPR^{mt} induction is concentration dependent, since it depends on both the levels of ROS and antioxidant. To support this concept, we added a panel of figures to correlate the antioxidant activity of NAC and GSH with UPR^{mt} induction. Here, we could show that both NAC and GSH can efficiently reduce mitochondrial ROS production induced by GTPP treatment, and these antioxidant activities correlate with the strength of UPR^{mt} inhibition (**Rebuttal Fig. 11a-d**). These data also show that both high levels of NAC and GSH (15 mM) were able to suppress UPR^{mt} induction down to DMSO control levels, underlining the significance of ROS for UPR^{mt} induction. In addition, employing a different antioxidant, which works as superoxide dismutase mimetic (MnTBAP), resulted in the same observation (**Extended Data Fig. 3f, g**). We added new figures containing these experimental data to the revised manuscript (**Extended Data Fig. 3b-g**).

Rebuttal Figure 11. a, b, Barplots showing (a) the mean of mitochondrial ROS ($O_2^{\cdot-}$) levels measured on FACS using MitoSOX and (b) relative transcript levels of indicated UPR^{mt} genes measured with qPCR of HeLa cells upon co-treatments of GTPP with titrated concentrations of NAC ($n = 3$ biological replicates). **c, d,** Barplots showing (c) the mean of mitochondrial ROS ($O_2^{\cdot-}$) levels measured on FACS using MitoSOX and (d) relative transcript levels of indicated UPR^{mt} genes measured with qPCR of HeLa cells upon co-treatments of GTPP with titrated concentrations of GSH ($n = 3$ biological replicates). All P values are calculated with a two-tailed Student's t-test and represented as $*P \leq 0.05$, $**P \leq 0.01$, and $***P \leq 0.001$. All error bars represent mean+SD.

Secondly, the authors used H₂O₂ sensors targeted to different mitochondrial sub-compartments to decipher where ROS are produced to activate UPR^{mt}. They used inhibitors of VDAC on the outer membrane and mPTP on the inner membrane to reach the conclusion that diffusion of ROS from IMS to cytosol is responsible for UPR^{mt} activation. The reviewer finds that the rationale is weak for the experiments. The authors should make it clear that HyPer7 is a direct sensor for H₂O₂, instead of superoxide. Peroxide can diffuse through the membranes, in contrast to the negatively charged superoxide that cannot cross the membrane freely. In other words, treatment by VDAC and mPTP inhibitors may or may not affect the distribution of H₂O₂. As such, the conclusion that IMS-generated ROS play a role in activating UPR^{mt} is weak.

We thank the referee for bringing up this point. We measured superoxide to directly monitor mitochondrial ROS production with MitoSOX, and H₂O₂ to follow the flow of ROS signalling from mitochondria to the cytosol with HyPer7. To clarify this, we updated the revised manuscript and specify that MitoSOX was used to measure superoxide ($O_2^{\cdot-}$), while HyPer7 was utilized for H₂O₂ measurements.

The reasoning for this approach is as follows: To our knowledge there is no available genetic-based reporter for superoxide that can be readily manipulated to measure the levels of ROS in sub-mitochondrial compartments. In addition, H_2O_2 are the most common type of ROS used for signalling because it is less reactive and thus less damaging in comparison to superoxide³⁵. Despite the physical properties of H_2O_2 that allow them to readily diffuse through membranes, mitochondria host the thioredoxin system to reduce H_2O_2 to water. This system has been shown to prevent H_2O_2 from freely crossing mitochondrial sub-compartments^{5,36}. Pak *et al.*³⁶ has shown that treatment with rotenone, a mitochondrial superoxide inducer, specifically increased H_2O_2 in the matrix. Only when an inhibitor of the thioredoxin system, auranofin, was added, H_2O_2 from the matrix passed to the IMS and cytosol. Thus, monitoring H_2O_2 has been shown to be a useful approach to monitor mitochondrial ROS signalling and approximate relative superoxide levels.

In regard to the valuable point about VDAC: VDAC has been shown to play a role in the release of ROS from mitochondria to the cytosol³⁷. Therefore, it is plausible to assume that VDAC could mediate the ROS release from mitochondria to the cytosol upon GTPP treatment to induce the UPR^{mt}. To address this point, as suggested by referee 2, we performed Hyper7 assays to monitor the H_2O_2 levels in different mitochondrial compartments upon different treatment combinations with GTPP and the VDAC1 inhibitor DIDS. Our data show that co-treatment with VDAC inhibitor abolished the H_2O_2 flow from mitochondria to the cytosol, but did not substantially reduce levels of H_2O_2 in the IMS and matrix (**Rebuttal Fig. 12**). This observation further supports our hypothesis that ROS produced upon GTPP treatment need to reach the cytosol to initiate UPR^{mt} signalling. We added new figures containing these experimental data to the revised manuscript (**Extended Data Fig. 4i**).

Rebuttal Figure 12. Barplots showing the mean of H_2O_2 level measured with Hyper7 reporters targeted to the IMS, the matrix and untargeted (cytosol+nucleus) upon co-treatments of GTPP with DIDS ($n = 5$ biological replicates, >100 cells were analysed for each replicate). All P values are calculated with a two-tailed Student's t -test and represented as $*P \leq 0.05$, $** P \leq 0.01$, and $***P \leq 0.001$. All error bars represent mean+SD.

(4) The data supporting the conclusion that “combined accumulation of ROS and c-mtProt signal the UPR^{mt}” (Fig. 4) are misleading. Oligomycin does not block protein import. It does so only when it is combined with antimycin. In these experiments, antimycin is also used as a ROS inducer. If antimycin alone does not induce UPR^{mt}, doesn't it mean that ROS have no role? I do not think that the data allow to make a conclusion that a combined action is required

for UPR^{mt} induction. Furthermore, the quality of the data supporting a combined role of ROS accumulation and cytosolic-mtProt (c-mtProt) in inducing UPR^{mt} is poor. In Figure 4e, the z-score map is not that informative for evaluating the robustness of gene activation.

We respectfully disagree with the statement that “oligomycin does not block protein import”. Oligomycin A treatment has been shown to inhibit mitochondrial protein import previously via ATP depletion^{4,38}. We utilized oligomycin A to induce mitochondrial import inhibition, which is an essential factor for c-mtProt accumulation. Our data (**Extended Data Fig.11b-h**) show that we could not induce c-mtProt in treatment with oligomycin A alone. However, when the combination treatment of antimycin A and oligomycin A was applied, c-mtProt accumulation was observed. These observations indicate that additional aspects, such as strength of import inhibition, cytosolic protein degradation and translation, might fine-tune c-mtProt accumulation. Despite the importance to investigate these additional aspects in the future, our dataset supports the conclusion that UPR^{mt} is activated only when simultaneous induction of both ROS and c-mtProt accumulation are achieved.

We have also added new figures inspecting different mitochondrial stressors and confirmed that individual induction of ROS or precursor proteins alone could not induce the UPR^{mt} (**Rebuttal Fig. 9c-e, Extended Data Fig. 11j-l**). These data also show that similar to treatment with oligomycin A, mitochondrial import inhibition (such as by DFP⁴) does not always result in c-mtProt, indicating that while import inhibition is an important inducer, additional factor(s) could fine tune c-mtProt.

We agree with referee 2’s suggestion and changed the z-score scaling to fold change in **Fig.4c-o** to facilitate a better interpretation of the data.

Other points:

1. The data supporting a differential activation of UPR^{mt} and ISR upon GTPP treatment are weak because it does not have statistical validation (Extended data Figure 1b and 1d).

We thank referee 2 for pointing out the missing statistical test. We have performed statistical tests and added *P*-values for **Extended Data Fig. 1b-d**.

2. Extended Data Figure 1 – the authors stated that “the primary ISR factor ATF4 was induced before the UPR^{mt}...”. There is no statistical analysis to support the conclusion. Despite that the data are strong showing that activation of the proteostatic genes is independent of CHOP and ATF4, other transcriptional factors in the ISR pathway were not examined. The statement that “ISR was not required for UPR^{mt} induction” is inaccurate.

We thank referee 2 for pointing out the missing statistical test. We have performed statistical tests and added *P*-values for **Extended Data Fig. 1b-d**. The statistical tests support our conclusion in the manuscript that ISR markers such as ATF4 and CHOP (DDIT3) were induced earlier than UPR^{mt}. However, we agree with referee 2 that the use of the term “ISR” in this context could be misleading, and therefore specified the term to “the ISR-ATF4 axis” throughout the revised manuscript.

3. The specific oxidation of DNAJA1 by GTPP-induced oxidative stress is an interesting observation. The authors showed that activation of proteostatic genes by GTPP is abolished in cells with KD of DNAJA1 (Figure 2c). However, the author did not show whether DNAJA1 KD affects global proteostasis and causes cell death. Reduced UPR^{mt} activation may just be a secondary effect of cell death, rather than a specific defect in UPR^{mt} signaling. HSP40 and 70 are also required for mitochondrial protein import. It is also possible that DNAJA1 KD directly affects mitochondrial biogenesis, which in turn affects cell signaling. Thus, the conclusion that “DNAJA1 is an essential component of UPR^{mt} signaling is premature.

We have performed cell death measurement with AnnexinV+PI staining in *DNAJA1* knock-down cells to address the referee’s concerns. We found only a neglectable increase in cell death of *DNAJA1* knock-down cells in comparison to cells transfected with siRNA control (~1% increase in non-viable cells, **Rebuttal Fig. 13a, b**). In addition, we measured induction of markers for different cellular stresses in *DNAJA1* knock-down cells to evaluate if there were major changes in proteostasis induced by the knock-down (**Rebuttal Fig. 13c-e**). Our data show that there was no substantial induction of ER stress, heat shock/unfolded protein response (UPR), and the ISR markers upon *DNAJA1* knock-down (**Rebuttal Fig. 13c**) in comparison to reference treatments where the corresponding stresses were specifically induced (**Rebuttal Fig. 13d**). We also found that while there was a statistically significant reduction of OXPHOS gene expression, the effect size was small, for instance compared to *NRF1* knock-down cells, which serve as a reference in which general mitochondrial biogenesis is reduced (**Rebuttal Fig. 13e**), indicating that *DNAJA1* knock-down does not affect general mitochondrial biogenesis. More importantly, as we have shown in the previous comment (**Referee 2 comment 2, Rebuttal Fig. 9**), there is no direct correlation between activation of the UPR^{mt} through HSF1 axis and general mitochondrial biogenesis. Taken together, we conclude that it is highly unlikely that the effect of *DNAJA1* knock-down on UPR^{mt} induction was mediated by changes in overall proteostasis or mitochondrial biogenesis.

Rebuttal Figure 13. a, Changes in cell death upon GTPP treatment, measured by annexin V + propidium iodide staining and cell sorting. Viable cells are represented on the fourth quadrant (Q4), while non-viable cells on the second (Q2) and third quadrants (Q3). Staurosporine (STS) was used as a positive control treatment. The mean of non-viable cells percentage is shown as a barplot in **b** ($n = 3$ biological replicates). **c,** Barplots depicting the mean of relative transcript levels of heat shock markers (left panel), ISR markers (mid panel), and ER stress markers (right panel) in HeLa *DNAJA1* knock-down cells in comparison to siControl measured with qPCR ($n = 3$ biological replicates). **d,** Barplots depicting the mean of relative transcript levels of heat shock markers (left panel), ISR markers (mid panel), and ER stress markers (right panel) in HeLa cells upon different treatments measured with qPCR as references to panel **c** ($n = 3$ biological replicates). **e,** Barplots depicting the mean of relative transcript levels of mitochondrial proteins in HeLa *DNAJA1* knock-down cells in comparison to siControl measured with qPCR (left panel, $n = 3$ biological replicates). Relative transcript levels of mitochondrial proteins under *NRF1* knock-down (right panel) is shown as reference. All P values are calculated with a two-tailed Student's t-test and represented as * $P \leq 0.05$, ** $P \leq 0.01$, and *** $P \leq 0.001$. All error bars represent mean+SD.

4. The use of “orphan mitochondrial proteins” is unnecessary. It sounds like uncharacterized or evolutionarily un-conserved proteins. Why not just “unimported proteins” or “preproteins?”

We have changed the use of the term “orphan mitochondrial proteins” to “mitochondrial preproteins” whenever we found it is appropriate as suggested by referee 2.

5. Reference 21: author’s names are wrong.

We thank the referee for observing this mistake and fixed it.

6. Extended Data Figure 7c: no annotation for the y-axis and no explanation for the experimental groups.

We thank referee 2 for pointing out the missing labels. We have added the label for the y-axis and experimental groups for **Extended Data Figure 8c**.

7. The authors referred the study being conducted “in humans” in many places throughout the manuscript. I think using “cultured human cells” is more appropriate.

We have changed the use of the term “in humans” to “human cells” whenever we found it is appropriate as suggested by referee 2.

8. The transcriptional data in Figure 4c-e lack statistical analysis.

We have added statistical analysis to the figures as suggested by referee 2 (**Fig. 4c-o**).

9. It would be good if the authors specify the nature of experiments (e.g., qPCR, RNS-seq, etc) and the type of cells used in the text or in the figure legends, so that one does not have to check the Method section to find this out.

We have added additional information about the nature of experiments and type of cell used in the figure legends throughout the manuscript.

Referee #3 (Remarks to the Author):

This is an interesting paper that shows that mitoROS and accumulation of mitochondrial proteins in the cytosol induce a UPR^{mt} signal. This signal depends on oxidation of cytosolic HSP40 protein DNAJA1 leading to HSP70 recruitment to mitochondria concomitant with release of HSF1 to nucleus. Subsequently, HSF1 increases gene expression. Much of the data depends on an artificial way of inducing ROS and c-mtProtein accumulation to activate this pathway. I very much enjoyed the conceptual framework in this paper. However, some of the experimental approaches are not optimal.

Major points:

Figure 1:

(1) The use of NAC is not appropriate (see this PMID: 29429900). Please remove all NAC experiments. I would use Martin Brand's compounds that suppress superoxide from complex I (S1QEL1.1) and complex III (S3QEL 2)- See his excellent review PMID: 33148057. Other strategies could be to try mito-tempo.

We thank the referee for suggesting the reference about the usage of NAC as an antioxidant. Despite the important finding that NAC acts as an antioxidant indirectly, we found that the strong ROS reducing property of NAC is still relevant for our experiments. Therefore, we decided to keep the experimental data produced using NAC treatments throughout the manuscript. However, we supplemented the NAC data by using additional antioxidants to validate our findings.

We have tried to use S3QEL2 (15, 30, 60, 100, 200 μ M) and mitoTEMPO (25, 50, 100, 200, 400 μ M) to reduce ROS produced upon GTPP treatment as suggested by the referee. However, we found that both co-treatments could not reduce ROS production and thus also did not affect UPR^{mt} induction by GTPP (**Rebuttal Fig. 14a, b**). Importantly, we found that mitoTEMPO could not reduce ROS that were produced by treatment with 10 μ M of antimycin A, while S3QEL2 only caused a minor reduction (**Rebuttal Fig. 14c**). Since both antimycin A and GTPP treatment induced relatively high levels of ROS (**Extended Data Fig. 3h**), it is possible that S3QEL2 and mitoTEMPO could not efficiently inhibit this higher level of mitochondrial ROS production.

Rebuttal Figure 14. a, b, Barplots showing (a) the mean of mitochondrial ROS ($O_2^{\cdot-}$) levels measured on FACS using MitoSOX and (b) relative transcript levels of indicated UPR^{mt} genes measured with qPCR of HeLa cells upon co-treatments of GTPP with titrated concentrations of S3QEL2 and mitoTEMPO. **c,** Barplot showing the mean of mitochondrial ROS ($O_2^{\cdot-}$) levels measured on FACS using MitoSOX of HeLa cells upon co-treatments of 10 μ M antimycin A with 100 μ M S3QEL2 and 200 μ M mitoTEMPO ($n = 3$ biological replicates). All P values are calculated with a two-tailed Student's t-test and represented as * $P \leq 0.05$, ** $P \leq 0.01$, and *** $P \leq 0.001$. All error bars represent mean+SD.

Thus, we decided to supplement treatment of NAC with GSH (reduced L-glutathione), which functionally is similar to NAC as a cellular glutathione booster, and MnTBAP, which works like mitoTEMPO as a superoxide dismutase mimetic. Titration of NAC, GSH and MnTBAP all showed comparable efficiency in reducing ROS produced by GTPP treatment when they were used as co-treatments (**Rebuttal Fig. 15a, c, e**). Moreover, this ROS reduction trend correlates with the ability of these antioxidants to inhibit UPR^{mt} induction (**Rebuttal Fig. 15b, d, f**), suggesting that the capacity of NAC in modulating UPR^{mt} is indeed mediated by its antioxidant activity. We added new figures containing these experimental data to the revised manuscript (**Extended Data Fig. 3b-g**).

Rebuttal Figure 15. **a, b**, Barplots showing **(a)** the mean of mitochondrial ROS ($O_2^{\cdot-}$) levels measured on FACS using MitoSOX and **(b)** relative transcript levels of indicated UPR^{mt} genes measured with qPCR of HeLa cells upon co-treatments of GTPP with titrated concentrations of NAC ($n = 3$ biological replicates). **c, d**, Barplots showing **(c)** the mean of mitochondrial ROS ($O_2^{\cdot-}$) levels measured on FACS using MitoSOX and **(d)** relative transcript levels of indicated UPR^{mt} genes measured with qPCR of HeLa cells upon co-treatments of GTPP with titrated concentrations of GSH ($n = 3$ biological replicates). **e, f**, Barplots showing **(e)** the mean of mitochondrial ROS ($O_2^{\cdot-}$) levels measured on FACS using MitoSOX and **(f)** relative transcript levels of indicated UPR^{mt} genes measured with qPCR of HeLa cells upon co-treatments of GTPP with titrated concentrations of MnTBAP ($n = 3$ biological replicates). All P values are calculated with a two-tailed Student's t -test and represented as * $P \leq 0.05$, ** $P \leq 0.01$, and *** $P \leq 0.001$. All error bars represent mean+SD.

(2) Antimycin sustains the response while rotenone diminishes the response. This suggests that complex III is main site of superoxide production. I would complement this with using myxothiazol (500nM) and Antimycin (500nM). Myxothiazol prevents complex III superoxide production while antimycin maintains complex III superoxide production.

We thank the referee for the suggestion and performed treatments with myxothiazol to complement our data. We needed to use higher concentrations of antimycin A (2 μ M) and myxothiazol (2 μ M) than suggested to be able to measure the ROS boosting effect by antimycin

A. However, we found that co-treatment with myxothiazol did not inhibit enhanced UPR^{mt} induction resulting from the GTPP + antimycin A treatment (**Rebuttal Fig. 16a**).

Rebuttal Figure 16. **a**, Barplots showing the mean of relative transcript levels of indicated UPR^{mt} genes measured with qPCR of HeLa cells upon co-treatments of GTPP with antimycin A and myxothiazol ($n = 3$ biological replicates). **b**, **c** Barplots showing the mean of mitochondrial ROS (O₂⁻) levels measured on FACS using MitoSOX of HeLa cells upon co-treatments of (b) antimycin A and myxothiazol, and (c) GTPP and myxothiazol ($n = 3$ biological replicates). **d**, Barplots showing the mean of relative transcript levels of indicated UPR^{mt} genes measured with qPCR of HeLa cells upon co-treatments of GTPP with myxothiazol ($n = 3$ biological replicates). All P values are calculated with a two-tailed Student's t-test and represented as * $P \leq 0.05$, ** $P \leq 0.01$, and *** $P \leq 0.001$. All error bars represent mean+SD.

As control, we confirmed that myxothiazol can inhibit ROS production induced by antimycin A treatment with the concentrations that we used (**Rebuttal Fig. 16b**). Intriguingly, we found that co-treatment of GTPP+myxothiazol increased both ROS production and UPR^{mt} induction (**Rebuttal Fig. 16c, d**). These data suggest that general inhibition of complex III increases GTPP-induced ROS production.

Combining these new and previous observations we found that

- Rotenone inhibits GTPP-induced mitochondrial ROS
- General inhibition of complex III enhances GTPP-induced mitochondrial ROS
- GTPP induces relatively high levels of ROS

Taken together, these findings point to possible ROS production via the reverse electron transfer³⁹ instead of directly via complex III, as we previously hypothesized. Further investigations that are beyond the scope of this manuscript would be necessary to pin point the exact site where ROS were produced upon GTPP treatment. Therefore, we updated the text to generally refer to mitochondrially-produced ROS and accommodate the new findings following the suggestions by referee 3 (**line 81-96**).

(3) What happens to this HSF1 response if you give paraquat or menadione that will generate ROS?

We thank the referee for the suggestion. We have performed 6 h treatment with 1 mM paraquat and 10 μ M menadione in HeLa cells and found that both menadione and paraquat increased ROS production (**Rebuttal Fig. 17a**). However, we found that both treatments did not induce HSF1 translocation nor UPR^{mt} activation (**Rebuttal Fig. 17b-d**), further supporting that increased ROS alone is not sufficient to induce the UPR^{mt}. We incorporated the experimental data of the menadione treatment to the revised manuscript (**Extended Data Fig. 11j-l**).

Rebuttal Figure 17. a, Barplot showing the mean of mitochondrial ROS ($O_2^{\cdot-}$) levels measured on FACS using MitoSOX of HeLa cells upon treatments with 10 μ M menadione and 1 mM

paraquat in comparison to DMSO. **b**, Western blot images of mitochondrial proteins in their precursor (p) and mature (m) forms of HeLa cells upon GTPP, menadione and paraquat treatments. **c**, Western blot images of HSF1 in cytosolic (C) and nuclear (N) fractions in HeLa cells upon treatments with GTPP, menadione and paraquat. **d**, Barplots showing the mean of relative transcript levels of indicated UPR^{mt} genes measured with qPCR of HeLa cells upon treatments with GTPP, menadione and paraquat ($n = 3$ biological replicates). All P values are calculated with a two-tailed Student's t -test and represented as $*P \leq 0.05$, $**P \leq 0.01$, and $***P \leq 0.001$. All error bars represent mean+SD.

(4) There is no physiological or pathology response here. All artificial ways to activate this system. Hypoxia is known to increase ROS and HSF1 responses. Also, doxycycline inhibits mitochondrial translation. Does doxycycline induce this pathway? It would be important to know the answers to these important interventions.

We agree with the referee that there is no physiological or pathological response at this point. The mechanisms of how the UPR^{mt} is signalled have eluded us for over 25 years, which has severely limited our capability to study the UPR^{mt} in the context of disease. Here, we attempted to reveal the cellular mechanism of UPR^{mt} signalling as a crucial step towards links to disease.

Following the suggestion by the referee, we have performed 6 h treatment of 10 μ M doxycycline and hypoxia in HeLa cells. We found that both doxycycline and hypoxia treatments increased ROS production to a different extent (**Rebuttal Fig. 18a**). However, both treatments failed to induce c-mtProt accumulation and did consequently fail to induce UPR^{mt} activation (**Rebuttal Fig. 18b, c**). This finding further supports that increased ROS alone are not sufficient to induce the UPR^{mt}. We incorporated the experimental data of the hypoxia treatment to the revised manuscript (**Extended Data Fig. 11j-l**).

Rebuttal Figure 18. a, Barplots showing the mean of mitochondrial ROS ($O_2^{\cdot-}$) levels measured on FACS using MitoSOX of HeLa cells upon treatments with 10 μ M doxycycline and

6 h hypoxia in comparison to DMSO and untreated samples, respectively. **b**, Western blot images of mitochondrial proteins in their precursor (p) and mature (m) forms of HeLa cells upon doxycycline and hypoxia treatments. **c**, Barplots showing the mean of relative transcript levels of indicated UPR^{mt} genes measured with qPCR of HeLa cells upon treatments with doxycycline and hypoxia ($n = 3$ biological replicates). All P values are calculated with a two-tailed Student's t-test and represented as $*P \leq 0.05$, $**P \leq 0.01$, and $***P \leq 0.001$. All error bars represent mean+SD.

To establish a pathological setting inducing UPR^{mt}, we employed overexpression of amyloid-beta (Abeta), as this peptide contributes to the toxicity of Alzheimer's disease by its accumulation within the mitochondria. This accumulation leads to energy depletion, ROS production and impaired mitochondrial import^{40,41}. We fused Abeta with an N-terminal mitochondrial targeting sequence and a C-terminal GFP and confirmed the mitochondrial localization of the fusion protein. Despite the mild induction of UPR^{mt} by transient overexpression of Abeta in HeLa cells for 24h, this activation is dependent on both DNAJA1 and HSF1 (**Rebuttal Fig. 3, Fig. 4k, l, Extended Data Fig. 10a, b**).

In addition, we have also added experimental data to induce the UPR^{mt} via simultaneous depletion of mitochondrial proteases, LONP1 and PITRM1. Depletion of these proteases had previously been used to induce UPR^{mt}^{4,16,17} and loss of function mutations in these proteases have been found in diseases^{16,42}. Strikingly, we found that this genetic induction of UPR^{mt} also depends on both DNAJA1 and HSF1 (**Rebuttal Fig. 4, Fig. 4m-o**). This underlines the importance and relevance of our findings for the understanding of UPR^{mt} signalling in humans under both physiological and diseases contexts. We added new figures containing these experimental data to the revised manuscript (**Fig. 4k-o, Extended Data Fig. 10a, b**).

(5) “We induced ROS production in the IMS using antimycin A and triggered c-mtProt accumulation by blocking mitochondrial protein import with the ATP synthase inhibitor oligomycin.” This is peculiar statement and results that accompanied this reasoning. Wade Harper and Richard Youle use combination of antimycin and oligomycin to depolarize mitochondrial membrane potential that will induce mitophagy and block protein import. But depolarized mitochondrial membrane potential decrease ROS as well (see Michael Murphy excellent review: PMID: 19061483). Not sure how to make sense of this experiment.

This is an interesting point brought up by the referee that has fascinated us as well. During the last author's postdoc time in Wade Harper's lab, they started using antimycin A/oligomycin A as a very potent mitophagy inducer, without however knowing exactly how this treatment induces mitophagy. This is in fact something we are trying to find out at the moment. What our data have shown so far is that the transcriptional UPR^{mt} is not required to induce mitophagy and mitophagy induction does not generally lead to UPR^{mt} induction.

Here, after having found that ROS and c-mtProt are required to signal the UPR^{mt}, we wanted to validate this mechanism by artificially inducing UPR^{mt} through direct induction of ROS and c-mtProt, skipping the need to induce mitochondrial protein misfolding. An antimycin A and oligomycin A co-treatment was able to do so.

(6) Please use ISRIB for ISR inhibition. PMID: 30674674

We thank the referee for the suggestion and have performed co-treatment experiments of GTPP and ISRIB to evaluate the effect of the ISR to UPR^{mt} induction. As shown in **Rebuttal Fig. 19**, inhibition of the ISR with ISRIB reduced UPR^{mt} induction mildly, suggesting that, while the ATF4 axis of the ISR is not required for the UPR^{mt}, different aspects of the ISR might still influence UPR^{mt} activation. Considering this result, we specified the axis that was referred to in the manuscript and changed “the ISR” term to “the ISR-ATF4 axis”.

Rebuttal Figure 19. Barplots showing the mean of relative transcript levels of indicated UPR^{mt} genes measured with qPCR of HeLa cells upon co-treatments of GTPP with ISRIB ($n = 3$ biological replicates). All P values are calculated with a two-tailed Student’s t-test and represented as $*P \leq 0.05$, $**P \leq 0.01$, and $***P \leq 0.001$. All error bars represent mean+SD.

1. Stanford, K. R. *et al.* Antimycin A-induced mitochondrial dysfunction activates vagal sensory neurons via ROS-dependent activation of TRPA1 and ROS-independent activation of TRPV1. *Brain Res.* **1715**, 94–105 (2019).
2. Wang, P. *et al.* Macrophage achieves self-protection against oxidative stress-induced ageing through the Mst-Nrf2 axis. *Nat. Commun.* **10**, 755 (2019).
3. Han, Y. H., Kim, S. H., Kim, S. Z. & Park, W. H. Antimycin A as a mitochondria damage agent induces an S phase arrest of the cell cycle in HeLa cells. *Life Sci.* **83**, 346–355 (2008).
4. Michaelis, J. B. *et al.* Protein import motor complex reacts to mitochondrial misfolding by reducing protein import and activating mitophagy. *Nat. Commun.* **13**, 5164 (2022).
5. Hoehne, M. N. *et al.* Spatial and temporal control of mitochondrial H₂O₂ release in intact human cells. *EMBO J.* **41**, e109169 (2022).
6. Harding, O. & Holzbaur, E. L. F. Damaged mitochondria recruit the effector NEMO to activate NF- κ B signaling. *bioRxiv* 2022.06.21.496850 (2022) doi:10.1101/2022.06.21.496850.
7. Li, H.-S. *et al.* HIF-1 α protects against oxidative stress by directly targeting mitochondria. *Redox Biol.* **25**, 101109 (2019).
8. Choi, H.-I. *et al.* Redox-regulated cochaperone activity of the human DnaJ homolog

- Hdj2. *Free Radic. Biol. Med.* **40**, 651–659 (2006).
9. Lee, J. *et al.* Inhibition of mitochondrial LonP1 protease by allosteric blockade of ATP binding and hydrolysis via CDDO and its derivatives. *J. Biol. Chem.* **298**, 101719 (2022).
 10. Münch, C. & Harper, J. W. Mitochondrial unfolded protein response controls matrix pre-RNA processing and translation. *Nature* **534**, 710–713 (2016).
 11. Katiyar, A. *et al.* HSF1 is required for induction of mitochondrial chaperones during the mitochondrial unfolded protein response. *FEBS Open Bio* **10**, 1135–1148 (2020).
 12. Kafkova, A. *et al.* Selective and Reversible Disruption of Mitochondrial Inner Membrane Protein Complexes by Lipophilic Cations. *bioRxiv* 2021.03.09.434520 (2021) doi:10.1101/2021.03.09.434520.
 13. Quirós, P. M. *et al.* Multi-omics analysis identifies ATF4 as a key regulator of the mitochondrial stress response in mammals. *J. Cell Biol.* **216**, 2027–2045 (2017).
 14. Rousseau, E. *et al.* Targeting expression of expanded polyglutamine proteins to the endoplasmic reticulum or mitochondria prevents their aggregation. *Proc. Natl. Acad. Sci. U. S. A.* **101**, 9648–9653 (2004).
 15. Deshwal, S., Fiedler, K. U. & Langer, T. Mitochondrial Proteases: Multifaceted Regulators of Mitochondrial Plasticity. *Annu. Rev. Biochem.* **89**, 501–528 (2020).
 16. Pérez, M. J. *et al.* Loss of function of the mitochondrial peptidase PITRM1 induces proteotoxic stress and Alzheimer’s disease-like pathology in human cerebral organoids. *Mol. Psychiatry* **26**, 5733–5750 (2021).
 17. Lu, B. *et al.* LonP1 Orchestrates UPR^{mt} and UPR^{ER} and Mitochondrial Dynamics to Regulate Heart Function. *bioRxiv* 564492 (2019) doi:10.1101/564492.
 18. Karlsson, M. *et al.* A single-cell type transcriptomics map of human tissues. *Sci. Adv.* **7**, (2021).
 19. Ortner, V., Ludwig, A., Riegel, E., Dunzinger, S. & Czerny, T. An artificial HSE promoter for efficient and selective detection of heat shock pathway activity. *Cell Stress Chaperones* **20**, 277–288 (2015).
 20. Ramirez, V. P., Stamatis, M., Shmukler, A. & Aneskievich, B. J. Basal and stress-inducible expression of HSPA6 in human keratinocytes is regulated by negative and positive promoter regions. *Cell Stress Chaperones* **20**, 95–107 (2015).
 21. Schäfer, J. A., Bozkurt, S., Michaelis, J. B., Klann, K. & Münch, C. Global mitochondrial protein import proteomics reveal distinct regulation by translation and translocation machinery. *Mol. Cell* (2021) doi:10.1016/j.molcel.2021.11.004.
 22. Siegelin, M. D. *et al.* Exploiting the mitochondrial unfolded protein response for cancer therapy in mice and human cells. *J. Clin. Invest.* **121**, 1349–1360 (2011).
 23. Wang, N. *et al.* Suppressing TRAP1 sensitizes glioblastoma multiforme cells to temozolomide. *Exp Ther Med* **22**, 1246 (2021).
 24. Fiesel, F. C., James, E. D., Hudec, R. & Springer, W. Mitochondrial targeted HSP90 inhibitor Gamitrinib-TPP (G-TPP) induces PINK1/Parkin-dependent mitophagy.

- Oncotarget* **8**, 106233–106248 (2017).
25. Fessler, E. *et al.* A pathway coordinated by DELE1 relays mitochondrial stress to the cytosol. *Nature* **579**, 433–437 (2020).
 26. Nishitoh, H. CHOP is a multifunctional transcription factor in the ER stress response. *J. Biochem.* **151**, 217–219 (2012).
 27. Fiorese, C. J. *et al.* The Transcription Factor ATF5 Mediates a Mammalian Mitochondrial UPR. *Curr. Biol.* **26**, 2037–2043 (2016).
 28. Mohrin, M. *et al.* Stem cell aging. A mitochondrial UPR-mediated metabolic checkpoint regulates hematopoietic stem cell aging. *Science* **347**, 1374–1377 (2015).
 29. Mak, T. W., Hauck, L., Grothe, D. & Billia, F. p53 regulates the cardiac transcriptome. *Proc. Natl. Acad. Sci.* **114**, 2331–2336 (2017).
 30. Tohme, S. *et al.* Hypoxia mediates mitochondrial biogenesis in hepatocellular carcinoma to promote tumor growth through HMGB1 and TLR9 interaction. *Hepatology* **66**, 182–197 (2017).
 31. Itami, N., Shiratsuki, S., Shirasuna, K., Kuwayama, T. & Iwata, H. Mitochondrial biogenesis and degradation are induced by CCCP treatment of porcine oocytes. *Reproduction* **150**, 97–104 (2015).
 32. Schwanhäusser, B. *et al.* Global quantification of mammalian gene expression control. *Nature* **473**, 337–342 (2011).
 33. Poveda-Huertes, D. *et al.* An Early mtUPR: Redistribution of the Nuclear Transcription Factor Rox1 to Mitochondria Protects against Intramitochondrial Proteotoxic Aggregates. *Mol. Cell* **77**, 180-188.e9 (2020).
 34. Nargund, A. M., Pellegrino, M. W., Fiorese, C. J., Baker, B. M. & Haynes, C. M. Mitochondrial import efficiency of ATFS-1 regulates mitochondrial UPR activation. *Science* **337**, 587–590 (2012).
 35. Sies, H. & Jones, D. P. Reactive oxygen species (ROS) as pleiotropic physiological signalling agents. *Nat. Rev. Mol. Cell Biol.* **21**, 363–383 (2020).
 36. Pak, V. V *et al.* Ultrasensitive Genetically Encoded Indicator for Hydrogen Peroxide Identifies Roles for the Oxidant in Cell Migration and Mitochondrial Function. *Cell Metab.* **31**, 642-653.e6 (2020).
 37. Han, D., Antunes, F., Canali, R., Rettori, D. & Cadenas, E. Voltage-dependent anion channels control the release of the superoxide anion from mitochondria to cytosol. *J. Biol. Chem.* **278**, 5557–5563 (2003).
 38. Eilers, M., Oppliger, W. & Schatz, G. Both ATP and an energized inner membrane are required to import a purified precursor protein into mitochondria. *EMBO J.* **6**, 1073–1077 (1987).
 39. Scialò, F., Fernández-Ayala, D. J. & Sanz, A. Role of Mitochondrial Reverse Electron Transport in ROS Signaling: Potential Roles in Health and Disease . *Frontiers in Physiology* vol. 8 (2017).
 40. Chen, J. X. & Yan, S. S. Role of mitochondrial amyloid-beta in Alzheimer's disease. *J. Alzheimers. Dis.* **20 Suppl 2**, S569-78 (2010).

41. Reddy, P. H. & Beal, M. F. Amyloid beta, mitochondrial dysfunction and synaptic damage: implications for cognitive decline in aging and Alzheimer's disease. *Trends Mol. Med.* **14**, 45–53 (2008).
42. Xu, Z. *et al.* Disuse-associated loss of the protease LONP1 in muscle impairs mitochondrial function and causes reduced skeletal muscle mass and strength. *Nat. Commun.* **13**, 894 (2022).

Reviewer Reports on the First Revision:

Referees' comments:

Referee #1 (Remarks to the Author):

In this revision, the authors provide an impressive amount of new data and sufficiently clarify earlier misconceptions. I provided challenging critiques to the original manuscript, which I did not feel offered a compelling case for publication in Nature. However, I am satisfied with their thorough response and find the current version to be more robust and impressive. Moreover, the authors do a better job at articulating how this work stitches together disparate areas of a complicated field to arrive at a unified model. Overall, I congratulate the authors on their efforts and have no further critiques.

Referee #2 (Remarks to the Author):

Summary of key results

UPRmt has been previously reported in cultured mammalian cells. The underlying mechanism is not well understood. In this study, the authors determined the mechanism by which mitochondrial protein misfolding stress (MMS) reprograms nuclear transcription and induces UPRmt. In most experiments, gamitrinib-triphenylphosphonium (GTPP) was used to inhibit the mitochondrial HSP90 and to induce UPRmt, with the goal of identifying the molecular components mediating the activation of UPRmt genes. The key findings include (1) GTPP induces ROS production in the intermembrane space of mitochondria, which oxidises specific cysteine and activates the cytosolic co-chaperone HSP40 (DNAJA1); (2) GTPP induces the accumulation of mitochondrial proteins in the cytosol (c-mtProt) that bind to DNAJA1 and HSP70; (3) These events decrease HSP70-HSF1 interaction, releasing HSF1 that promotes its nuclear translocation. This consequently activates the UPRmt-responsive genes that encode mitochondrial chaperones and proteases. (4) The authors concluded that activation of UPRmt is dependent on both ROS and c-mtProt, and neither of these two events can activate UPRmt alone.

Concerns

1) A major concern in the initial review was the validity of using the HSP90 inhibitor gamitrinib-triphenylphosphonium (GTPP) as a model to induce mitochondrial misfolding stress and cytosolic adaptive responses, and the relatively moderate responses induced by GTPP. Although GTPP is preferentially targeted to mitochondria, it also inhibits cytosolic Hsp90 to some extent. Possibility therefore exists that subtle inhibition of cytosolic Hsp90 affects cytosolic proteostasis and/or mitochondrial protein import, which subsequently induces or contributes to the stress response with HSF1 being translocated to the nuclear to activate mitochondrial chaperone genes. To exclude a potential effect of GTPP on the cytosolic proteostasis which subsequently causes HSF1 activation, a validation experiment would be to disrupt the mitochondrial target of GTPP, TRAP1. This will help to determine whether the UPRmt target genes can still be fully activated by GTPP. This has been done in a previous paper (Munch and Harper, Nature 534:710; extended Data Figure 1f), but the data were not particularly strong. During the revision, the authors added new data showing that various conditions that cause mitochondrial stress, including the double knockdown of LONP1 and PITRM1, activate UPRmt in a HSF1-dependent manner. These are all good data, suggesting that severe mitochondrial damage does induce UPRmt via HSF1 activation. Regarding the current manuscript, it is fine with me as long as the authors attenuate the tone when describing the GTPP-based experiments and appreciate the limitations. I feel more comfortable to emphasize the new data instead of those from the GTPP-based experiments.

2) Another major concern has been the antimycin+oligomycin condition that was used to support the idea that both ROS generation and c-mtProt accumulation are required for UPR induction. The

authors showed that only a combination of antimycin and oligomycin, but not individually, can induce UPRmt. This was interpreted as to antimycin-induced ROS production and the collapse of membrane potential and protein import by a combined action of antimycin and oligomycin are both required for the activation of UPRmt. This reasoning is wrong. Antimycin+oligomycin shuts down mitochondrial biogenesis. Although the authors showed increased ROS production in these cells (Extended Data Figure 11a), there is no guaranty that they are generated by mitochondrial inner membrane space because the electron transport chain is supposed to be lost. The authors have added additional ROS and c-mtProt stressors and found that none of them can induce UPRmt alone, which supports their double-hit hypothesis. In my view, they should leave the double hit hypothesis open at the end of the manuscript by omitting the antimycin+oligomycin experiment. If they want to include this experiment, they can only claim that c-mtProts induced by antimycin+oligomycin can specifically contribute to the activation of UPRmt. The statement for a role of IMS-based ROS production in the activation process in this particular experiment is unfounded. Of course, this opens a new question as to what really makes the difference between antimycin+oligomycin and other conditions that block protein import (such as CCCP and DFP treatment) but do not activate UPRmt. Drugs such as CCCP are known to have off-targets that may blunt stress responses, but this does not have to be addressed in the current study.

Referee #3 (Remarks to the Author):

I am satisfied.

Author Rebuttals to First Revision:

We would like to thank all the referees for the time spent and their comments and suggestions to improve the manuscript. Our responses are in blue font.

Referee #1 (Remarks to the Author):

In this revision, the authors provide an impressive amount of new data and sufficiently clarify earlier misconceptions. I provided challenging critiques to the original manuscript, which I did not feel offered a compelling case for publication in Nature. However, I am satisfied with their thorough response and find the current version to be more robust and impressive. Moreover, the authors do a better job at articulating how this work stitches together disparate areas of a complicated field to arrive at a unified model. Overall, I congratulate the authors on their efforts and have no further critiques.

We are very happy that the reviewer appreciates our efforts and supports publication of our findings.

Referee #2 (Remarks to the Author):

Summary of key results

UPRmt has been previously reported in cultured mammalian cells. The underlying mechanism is not well understood. In this study, the authors determined the mechanism by which mitochondrial protein misfolding stress (MMS) reprograms nuclear transcription and induces UPRmt. In most experiments, gamitrinib-triphenylphosphonium (GTPP) was used to inhibit the mitochondrial HSP90 and to induce UPRmt, with the goal of identifying the molecular components mediating the activation of UPRmt genes. The key findings include (1) GTPP induces ROS production in the intermembrane space of mitochondria, which oxidises specific cysteine and activates the cytosolic co-chaperone HSP40 (DNAJA1); (2) GTPP induces the accumulation of mitochondrial proteins in the cytosol (c-mtProt) that bind to DNAJA1 and HSP70; (3) These events decrease HSP70-HSF1 interaction, releasing HSF1 that promotes its nuclear translocation. This consequently activates the UPRmt-responsive genes that encode mitochondrial chaperones and proteases. (4) The authors concluded that activation of UPRmt is dependent on both ROS and c-mtProt, and neither of these two events can activate UPRmt alone.

We would like to amend that, based on the comments during the first round of revision, our revised manuscript referred to ROS produced in mitochondria, without specifying the sub-mitochondrial compartment.

Concerns

1) A major concern in the initial review was the validity of using the HSP90 inhibitor gamitrinib-triphenylphosphonium (GTPP) as a model to induce mitochondrial misfolding stress and cytosolic adaptive responses, and the relatively moderate responses induced by GTPP. Although GTPP is preferentially targeted to mitochondria, it also inhibits cytosolic Hsp90 to some extent. Possibility therefore exists that subtle inhibition of cytosolic Hsp90 affects cytosolic proteostasis and/or mitochondrial protein import, which subsequently induces

or contributes to the stress response with HSF1 being translocated to the nuclear to activate mitochondrial chaperone genes. To exclude a potential effect of GTPP on the cytosolic proteostasis which subsequently causes HSF1 activation, a validation experiment would be to disrupt the mitochondrial target of GTPP, TRAP1. This will help to determine whether the UPR^{mt} target genes can still be fully activated by GTPP. This has been done in a previous paper (Munch and Harper, Nature 534:710; extended Data Figure 1f), but the data were not particularly strong. During the revision, the authors added new data showing that various conditions that cause mitochondrial stress, including the double knockdown of LONP1 and PITRM1, activate UPR^{mt} in a HSF1-dependent manner. These are all good data, suggesting that severe mitochondrial damage does induce UPR^{mt} via HSF1 activation. Regarding the current manuscript, it is fine with me as long as the authors attenuate the tone when describing the GTPP-based experiments and appreciate the limitations. I feel more comfortable to emphasize the new data instead of those from the GTPP-based experiments.

We appreciate the reviewer's comment and agree that in all treatments, there is a risk of side effects, which can influence observations. As pointed out, we included several new treatments that confirmed our findings, during the first round of revision. Now, we made extensive textual changes to shorten the manuscript and rebalance the focus on the different stress conditions, as suggested.

2) Another major concern has been the antimycin+oligomycin condition that was used to support the idea that both ROS generation and c-mtProt accumulation are required for UPR induction. The authors showed that only a combination of antimycin and oligomycin, but not individually, can induce UPR^{mt}. This was interpreted as to antimycin-induced ROS production and the collapse of membrane potential and protein import by a combined action of antimycin and oligomycin are both required for the activation of UPR^{mt}. This reasoning is wrong. Antimycin+oligomycin shuts down mitochondrial biogenesis. Although the authors showed increased ROS production in these cells (Extended Data Figure 11a), there is no guaranty that they are generated by mitochondrial inner membrane space because the electron transport chain is supposed to be lost. The authors have added additional ROS and c-mtProt stressors and found that none of them can induce UPR^{mt} alone, which supports their double-hit hypothesis. In my view, they should leave the double hit hypothesis open at the end of the manuscript by omitting the antimycin+oligomycin experiment. If they want to include this experiment, they can only claim that c-mtProts induced by antimycin+oligomycin can specifically contribute to the activation of UPR^{mt}. The statement for a role of IMS-based ROS production in the activation process in this particular experiment is unfounded. Of course, this opens a new question as to what really makes the difference between antimycin+oligomycin and other conditions that block protein import (such as CCCP and DFP treatment) but do not activate UPR^{mt}. Drugs such as CCCP are known to have off-targets that may blunt stress responses, but this does not have to be addressed in the current study.

We apologize for any confusion we may have caused and attempted to explain these experiments more clearly in the revised version of the manuscript.

As also pointed out by reviewer 2 in comment 1, the requirement for both ROS and c-mtProt for UPR^{mt} activation is supported by data from different UPR^{mt} inducers presented in the manuscript, including GTPP, antimycin+oligomycin, CDDO, Ucf-101 (**Fig. 1d, 4c, d**,

Extended Data Fig. 3b-g, 11a-i). In addition, the data from different mitochondrial stressors confirmed that induction of ROS or c-mtProt alone is not sufficient to activate the UPR^{mt} (**Extended Data Fig. 11j-l**). Therefore, we are convinced that both ROS and c-mtProt are required for UPR^{mt} activation in human cells.

Treatments with OXPHOS inhibitors, such as antimycin A and rotenone, shut down mitochondrial electron transport chain (ETC). However, these treatments have been established as potent inducers of mitochondrial ROS, especially when they are used acutely¹⁻³. While oligomycin A inhibits respiration, it does not directly inhibit electron transport chain. Treatment with antimycin A, on the other hand, inhibits ETC from complex III to cytochrome c but not the supply of electrons to complex III. Thus, this inhibition leads to ROS production via complex III. This mechanism would explain increased mitochondrial ROS production we measured in HeLa cells treated with antimycinA+oligomycinA for 6h. Importantly, this observation is not unique to our study but has been described previously^{4,5}.

To avoid a potential misinterpretation and not to overstate the experiment, we now use these data to underline the importance of the two signals for UPR^{mt} signalling, which bypasses the requirement for mitochondria misfolding stress. We feel that this revised version still conveys the importance of both ROS and c-mtProt as mitochondria-derived signals for UPR^{mt} activation, without over-emphasizing their role as part of a double hit model or excluding the possibility of additional signals.

Referee #3 (Remarks to the Author):

I am satisfied.

We are happy our revisions satisfied the reviewer's requests.

References

1. Quinlan, C. L., Gerencser, A. A., Treberg, J. R. & Brand, M. D. The Mechanism of Superoxide Production by the Antimycin-inhibited Mitochondrial Q-cycle*. *J. Biol. Chem.* **286**, 31361–31372 (2011).
2. Hoehne, M. N. *et al.* Spatial and temporal control of mitochondrial H₂O₂ release in intact human cells. *EMBO J.* **41**, e109169 (2022).
3. Pak, V. V *et al.* Ultrasensitive Genetically Encoded Indicator for Hydrogen Peroxide Identifies Roles for the Oxidant in Cell Migration and Mitochondrial Function. *Cell Metab.* **31**, 642-653.e6 (2020).
4. Liu, X. *et al.* The lysosomal membrane protein LAMP-2 is dispensable for PINK1/Parkin-mediated mitophagy. *FEBS Lett.* **594**, 823–840 (2020).
5. Kataura, T. *et al.* NDP52 acts as a redox sensor in PINK1/Parkin-mediated mitophagy. *EMBO J.* **42**, e111372 (2023).

Reviewer Reports on the Second Revision:

Referees' comments:

Referee #2 (Remarks to the Author):

The revision has improved the manuscript, with more balanced interpretation of the data. It is now acceptable for publication.